# Instantaneous movement-unrelated midbrain activity modifies ongoing eye movements

Antimo Buonocore[1,2†*], Xiaoguang Tian[1,2†], Fatemeh Khademi[1,2], Ziad M Hafed[1,2]

[1]Werner Reichardt Centre for Integrative Neuroscience, Tübingen University, Tübingen, Germany; [2]Hertie Institute for Clinical Brain Research, Tübingen University, Tübingen, Germany

**Abstract** At any moment in time, new information is sampled from the environment and interacts with ongoing brain state. Often, such interaction takes place within individual circuits that are capable of both mediating the internally ongoing plan as well as representing exogenous sensory events. Here, we investigated how sensory-driven neural activity can be integrated, very often in the same neuron types, into ongoing saccade motor commands. Despite the ballistic nature of saccades, visually induced action potentials in the rhesus macaque superior colliculus (SC), a structure known to drive eye movements, not only occurred intra-saccadically, but they were also associated with highly predictable modifications of ongoing eye movements. Such predictable modifications reflected a simultaneity of movement-related discharge at one SC site and visually induced activity at another. Our results suggest instantaneous readout of the SC during movement generation, irrespective of activity source, and they explain a significant component of kinematic variability of motor outputs.

*For correspondence:
antimo.buonocore@cin.uni-tuebingen.de

†These authors contributed equally to this work

Competing interests: The authors declare that no competing interests exist.

## Introduction

A hallmark of the central nervous system is its ability to process an incredibly complex amount of incoming information from the environment in parallel. This is achieved through multiplexing of functions, either at the level of individual brain areas or even at the level of individual neurons themselves. For example, in different motor modalities like arm (*Alexander and Crutcher, 1990*; *Shen and Alexander, 1997*; *Breveglieri et al., 2016*) or eye (*Goldberg and Wurtz, 1972b*; *Goldberg and Wurtz, 1972a*; *Wurtz and Goldberg, 1972*; *Mohler and Wurtz, 1976*; *Bruce and Goldberg, 1985*; *Jagadisan and Gandhi, 2019*) movements, a large fraction of the neurons contributing to the motor command are also intrinsically sensory in nature, hence being described as sensory-motor neurons. In this study, we aimed to investigate the implications of such sensory and motor multiplexing using vision and the oculomotor system as our model of choice.

A number of brain areas implicated in eye movement control, such as the midbrain superior colliculus (SC) (*Wurtz and Albano, 1980*; *Munoz and Wurtz, 1995*), frontal eye fields (FEF) (*Bruce and Goldberg, 1985*; *Schall and Hanes, 1993*; *Schall et al., 1995*; *Tehovnik et al., 2000*), and lateral intra-parietal area (LIP) (*Mazzoni et al., 1996*), contain many so-called visual-motor neurons. These neurons burst both in reaction to visual stimuli entering into their response fields (RF's) as well as in association with triggering eye movements towards these RF's. In some neurons, for example in the SC (*Mohler and Wurtz, 1976*; *Mays and Sparks, 1980*; *Edelman and Goldberg, 2001*; *Willeke et al., 2019*), even the motor bursts themselves are contingent on the presence of a visual target at the movement endpoint. In the laboratory, the properties of visual and motor bursts are frequently studied in isolation, by dissociating the time of visual onsets (evoking 'visual' bursts) from the time of saccade triggering (evoking 'motor' bursts). However, in real life, exogenous sensory

events can happen at any time in relation to our own ongoing internal state. Thus, 'visual' spikes at one visual field location may, in principle, be present at the same time as 'motor' spikes for a saccade to another location. What are the implications of such simultaneity? Answering this question is important to clarify mechanisms of readout from circuits in which functional multiplexing is prevalent.

In the SC, our focus here, there have been many debates about how this structure contributes to saccade control (*Waitzman et al., 1991*; *Smalianchuk et al., 2018*). In recent proposals (*Goossens and Van Opstal, 2006*; *van Opstal and Goossens, 2008*; *Goossens and van Opstal, 2012*), it was suggested that every spike emitted by SC neurons during their 'motor' bursts contributes a mini-vector of movement tendency, such that the aggregate sum of all output spikes is read out by downstream structures to result in a given movement trajectory. However, implicit in these models is the assumption that only action potentials within a narrow time window around movement triggering (the 'motor' burst) matter. Any other spiking, by the same or other neurons, before or after the eye movement is irrelevant. This causes a significant readout problem, since downstream neurons do not necessarily have the privilege of knowing which spikes should now count for a given eye movement implementation and which not (*Jagadisan and Gandhi, 2019*).

Indeed, from an ecological perspective, an important reason for multiplexing could be exactly to maintain flexibility to rapidly react to the outside world, even in a late motor control structure, and there is rich behavioral evidence for this (*Miles et al., 1986*; *Gellman et al., 1990*; *Masson and Perrinet, 2012*; *Buonocore et al., 2016*; *Buonocore et al., 2019*). In that sense, rather than invoking mechanisms that allow actively ignoring 'other spiking' activity outside of the currently triggered eye movement (whether spatially or temporally), one would predict that SC readout, at any one moment, should be quite sensitive to any spiking activity regardless of its source.

We experimentally tested this hypothesis. We 'injected' SC spiking activity around the time of saccade generation, but at a spatially dissociated location, similar in principle to dual-site suprathreshold SC microstimulation to alter saccade metrics (*Katnani and Gandhi, 2011*; *Katnani et al., 2012*). We found that the entire landscape of SC activity, not just at the movement burst site, can instantaneously contribute to individual saccade metrics, and in a lawful manner, thus explaining a component of behavioral variability previously unaccounted for. Interestingly, the detailed properties of such contribution depend on the location of the movement-unrelated activity on the SC topographic map relative to the movement burst location. This places important constraints on existing models of saccade generation by the SC, and also allows generating new testable hypotheses about the functional role of SC motor bursts in general.

## Results

### Stimulus-driven SC 'visual' bursts can occur intra-saccadically

We first tested the hypothesis that visually-induced action potentials can occur in the SC intra-saccadically; that is, putatively simultaneously with motor-related bursts. We exploited the topographic nature of the SC in representing visual and motor space (*Cynader and Berman, 1972*; *Robinson, 1972*; *Chen et al., 2019*). We asked two monkeys to maintain steady fixation on a central spot. Prior work has shown that this condition gives rise to frequent microsaccades, which are associated with movement-related bursts in the rostral region of the SC representing small visual eccentricities and movement vectors (*Hafed et al., 2009*; *Hafed and Krauzlis, 2012*; *Chen et al., 2019*; *Willeke et al., 2019*). In experiments 1 and 2, we then presented a visual stimulus at a more eccentric location, and we recorded neural activity from SC sites representing this location (*Figure 1*). For experiment 1, the stimulus consisted of a vertical sine wave grating of 2.2 cycles/deg spatial frequency and variable contrast (*Chen et al., 2015*; Materials and methods). For experiment 2, the stimulus consisted of a high contrast vertical gabor grating of variable spatial frequency and constant contrast (*Khademi et al., 2020*; Materials and methods). Depending on the timing of the visual burst relative to a given microsaccade, we could measure visual burst strength (in both visual and visual-motor neurons; Materials and methods) either in isolation of microsaccades or when a microsaccade was in-flight. If SC visual bursts could still occur intra-saccadically, then one would expect that visual burst strength should be generally similar whether the burst timing happened when a microsaccade was being triggered or not. We ensured that all sites did not simultaneously burst for microsaccade

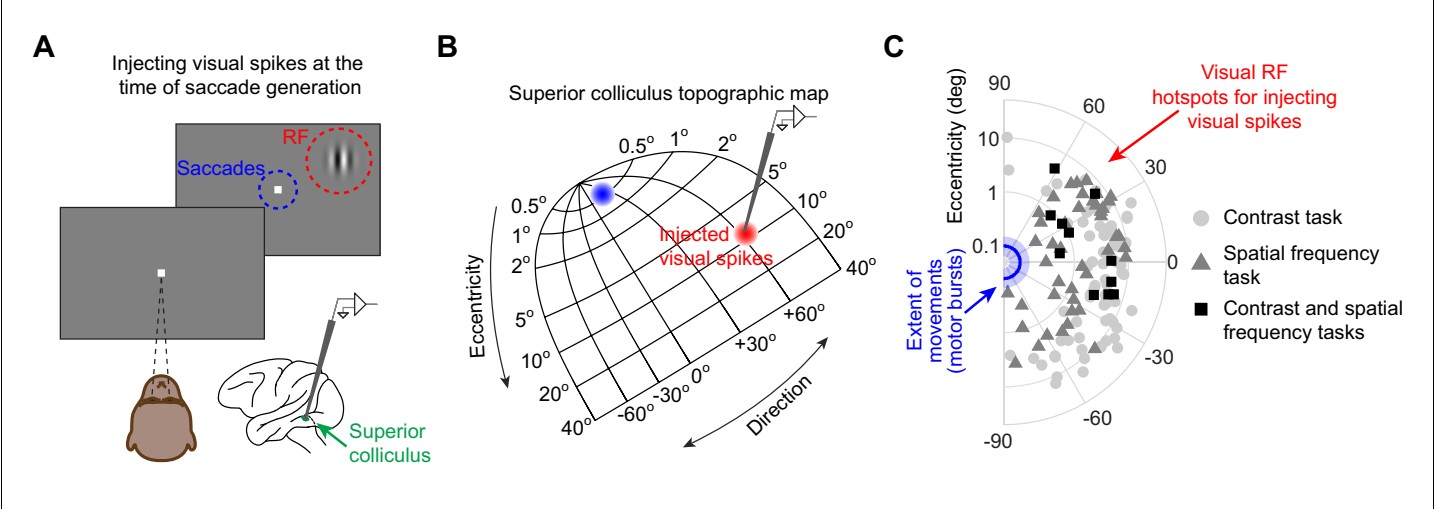

**Figure 1.** Injecting arbitrary, movement-unrelated spiking activity into the SC map around the time of saccade generation. (**A**) A monkey steadily fixated while we presented an eccentric stimulus in a recorded neuron's RF (red). In experiment 1, the stimulus consisted of a vertical grating of 2.2 cycles/deg spatial frequency, and the stimulus contrast was varied across trials (***Chen et al., 2015***). In experiment 2, the stimulus consisted of a high contrast vertical grating having either 0.56, 2.2, or 4.4 cycles/deg spatial frequency (***Khademi et al., 2020***). The stimulus location was spatially dissociated from the motor range of microsaccades being generated (blue). This allowed us to experimentally inject movement-unrelated 'visual' spikes into the SC map around the time of microsaccade generation. (**B**) We injected 'visual' spikes at eccentric retinotopic locations (red) distinct from the neurons that would normally exhibit motor bursts for microsaccades (blue). The shown SC topographic map is based on our earlier dense mappings revealing both foveal and upper visual field tissue area magnification (***Hafed et al., 2021***). (**C**) Across experiments 1 and 2, we measured 'visual' spikes from a total of 128 neurons with RF hotspots indicated by the symbols. The blue line and shaded area denote the mean and 95% confidence interval, respectively, of all microsaccade amplitudes that we observed. The neurons in which we injected 'visual' spikes (symbols) were not involved in generating these microsaccades (***Figure 1—figure supplement 1***; also see ***Khademi et al., 2020***). The origin of the shown log-polar plot corresponds to 0.03 deg eccentricity (***Hafed and Krauzlis, 2012***). Across experiments 1 and 2, 11 neurons were run on both experiments, 73 neurons were run on only experiment 1, and 44 neurons were run on only experiment 2.

The online version of this article includes the following source data and figure supplement(s) for figure 1:

**Source data 1.** Excel table with the source data for this figure.

**Figure supplement 1.** Injected 'visual' spikes in our experiments were in neurons that were not directly involved in generating the microsaccades that were being altered in our main analyses.

**Figure supplement 1—source data 1.** Excel table with the source data for this figure.

generation (***Figure 1C***; ***Figure 1—figure supplement 1***), to ensure that we were only measuring visual bursts and not concurrent movement-related activity. Such movement-related activity was expectedly in more rostral SC sites, representing foveal visual eccentricities (***Chen et al., 2019***), as we also explicitly demonstrate in our experiment three described later.

Regardless of microsaccade direction, 'visual' bursts could still occur in the SC even if there was an ongoing eye movement. To illustrate this, ***Figure 2A*** shows the stimulus-driven visual burst of an example neuron from experiment one with and without concurrent microsaccades. The neuron had a preferred eccentricity of 3.4 deg, and the stimulus in this case consisted of a vertical sine wave grating of 40% or 80% contrast (Materials and methods). The spike rasters in the figure are color-coded depending on whether there were no microsaccades around the visual stimulus onset (gray) or whether there were movements in the same session that temporally overlapped (even partially) with the interval of visual burst occurrence (red); we defined this visual burst interval (for the current study) to be 30–100 ms, and this was chosen based on the firing rate curves also shown in the same figure (bottom). The gray firing rate curve shows average firing rate when there were no microsaccades from −100 ms to +150 ms relative to stimulus onset, and the red curve shows average firing rate when the visual burst (shaded interval) coincided with at least a part of an ongoing microsaccade. As can be seen, intra-saccadic 'visual' bursts could still occur, and they were similar in strength to saccade-free visual bursts (t(92) = −0.43, p=0.67 for a t-test on peak firing rate after stimulus onset with and without microsaccades). This was also true regardless of microsaccade direction relative to the RF location (indicated in the figure by the color-coded horizontal lines in the rasters,

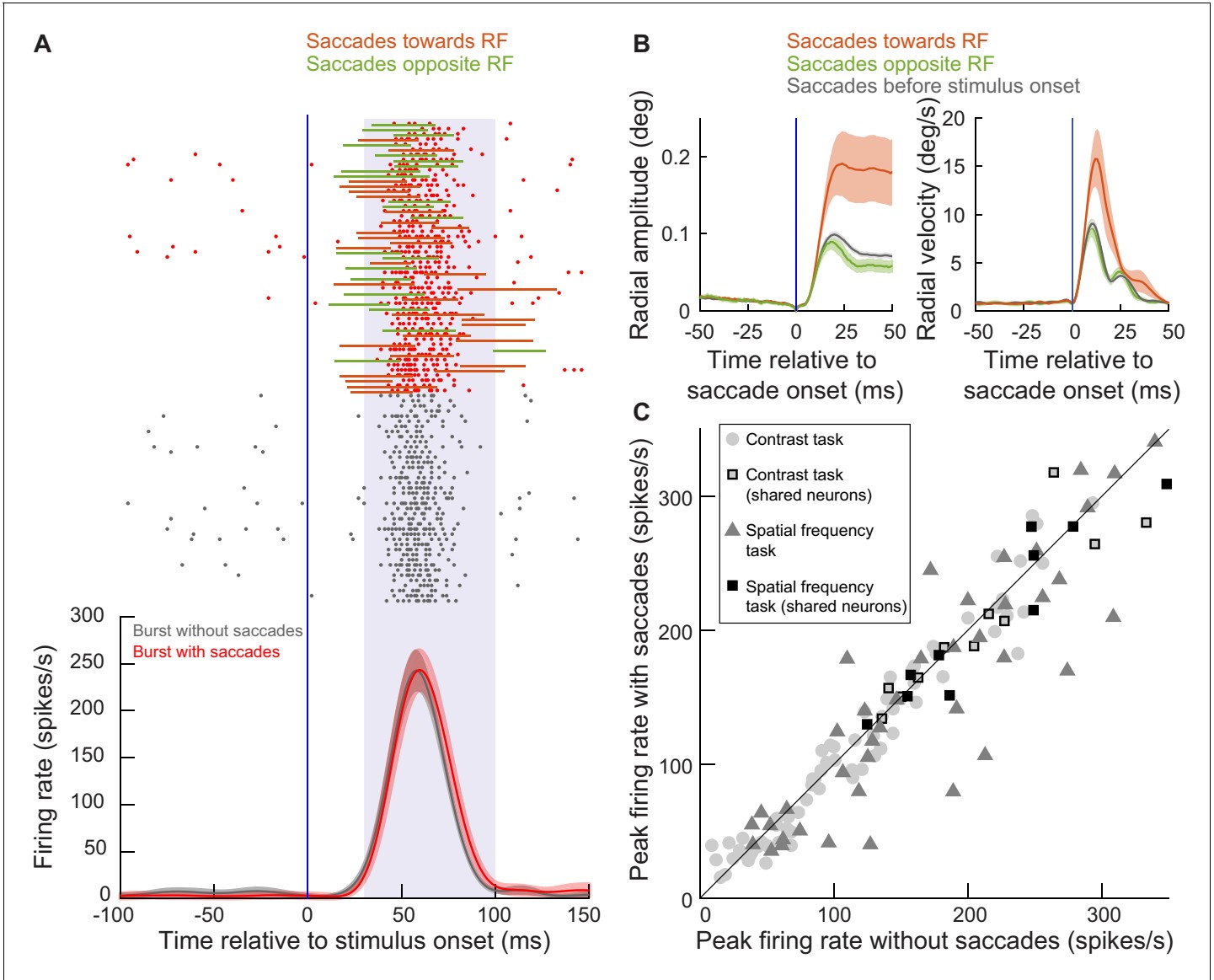

**Figure 2.** SC visual bursts still occurred intra-saccadically. (**A**) We measured the firing rate of an example neuron (from experiment 1) when a stimulus appeared inside its RF without any nearby microsaccades (gray firing rate curve and spike rasters) or when the same stimulus appeared while microsaccades were being executed around the time of visual burst occurrence (red firing rate curve and spike rasters). The stimulus eccentricity was 3.4 deg. For the red rasters, each trial also has associated with it an indication of microsaccade onset and end times relative to the visual burst (horizontal lines; colors indicate whether the microsaccade was towards the RF or opposite it as per the legend). For all of the movements, the visual burst overlapped with at least parts of the movements. Error bars denote 95% confidence intervals, and the shaded region between 30 and 100 ms denotes our estimate of visual burst interval. There was no statistically significant difference between peak firing rate with and without microsaccades (p=0.67, t-test). The numbers of trials and microsaccades can be inferred from the rasters. (**B**) For the same example session in **A**, we plotted the mean radial amplitude (left) and mean radial eye velocity (right) for the microsaccades towards or opposite the RF in **A**. The black curves show baseline microsaccade amplitude and peak velocity (for movements occurring within 100 ms before stimulus onset). Movements towards the RF were increased in size when they coincided with a peripheral visual burst; our subsequent analyses provide a mechanism for this increase. Opposite microsaccades are also shown, and they were slightly truncated. Error bars denote s.e.m. (**C**) At the population level, we plotted peak firing rate with saccades detected during a visual burst (y-axis) or without saccades around the visual burst (x-axis). The different symbols show firing rate measurements in either experiment 1 (contrast task) or experiment 2 (spatial frequency task); all neurons from each experiment are shown. Note that some neurons were run on both tasks sequentially in the same session (*Figure 1*), resulting in a larger number of symbols than total number of neurons.

The online version of this article includes the following source data and figure supplement(s) for figure 2:

**Source data 1.** Excel table with the source data for this figure.

**Figure supplement 1.** Visual bursts in the SC could happen intra-saccadically whether the movement being generated was towards the recorded neurons' RF locations or opposite them.

*Figure 2 continued on next page*

*Figure 2 continued*

**Figure supplement 1—source data 1.** Excel table with the source data for this figure.

which highlight movements either towards or away from the RF location). Therefore, intra-saccadic 'visual' bursts are possible.

Interestingly, the microsaccades temporally coinciding with visual burst occurrence in this example session had clearly different metrics from baseline microsaccades, and in a manner that depended on their direction relative to the RF location (*Figure 2B*). Movements toward the RF location were increased in size (but they were still an order of magnitude smaller than stimulus eccentricity); movements opposite the RF location appeared truncated (they were slightly reduced in size despite a smaller reduction in their peak velocity) (*Buonocore et al., 2016*; *Buonocore et al., 2017*). These behavioral observations are consistent with earlier reports (*Hafed and Ignashchenkova, 2013*; *Buonocore et al., 2016*; *Buonocore et al., 2017*; *Tian et al., 2018*), and the remainder of the current study provides a detailed mechanistic account for them. These observations also occurred for for peripheral stimuli more eccentric than 3.4 deg, as we elaborate shortly.

Across the entire population of neurons recorded from both experiments 1 and 2, we found that 'visual' bursts in the SC could occur intra-saccadically. For each neuron, we plotted in *Figure 2C* peak firing rate after stimulus onset when there was a concurrent microsaccade being generated as a function of peak firing rate when there was no concurrent microsaccade. For this analysis, we pooled trials from the highest three contrasts (20%, 40%, and 80%) in experiment one for simplicity (Materials and methods), but similar conclusions could also be reached for individual stimulus contrasts. Similarly, for the neurons in experiment 2, we also pooled trials from all spatial frequencies (Materials and methods). Note that some neurons were collected in both experiments (Materials and methods), meaning that there are more data points in *Figure 2C* than actual neurons (as indicated in the figure legend). As can be seen, intra-saccadic 'visual' bursts in the SC could still clearly occur. Statistically, we compared all points in *Figure 2C* and found mild, but significant, modulations of visual burst strength (t(137) = 2.842, p=0.005). Moreover, SC visual bursts could still occur intra-saccadically whether the stimulus was activating the same SC side generating a given movement or the opposite SC side (*Figure 2—figure supplement 1*). However, expectedly (*Chen et al., 2015*), there were modulations in visual burst strength that depended on microsaccade direction relative to the RF location. This is consistent with (*Chen et al., 2015*), although that study aligned microsaccades to stimulus, rather than burst, onset (meaning that it studied slightly earlier microsaccades than the ones that we were interested in here).

Therefore, at the time of movement execution (that is, at the time of a movement-related burst in one part of the SC map; here, the foveal representation associated with microsaccades), it is possible to have spatially dissociated visual bursts in another part of the map. We next investigated how such additional 'visual' spikes (at an unrelated spatial location relative to the movements) affected the eye movements that they were coincident with (similar to the example situation that happened in *Figure 2B*). We also studied whether there was an impact of spatial disparity between the locus of the additional spikes and the movement endpoints.

## Peri-saccadic stimulus-driven 'visual' bursts systematically influence eye movement metrics

If 'visual' bursts can be present somewhere on the SC map at a time when 'motor' bursts elsewhere on the map are to be read out by downstream neurons, then one might expect that each additional 'visual' spike on the map should contribute to the executed movement metrics and cause a change in saccades. This would suggest a highly lawful relationship between the strength of the peri-saccadic 'visual' burst and the amount of eye movement alteration that is observed. We explored this by relating the behavioral properties of the saccades in our task to the temporal relationship between their onset and the presence of 'visual' spikes in the SC map caused by an unrelated stimulus onset.

We first confirmed a clear general relationship between microsaccade amplitudes and eccentric stimulus onsets, like shown in the example session of *Figure 2B* (*Hafed and Ignashchenkova, 2013*; *Buonocore et al., 2017*; *Tian et al., 2018*; *Malevich et al., 2020b*). Our stimuli in experiment 1

consisted of vertical sine wave gratings having different luminance contrasts (Materials and methods). We plotted the time course of microsaccade amplitudes relative to grating onset for microsaccades that were spatially congruent with grating location (that is, having directions towards grating location; Materials and methods). For the present analysis, we only focused on stimulus eccentricities of ≤4.5 deg because these had the strongest effects on microsaccades (*Figure 3*); in later analyses, we also explicitly explored the farther eccentricities in more detail, and we found similar results that we describe shortly. As expected (*Hafed and Ignashchenkova, 2013*; *Buonocore et al., 2017*; *Tian et al., 2018*; *Malevich et al., 2020b*), there was a transient increase in microsaccade amplitude approximately 80–90 ms after grating onset (*Figure 3A*). Critically, the increase reflected the stimulus properties, because it was stronger with higher stimulus contrast (main effect of contrast: $F(2,713) = 81.55$, $p<1.27427*10^{-32}$), and there were also different temporal dynamics: amplitude increases occurred earlier for higher (~75 ms) than lower (~85 ms) contrasts. The increases also occurred, but to a lesser extent, for more eccentric peripheral stimuli (*Figure 3— figure supplement 1*).

Because we had simultaneously recorded neural data, we then analyzed, for the same trials, the SC visual bursts that were associated with the appearing gratings in these sessions (*Figure 3B*). For simplicity, we included all trials (regardless of eye movements) in the illustration of *Figure 3B*, especially since the visual bursts were largely unaffected whether they occurred intra-saccadically or without any saccadic movements (*Figure 2C*). The visual bursts started earlier, and were stronger, for higher stimulus contrasts (*Figure 3B*; *Li and Basso, 2008*; *Marino et al., 2012*; *Chen et al., 2015*), similar to the amplitude changes in the microsaccades (*Figure 3A*). Moreover, the timing of the microsaccadic effects (*Figure 3A*) was similar to the timing of the SC visual bursts (*Figure 3B*), showing a short lag of ~20 ms relative to the bursts that is consistent with an efferent processing delay from SC neurons to the final extraocular muscle drive (*Miyashita and Hikosaka, 1996*; *Jagadisan and Gandhi, 2017*; *Smalianchuk et al., 2018*).

Interestingly, when we experimentally altered the properties of the SC visual bursts by using different stimulus properties, namely spatial frequencies in experiment 2, similar analyses to *Figure 3* on microsaccade amplitudes also revealed altered influences on the movements themselves. Specifically, different spatial frequencies are known to give rise to different response strengths and response latencies in SC visual bursts (*Chen et al., 2018*; *Khademi et al., 2020*). Consistent with this, the time courses of microsaccade amplitudes reflected clear dependencies on spatial frequency (*Figure 3—figure supplement 2A,C*). Moreover, there was again a dependence of effects on the eccentricity of the visual bursts (*Figure 3—figure supplement 2B,D*).

Therefore, as we hypothesized in previous reports (*Hafed and Ignashchenkova, 2013*; *Buonocore et al., 2017*; *Malevich et al., 2020b*), not only is it possible for SC visual bursts to occur intra-saccadically (*Figure 2*), but such bursts are temporally aligned with concurrent changes in microsaccade amplitudes (*Figure 3*). We next uncovered a highly lawful impact of each injected extra 'spike' per recorded neuron on saccade metrics.

## There is a linear relationship between intra-saccadic 'visual' spikes and eye movement amplitude increases

The number of extra 'visual' spikes per recorded neuron occurring intra-saccadically was linearly related to metric alterations in microsaccades. For each eye movement toward the recently appearing stimulus (that is, congruent with stimulus location), we counted how many 'visual' spikes by the concurrently recorded neuron occurred in the interval 0–20 ms after movement onset. That is, we tested for the impact of the number of extra 'visual' spikes by a given recorded neuron as the SC population was being read out, intra-saccadically, by downstream pre-motor and motor structures to execute the currently triggered movement. This per-neuron spike count was a proxy for how adding additional 'visual' spikes in the SC population at a site unrelated to the movement vector can 'leak' downstream when the saccade gate is opened; this is, in fact, the reason why we picked such a strict intra-saccadic period of 0–20 ms after movement onset (subsequent analyses explored the full time course of impacts expected from movement-unrelated SC activity on the eye movement metrics). Moreover, since the extra spikes were more eccentric than the sizes of the congruent microsaccades (*Figure 1*), we expected that the contribution would act to increase microsaccade amplitudes (as in *Figure 3A*). We focused, for now, on neurons at eccentricities ≤ 4.5 deg (but still more eccentric than microsaccade amplitude; *Figure 1B,C*) because our earlier analyses showed that

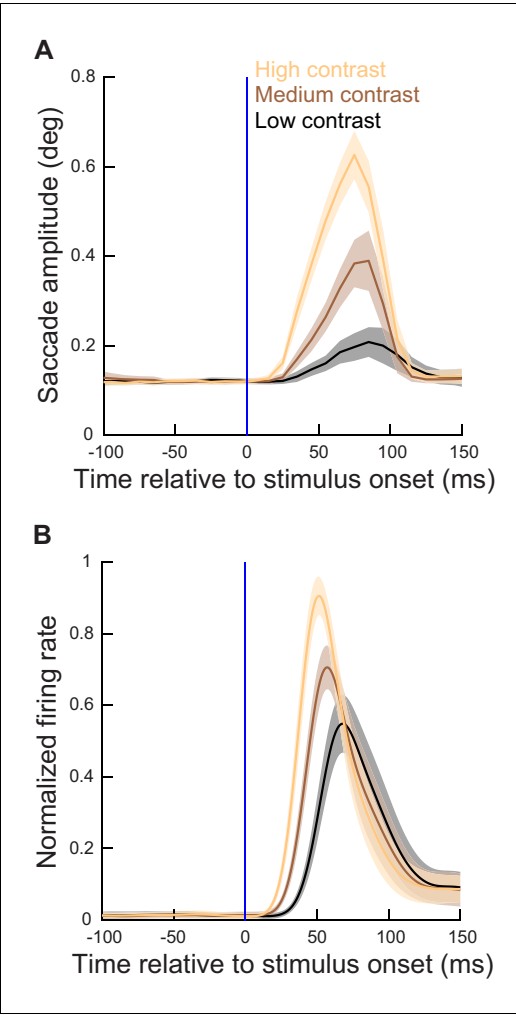

**Figure 3.** Microsaccade metrics were altered when the movements coincided with SC visual bursts, and the alteration was related to SC visual burst strength. (A) Time course of microsaccade amplitude in the contrast task (experiment 1) relative to stimulus onset (for neurons with eccentricities ≤ 4.5 deg). The data were subdivided according to stimulus contrast (three different colors representing the three highest contrasts in our task). Movement amplitudes were small (microsaccades) in the baseline pre-stimulus interval, but they sharply increased after stimulus onset, reaching a peak at around 70–80 ms. Moreover, the metric alteration clearly depended on stimulus contrast. N = 288, 206, and 222 microsaccades for the highest, second highest, and lowest contrast, respectively. (B) Normalized firing rates relative to stimulus onset for the extra-foveal neurons (≤4.5 deg preferred eccentricity) that we recorded simultaneously in experiment one with the eye movement data in A. The alterations in movement metrics in A were strongly related, in both time and amplitude, with the properties of the SC visual bursts. *Figure 3—figure supplement 1* shows the results obtained from more eccentric neurons and stimuli (>4.5 deg), and

the clearest metric changes to tiny microsaccades occurred under these circumstances (*Figure 3*, *Figure 3—figure supplements 1–3*).

We found a clear, lawful relationship between the amount of 'extra' spikes that occurred intra-saccadically and movement metrics. These spikes were unrelated to the originally planned 'motor' burst; they were spatially dissociated but temporally coincident with saccade triggering, and they were also driven by an exogenous visual stimulus onset. To demonstrate this observation, we plotted in *Figure 4A* the average microsaccadic eye movement trajectory in the absence of any additional SC 'visual' bursts during experiment 1 (dark red; the curve labeled 0 spikes; Methods). These '0 spike' microsaccades were, like all other movements in *Figure 4A*, movements that occurred shortly after stimulus onset (Materials and methods); they just happened to not have any 'visual' spikes occurring during the first 20 ms of their execution. These microsaccades were also all towards the eccentric RF location. We then plotted average microsaccade size whenever any given recorded eccentric neuron had a visual burst such that 1 spike of this visual burst happened to occur in the interval 0–20 ms after movement onset (red; 1 spike). The amplitude of the microsaccade was significantly larger than with 0 spikes. We then progressively looked for movements with 2, 3, 4, or 5 'visual' spikes per recorded neuron; there were progressively larger and larger microsaccades (*Figure 4A*). Therefore, for microsaccades towards the eccentric RF location, there was a lawful relationship between intra-saccadic 'visual' spikes and movement amplitude.

Across all data from experiment 1, the number of 'visual' spikes (per recorded neuron) that occurred intra-saccadically was monotonically and linearly driving the amplitude increase of the (smaller) saccades (*Figure 4B*) (Towards condition, F-statistic vs. constant model: F = 426, p<0.0001; estimated coefficients: intercept = 0.14253, t = 28.989, p<0.0001; slope = 0.066294, t = 20.644, p<0.0001); this relationship also held when we excluded the '0 spike' movements from the analysis. Incidentally, the peak velocities of the movements also increased systematically (*Figure 4C*), consistent with previous behavioral observations (*Buonocore et al., 2017*). On the other hand, microsaccades directed opposite to the RF (*Figure 4B*) did not show a similar large positive slope; and a trend for a negative slope was not statistically significant (Opposite condition, F-statistic vs. constant model: F = 2.22, p=0.137;

*Figure 3 continued*

*Figure 3—figure supplement 2* shows similar observations from the spatial frequency task (experiment 2). A subsequent figure (*Figure 4—figure supplement 3*) described the full dependence on eccentricity in our data. Error bars denote 95% confidence intervals.

The online version of this article includes the following source data and figure supplement(s) for figure 3:

**Source data 1.** Excel table with the source data for this figure.

**Figure supplement 1.** More eccentric stimuli relative to the generated movement amplitudes had weaker effects on metric alterations than the less eccentric stimuli of *Figure 3*.

**Figure supplement 1—source data 1.** Excel table with the source data for this figure.

**Figure supplement 2.** Results similar to those in *Figure 3* and *Figure 3—figure supplement 1* but with the spatial frequency task (experiment 2).

**Figure supplement 2—source data 1.** Excel table with the source data for this figure.

**Figure supplement 3.** Despite smaller effects on microsaccade amplitudes (*Figure 3—figure supplements 1* and *2*), more eccentric visual bursts were not weaker than more central ones.

**Figure supplement 3—source data 1.** Excel table with the source data for this figure.

estimated coefficients: intercept = 0.1185, t = 56.261, p<0.0001; slope = −0.0028872, t = −1.488, p=0.137). This suggests that it is difficult to reduce microsaccade size below the already small amplitude of these tiny eye movements (*Hafed, 2011*).

These results suggest that there is an instantaneous specification of saccade metrics described by the overall activity present on the SC map, and they provide a much more nuanced view of the correlations between SC visual bursts and microsaccade amplitudes shown in *Figures 2B* and *3*. Every SC spike matters: all activity happening intra-saccadically and at locations of the SC map different from the saccade endpoint goal is interpreted as part of the motor command by downstream neurons. Most interestingly, visual spiking activity in even purely visual neurons was still positively correlated with increased microsaccade amplitudes, although the effect was weaker than that of visual spiking activity in the deeper visual-motor neurons of the SC (*Figure 4—figure supplement 1*). This difference between visual and visual-motor neurons makes sense in hindsight: the visual-motor neurons are presumed to be much closer to the output of the SC than the visual neurons (*Mohler and Wurtz, 1976*).

We also considered the same analyses as in *Figure 4* (that is, with congruent movements, and also including the '0 spike' trials) but for more eccentric SC 'visual' bursts (*Figure 4—figure supplement 2*). The effects were still present but with a notably smaller slope than in *Figure 4*, suggesting that the distance of the 'extra' spiking activity on the SC map from the planned movement vector matters (Towards condition, F-statistic vs. constant model: F = 45.1, p<2.03*10$^{-11}$; estimated coefficients: intercept = 0.12368, t = 57.252, p<0.0001; slope = 0.0098876, t = 6.7195, p<2.024*10$^{-11}$). This observation, while still showing that every spike matters, is not predicted by recent models of saccade generation by the SC (*Goossens and Van Opstal, 2006*; *van Opstal and Goossens, 2008*; *Goossens and van Opstal, 2012*), which do not necessarily implement any kind of local versus remote interactions in how the SC influences saccade trajectories through individual spike effects.

In fact, we found an almost sudden change in local versus remote interactions in terms of readout. Specifically, for all microsaccades towards the RF location from experiment 1, we repeated analyses similar to *Figure 4B*, but now taking neuronal preferred eccentricity into account. We added eccentricity to our generalized linear model analysis, and there was a significant interaction between eccentricity and the number of injected spikes (slope = −0.0080709, t = −14.585, p<0.0001): the slope of the relationship between the number of 'injected' visual spikes and microsaccade amplitude decreased as a function of increasing eccentricity. To visualize this, we created a running average of neuronal preferred eccentricity. For each eccentricity range, we then re-analyzed the data as we did for *Figure 4B*. In all cases, there was a linear relationship between each additional 'injected' visual spike and microsaccade amplitude (*Figure 4—figure supplement 3A*), consistent with *Figure 4B*. However, the slope of the relationship decreased with increasing eccentricity. This is better demonstrated in *Figure 4—figure supplement 3B*, in which we plotted the slope parameter of the generalized linear model as a function of eccentricity. For eccentricities larger than approximately 4–5 deg, there was a weaker impact of additional 'injected' visual spikes on microsaccade amplitudes than for smaller eccentricities (which justifies our choice in other figures to focus on neurons with preferred eccentricities ≤ 4.5 deg). However, and most critically, the slope always remained positive. This means that there was never a negative impact of 'injected' visual spikes on microsaccade

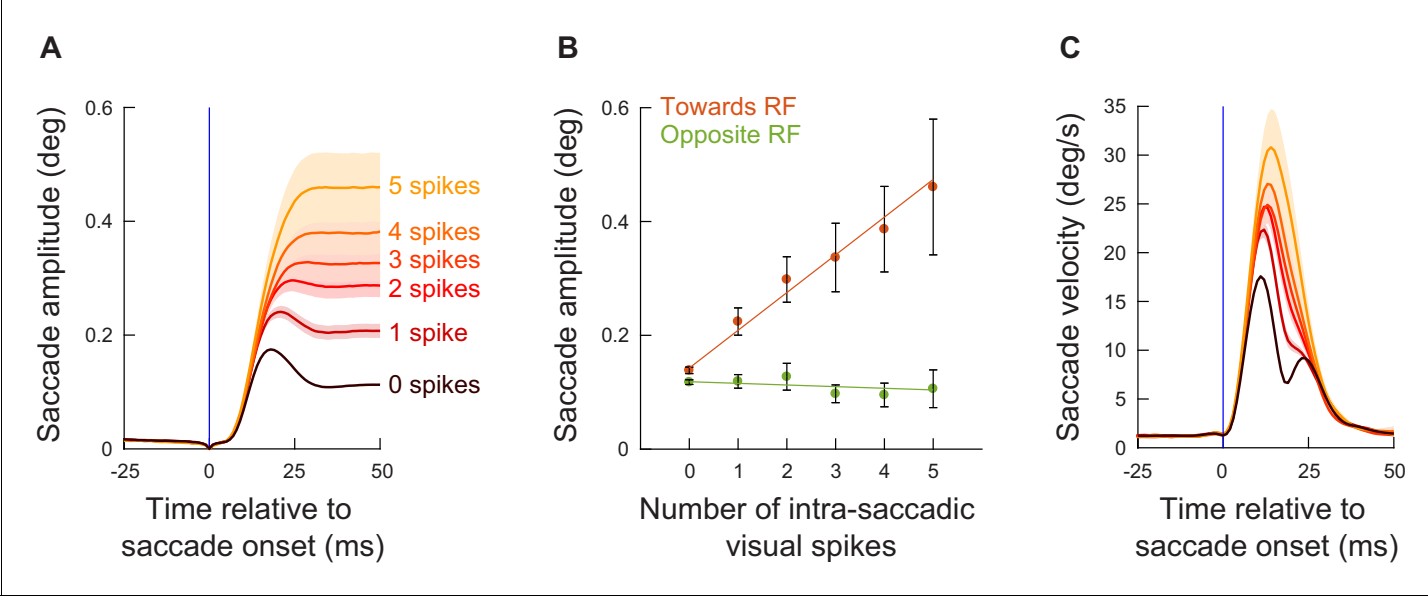

**Figure 4.** The number of exogenous, movement-unrelated 'visual' spikes to occur intra-saccadically linearly added to the executed movement's amplitude. (A) For every recorded neuron from experiment 1 (*Figure 3A,B*) and every microsaccade to occur near the visual burst interval (*Figure 2*), we counted the number of spikes recorded from the neuron that occurred intra-saccadically (0–20 ms after movement onset). We did this for movements directed towards the RF location (*Figure 1C*; Materials and methods). We then plotted radial eye position (aligned to zero in both the x- and y-axes) relative to saccade onset after categorizing the movements by the number of intra-saccadic spikes. When no spikes were recorded during the eye movement, saccade amplitudes were small (darkest curve). Adding 'visual' spikes into the SC map during the ongoing movements systematically increased movement amplitudes. Error bars denote s.e.m. (B) To summarize the results in A, we plotted mean saccade amplitude against the number of intra-saccadic 'visual' spikes for movements directed towards the RF locations (faint red dots). There was a linear increase in amplitude with each additional spike per recorded neuron (orange line representing the best linear fit of the underlying raw data). Even intra-saccadic spikes from visual neurons (more dissociated from the motor output of the SC than visual-motor neurons) were still associated with increased amplitudes (*Figure 4—figure supplement 1*). For movements opposite the RF locations (faint green dots and green line), there was no impact of intra-saccadic 'visual' spikes on movement amplitudes. The numbers of movements contributing to each x-axis value are 1772, 383, 237, 145, 113, and 78 (towards) or 1549, 238, 104, 63, 36, 23 (opposite) for 0, 1, 2, 3, 4, and 5 spikes, respectively. (C) For the movements towards the RF locations (A), peak radial eye velocities also increased, as expected (*Buonocore et al., 2017*). Error bars denote one standard error of the mean (A, C) and 95% confidence intervals (B). *Figure 4—figure supplement 2* shows results for intra-saccadic spikes from more eccentric neurons (>4.5 deg), and *Figure 4—figure supplement 3* shows the full dependence on neuronal preferred eccentricity. Finally, *Figure 4—figure supplement 4* shows the same analyses of B but for the data from experiment 2.

The online version of this article includes the following source data and figure supplement(s) for figure 4:

**Source data 1.** Excel table with the source data for this figure.
**Figure supplement 1.** Same analysis as in *Figure 4B* (for movements toward RF's), but separating visual and visual-motor neurons.
**Figure supplement 1—source data 1.** Excel table with the source data for this figure.
**Figure supplement 2.** Intra-saccadic 'visual' spikes from more eccentric neurons in experiment one still linearly increased microsaccade amplitudes, but with a much weaker effect size.
**Figure supplement 2—source data 1.** Excel table with the source data for this figure.
**Figure supplement 3.** Injected visual spikes always increased microsaccade amplitudes, but the effectiveness was diminished with larger neuronal eccentricities.
**Figure supplement 3—source data 1.** Excel table with the source data for this figure.
**Figure supplement 4.** The analysis of *Figure 4B* but during the spatial frequency task (experiment 2).
**Figure supplement 4—source data 1.** Excel table with the source data for this figure.

amplitudes. The readout always involved 'adding' to the movement amplitude. We also confirmed this conclusion with yet more data from experiment three having eccentric stimuli, as we discuss shortly when describing the results from that experiment (e.g. Figures 8–12), and also with additional analyses of experiments 1 and 2 (e.g. see Figure 6 below).

As stated above, the observations of *Figure 4—figure supplement 3* are not easy to reconcile with recent models of SC readout for saccadic eye movements (*Goossens and Van Opstal, 2006*; *van Opstal and Goossens, 2008*; *Goossens and van Opstal, 2012*). They are also not easily

reconcilable with classic vector averaging models as well (*Lee et al., 1988*; *Brecht et al., 2004*; *Walton et al., 2005*; *Katnani et al., 2012*). However, we think that these observations are rendered plausible with newer ideas (*Jagadisan and Gandhi, 2019*) on temporal alignment of population activity in the SC at the time of saccade triggering, as we explain in Discussion.

Finally, and for completeness, we repeated the same analyses of *Figure 4*, but this time for the neurons collected during experiment 2. The results are shown in *Figure 4—figure supplement 4*, and they are all consistent with the results that we obtained from *Figure 4*. Therefore, there was a clear and lawful relationship between the number of 'injected' visual spikes injected into the SC map by each active neuron and the executed movement amplitude.

## Tight temporal alignment between 'visual' spikes and movement onsets is needed for the spikes to alter eye movements

To further investigate the results of *Figure 4* and its related figure supplements, we next explored more detailed temporal interactions between SC visual bursts and saccade metric changes. Across all trials from all neurons analyzed in *Figure 4* (i.e. ≤4.5 deg eccentricity and in experiment 1), we measured the time of any given trial's visual burst peak relative to either microsaccade onset (*Figure 5A*), microsaccade peak velocity (*Figure 5B*), or microsaccade end (*Figure 5C*), and we sorted the trials based on burst peak time relative to microsaccade onset (i.e. the trial sorting in all panels in *Figure 5* was always based on the data from panel A). We then plotted individual trial

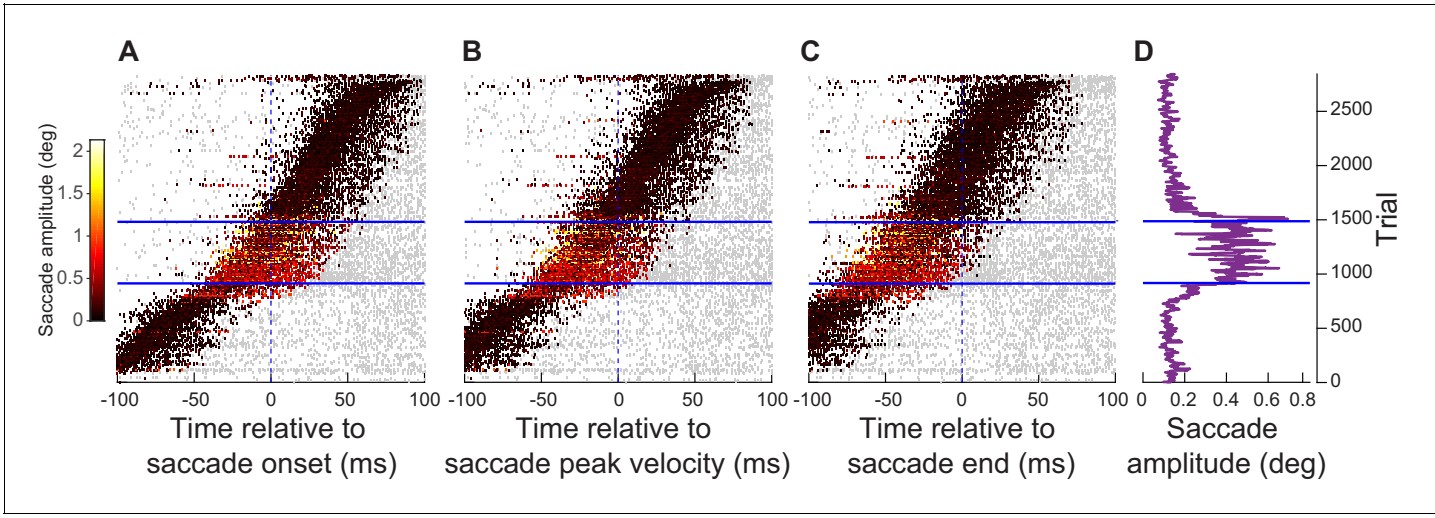

**Figure 5.** Exogenous, movement-unrelated SC spikes had the greatest impact on movement metrics when they occurred peri-saccadically. (A) Individual trial spike rasters across all neurons ≤ 4.5 deg in eccentricity and all movements towards RF locations from experiment 1. The spike rasters are sorted based on the time of the visual burst (peak firing rate after stimulus onset) relative to saccade onset (bottom left: trials with visual bursts earlier than microsaccades; top right: trials with visual bursts later than microsaccades). The spike rasters are plotted in gray except during the interval 30–100 ms after stimulus onset (our visual burst interval; *Figure 2*) to highlight the relative timing of the visual burst to movement onset. Spikes in the visual burst interval are color-coded according to the observed movement amplitude on a given trial (legend on the left). As can be seen, microsaccades were enlarged when extra-foveal SC spiking (stimulus-driven visual bursts) occurred right before and during the microsaccades (see marginal plot of movement amplitudes in D). (B) Same as A, and with the same trial sorting, but with burst timing now aligned to movement peak velocity. (C) Same as A, B, and with the same trial sorting, but with burst timing now aligned to movement end. The biggest amplitude effects occurred when the exogenous 'visual' spikes occurred pre- and intra-saccadically, but not post-saccadically. (D) Microsaccade amplitudes (20-trial moving average) on all sorted trials in A–C. Blue horizontal lines denote the range of trials for which there was a significant increase in movement amplitudes (Materials and methods). Note that the numbers of trials are evident in figure. *Figure 5—figure supplement 1* shows similar results from experiment 2, and *Figure 5—figure supplement 2* shows similar results from the far neurons of experiment 1.

The online version of this article includes the following source data and figure supplement(s) for figure 5:

**Source data 1.** Excel table with the source data for this figure.
**Figure supplement 1.** Analyses similar to those in *Figure 5* but from experiment 2.
**Figure supplement 1—source data 1.** Excel table with the source data for this figure.
**Figure supplement 2.** Analyses similar to *Figure 5* but for the far neurons of experiment 1.
**Figure supplement 2—source data 1.** Excel table with the source data for this figure.

spike rasters with the bottom set of rasters representing trials with the SC 'visual' burst happening much earlier than microsaccade onset and the top set being trials with the SC 'visual' burst occurring after microsaccade end. The rasters were plotted in gray in *Figure 5*, except that during a putative visual burst interval (30–100 ms from stimulus onset), we color-coded the rasters by the microsaccade amplitude observed in the same trials (same color coding scheme as in *Figure 4A*; note that if there was no spike in the measurement interval for a given trial, then there was no coloring made in the figure). The marginal plot in *Figure 5D* shows microsaccade amplitudes for the sorted trials (Materials and methods). We used this marginal plot as a basis for estimating which sorted trials were associated with the beginning of microsaccade amplitude increases (from the bottom of the raster and moving upward) and which trials were associated with the end of the microsaccade amplitude increases (horizontal blue lines; Materials and methods). As can be seen, whenever SC 'visual' bursts occurred pre- and intra-saccadically, microsaccade amplitudes were dramatically increased by two- to three-fold relative to baseline microsaccade amplitudes (blue horizontal lines). For visual bursts after peak velocity (*Figure 5C*), the effect was diminished, consistent with efferent delays from SC activity to extraocular muscle activation (*Miyashita and Hikosaka, 1996*; *Munoz et al., 1996*; *Stanford et al., 1996*; *Gandhi and Keller, 1999b*; *Katnani and Gandhi, 2012*; *Jagadisan and Gandhi, 2017*; *Smalianchuk et al., 2018*). The same results were obtained when we repeated the same analyses for the neurons collected during experiment 2 (*Figure 5—figure supplement 1*).

Importantly, in our earlier analyses (*Figure 3—figure supplement 1*; *Figure 4—figure supplements 2* and *3*), we had observed that the effects with more eccentric visual bursts on microsaccade amplitudes were still present, albeit with a significantly weaker magnitude. This would suggest that the same temporal relationship between the injected spikes and the movement amplitudes should still exist for the far neurons. This was indeed the case, as demonstrate in *Figure 5—figure supplement 2*. In this figure, we repeated the same analyses of *Figure 5* but only for neurons with eccentricities > 4.5 deg. We still found that it was the pre- and intra-saccadic injected 'visual' spikes that were associated with the increased microsaccade amplitudes. The tight temporal relationship did not depend on the eccentricity of the injected 'visual' spikes.

Therefore, our results so far suggest that at the time at which SC activity is to be read out by downstream neurons to implement a saccadic eye movement (right before movement onset to right before movement end, e.g. *Miyashita and Hikosaka, 1996*; *Munoz et al., 1996*; *Stanford et al., 1996*; *Gandhi and Keller, 1999b*; *Katnani and Gandhi, 2012*; *Jagadisan and Gandhi, 2017*; *Smalianchuk et al., 2018*), additional movement-unrelated SC spiking activity is also read out and has a direct impact on eye movement metrics.

Having said that, one problem with the analysis of *Figure 5* is that our 'visual burst interval' was still arbitrarily defined as a period 30–100 ms after stimulus onset (*Figure 2*). In reality, spiking activity could vary with different stimulus parameters like stimulus contrast or spatial frequency (e.g. *Figure 3* and its associated figure supplements). Therefore, to obtain even more precise knowledge of the time needed for any injected 'visual' spikes to start influencing microsaccade metrics, we next selected all individual trial spike rasters from *Figure 5A* (i.e. experiment 1), and we counted the number of spikes occurring within any given 5 ms time bin relative to eye movement onset. We did this for all time bins between −100 ms and +100 ms from movement onset, and we also binned the movements by their amplitude ranges (*Figure 6A*). The two smallest microsaccade amplitude bins reflected baseline movement amplitudes (see *Figure 3A*), and they expectedly occurred when there was no 'extra' spiking activity in the SC around their onset (*Figure 6A*, two darkest reds). For all other amplitude bins, the larger movements were always associated with the presence of extra 'visual' spikes on the SC map (more eccentric than the normal microsaccade amplitudes) occurring between −30 ms and +30 ms from saccade onset (*Figure 6A*). Note how the timing of the effect was constant across amplitude bins, suggesting that it was the relative timing of extra 'visual' spikes and movement onset that mattered; the amplitude effect (that is, the different colored curves) simply reflected the total number of spikes that occurred during the critical time window of movement triggering. This is consistent with *Figure 4* and also with new ideas related to population temporal alignment in the SC (*Jagadisan and Gandhi, 2019*). Therefore, additional 'visual' spikes in the SC at a time consistent with saccade-related readout by downstream neurons essentially 'leak' into the saccade being generated.

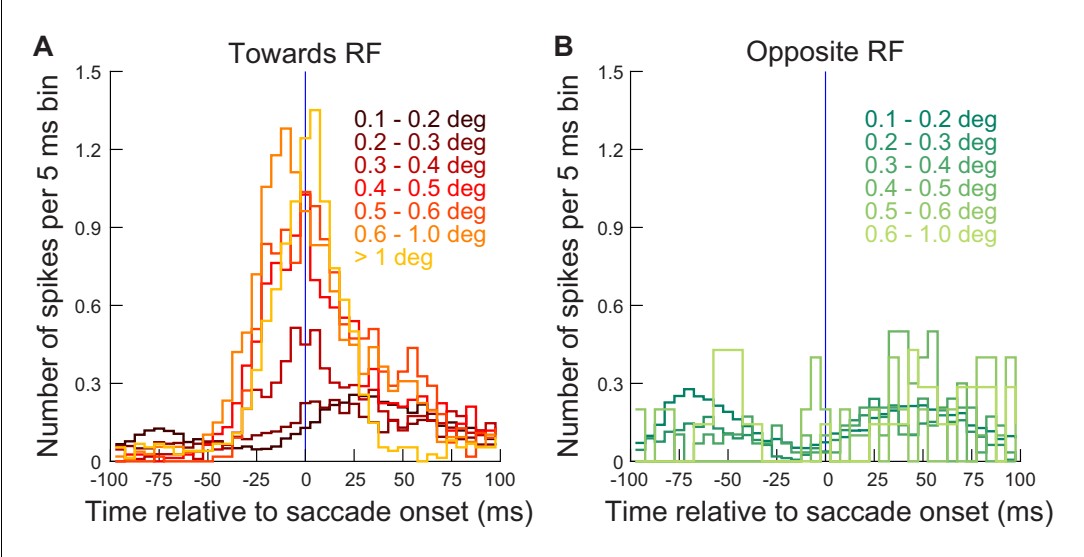

**Figure 6.** Exogenous, movement-unrelated 'visual' spikes affected movement metrics when they occurred within approximately ±30 ms from movement onset. (A) For the different microsaccade amplitude ranges from *Figure 5* (color-coded curves), we counted the number of exogenous spikes occurring from a recorded extra-foveal SC neuron (≤4.5 deg) within any given 5 ms time bin around movement onset (range of times tested: −100 ms to +100 ms from movement onset). The lowest two microsaccade amplitude ranges (0.1–0.2 and 0.2–0.3 deg) reflected baseline amplitudes during steady-state fixation (e.g. *Figure 3*), and they were not correlated with additional extra-foveal spiking activity around their onset (two darkest red curves). For all other larger microsaccades, they were clearly associated with precise timing of extra-foveal 'visual' spikes occurring within approximately ±30 ms from movement onset, regardless of movement size. The data shown are from experiment 1; similar observations were made from experiment 2 (*Figure 6—figure supplement 1*). The number of movements contributing to this figure is the same as in *Figure 5*. (B) Same as A but for movements opposite the recorded neuron's RF locations. There were fewer spikes during the peri-saccadic interval, suggesting that it was easier to trigger eye movements when there was no activity present in the opposite SC. *Figure 6—figure supplement 2* shows similar results from the far neurons (>4.5 deg eccentricity) of the same experiment (experiment 1).

The online version of this article includes the following source data and figure supplement(s) for figure 6:

**Source data 1.** Excel table with the source data for this figure.

**Figure supplement 1.** Same analysis as in *Figure 6* but for the neurons recorded during experiment 2 (≤4.5 deg).

**Figure supplement 1—source data 1.** Excel table with the source data for this figure.

**Figure supplement 2.** Same as *Figure 6* but for the far neurons from experiment 1.

**Figure supplement 2—source data 1.** Excel table with the source data for this figure.

On the other hand, the pattern of *Figure 6A* was not present for movements going opposite to the recorded neuron's RF's, for which, if anything, there was a lower number of spikes happening during the peri-saccadic interval (*Figure 6B*). This suggests that it was easier to trigger microsaccades in one direction when no activity was present in the opposite SC. Moreover, all these effects were directly replicated with the spatial frequency task as well (experiment 2; *Figure 6—figure supplement 1*).

Once again, when we repeated the analyses of *Figure 6* but now for only the far neurons from the same experiment, we got a qualitatively and quantitatively similar result (*Figure 6—figure supplement 2*). This is an important finding because it suggests that once a movement-unrelated spike is properly temporally aligned with the motor burst, it can still have a similar behavioral impact on movement amplitudes whether the spike was far or near in eccentricity. This suggests that the global reduction of behavioral effects that we saw with more eccentric stimuli (e.g. *Figure 3—figure supplement 1*) could reflect a reduced likelihood of temporal alignment between the 'visual' spikes and the population of neurons bursting as part of the movement command. As stated above, this idea of temporal alignment is consistent with recent novel hypotheses about the role of SC population temporal alignment in enabling the triggering of saccades (*Jagadisan and Gandhi, 2019*; see Discussion).

## Any peri-saccadic 'visual' spikes, even outside of 'visual' bursts, influence ongoing eye movements

The strongest evidence that any 'extra' spiking activity present on the SC map can systematically alter the amplitude of the eye movements, irrespective of our experimental manipulation of visual bursts, can be seen from the analyses of *Figure 7*. Here, we exploited an important property of experiment 2: the presented stimulus remained on the display inside a neuron's RF for a substantial period of time of up to 1300 ms (sometimes up to 3000 ms) (*Khademi et al., 2020*). This meant that after the initial visual bursts had subsided, SC neurons maintained a lower level of 'sustained' discharge for a prolonged period of time, a discharge that was often absent in the absence of stimuli since some SC neurons do not exhibit any baseline discharge. This meant that we could now ask whether SC activity long after the visual bursts was still read out at the time of movement triggering (i.e. whether the previous results in *Figures 2–6* were contingent on 'bursting' activity in the SC, or whether any spiking could still matter).

Consider, for example, the neuron in *Figure 7A*, which showed robust sustained activity for its preferred spatial frequency. We selected all microsaccades occurring >550 ms after stimulus onset in this neuron. We then asked whether we could replicate results similar to those in *Figure 4B*, but only for these movements occurring outside of the early 'visual' bursts. In *Figure 7B*, we plotted results for movements towards the RF location (this time, combining the lowest two spatial frequencies to increase our data availability, especially because sustained discharge is significantly lower in firing rate than burst discharge). And, in *Figure 7C*, we plotted movements opposite the RF location. The 'towards movements' were increased in amplitude with every injected extra spike from the 'sustained' discharge of the recorded neuron (F-statistic vs. constant model: F = 383, p=0.0325; estimated coefficients: intercept = 0.1105, t = 17.67, p=0.0360; slope = 0.0948, t = 19.57, p=0.0324), whereas opposite movements were not (F-statistic vs. constant model: F = 6.15, p=0.2440; estimated coefficients: intercept = 0.1348, t = 24.49, p=0.0250; slope = −0.0101, t = −2.47, p=0.2440). Another example neuron's results are shown in *Figure 7—figure supplement 1*, and both neurons were consistent with each other. Therefore, there was actually no need for a stimulus-driven visual burst to be present in the SC for us to observe effects of extraneous spiking activity on triggered eye movements. Even when the spikes were no longer strongly associated with the stimulus-induced visual burst (i.e. with stimulus onset), their presence on the SC map at a site more eccentric than microsaccade amplitudes was enough to modulate eye movement amplitudes in a systematic manner, increasing the amplitude above baseline levels when more spikes were present.

Across all neurons collected from experiment 2, in which we had the opportunity to look for spiking outside of the 'burst' intervals due to the longer trial durations, we found robust effects of individual neuronal spiking and microsaccade amplitudes (*Figure 7D,E*). These results were also statistically validated. For movements towards the RF, every additional 'sustained' spike linearly increased microsaccade amplitude with a slope of 0.0126 deg/spike (F-statistic vs. constant model: F = 13.1, p=0.0364; estimated coefficients: intercept = 0.1098, t = 12.87, p=0.0010; slope = 0.0126, t = 3.61, p=0.0363). Such modulation was again not visible for movements going in the opposite direction from the recorded neuron's RF's (*Figure 7E*), again suggesting that there is a lower limit to how small microsaccades can become with opposite drive from the other SC (F-statistic vs. constant model: F = 5, p=0.1110; estimated coefficients: intercept = 0.1169, t = 13.31, p=0.0009; slope = −0.0080, t = −2.23, p=0.1112).

Of course, quantitatively, the impact of each spike in *Figure 7D* (for 'towards' movements) was smaller in magnitude than the impact of each spike in *Figure 4B* (for similar 'towards' movements). In other words, a single spike during the 'sustained' discharge caused a smaller microsaccade amplitude increase than a single spike during 'burst' discharge. However, this is fully expected: during the visual bursts, a large population of SC neurons are expected to be bursting simultaneously (*Lee et al., 1988*); on the other hand, during 'sustained' discharge, different individual neurons may or may not be simultaneously active depending on a variety of factors related to their individual spatio-temporal RF properties (*Churan et al., 2012*). Thus, a smaller population of simultaneously spiking neurons is expected. In that regard, the results of *Figure 7D,E* provide the most compelling evidence in our experiments so far that every additional SC spike that is available at movement triggering can alter movement metrics.

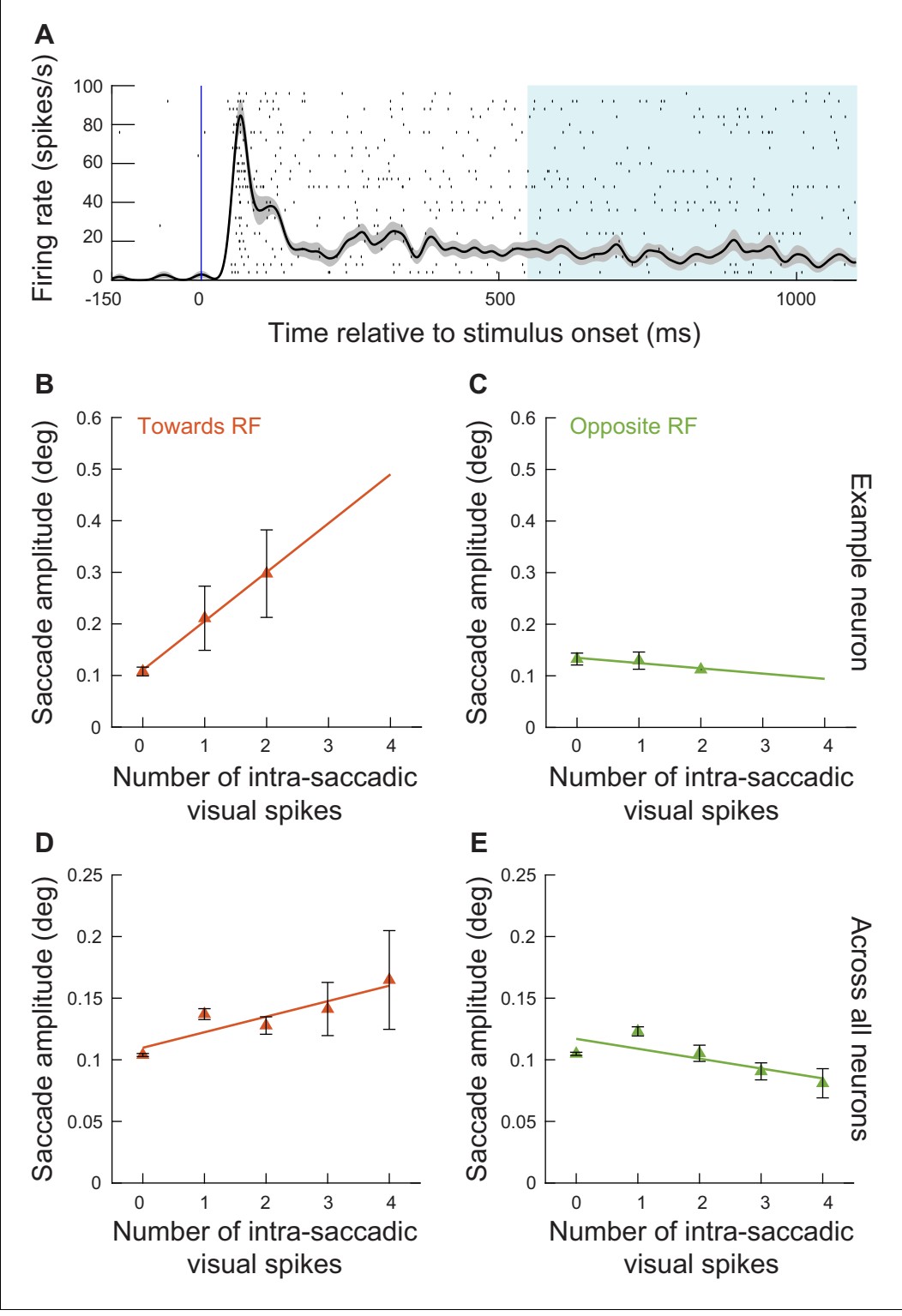

**Figure 7.** Exogenous, movement-unrelated spikes influenced eye movement metrics even when they did not occur within stimulus-driven 'visual' bursts. (**A**) In experiment 2, we had a prolonged period of fixation after stimulus onset. This meant that there was low-level discharge present in the SC long after the end of the initial 'visual' burst, as shown in this example neuron (spike raster and average firing rate across trials with the preferred spatial frequency in the RF; error bars denote s.e.m.). This allowed us to select all microsaccades occurring >550 ms after stimulus onset, and to ask whether movement-unrelated SC spiking activity that was coincident with these

*Figure 7 continued on next page*

*Figure 7 continued*

movements still influenced their metrics. (B) For the example neuron in A and for microsaccades > 550 ms after stimulus onset and towards the RF, we performed an analysis like that of *Figure 4B*. There was a positive correlation between the number of intra-saccadic spikes and movement amplitude. Note that we combined trials with the lowest two spatial frequencies to increase the numbers of observations in this analysis. The numbers of movements contributing to each x-axis point are 62, 10, and 4 microsaccades for 0, 1, and 2 spikes, respectively. (C) Same as B but for movements opposite the RF from the same session. The numbers of movements contributing to each x-axis point are 77, 20, and 1 microsaccades for 0, 1, and 2 spikes, respectively. (D) Relationship between the number of intra-saccadic SC 'sustained' spikes (i.e. not part of 'visual' bursts) and microsaccade amplitudes for eye movements triggered >550 ms after grating onset from all sessions of experiment 2. We included trials from all spatial frequencies. For each microsaccade towards the RF location, we counted how many 'sustained' spikes were emitted by a given recorded neuron in the interval 0–20 ms after microsaccade onset. We then plotted microsaccade amplitude as a function of intra-saccadic 'sustained' spikes. Even when the spikes occurred outside of 'visual' bursts, they still had an influence on movement metrics. The numbers of movements contributing to each x-axis point are 4009, 747, 226, 62, and 26 microsaccades for 0, 1, 2, 3, and 4 spikes, respectively. (E) Same as D but for movements opposite the RF location. There was no increase in microsaccade amplitude. The numbers of movements contributing to each x-axis point are 4114, 721, 157, 45, and 16 microsaccades for 0, 1, 2, 3, and 4 spikes, respectively. Error bars in B–D denote 95% confidence intervals.

The online version of this article includes the following source data and figure supplement(s) for figure 7:

**Source data 1.** Excel table with the source data for this figure.
**Figure supplement 1.** A second example neuron from experiment 2.
**Figure supplement 1—source data 1.** Excel table with the source data for this figure.

---

To summarize the overall results so far, we found that there is a tight time window around saccade onset (*Figures 5–7*) in which any movement-unrelated spikes in sites other than the saccade goal representation can induce a systematic variation in the motor program.

## Saccade-related movement bursts occur simultaneously with stimulus-driven 'visual' bursts at separate SC sites

Our results demonstrate that as little as one single extra action potential by each visually activated neuron was sufficient, within a specific time window, to alter ongoing microsaccades (*Figures 4–7*). However, it still remains unclear whether the movement bursts for these microsaccades did indeed occur in the SC or not. In other words, in all of the above experiments, our primary hypothesis was that the visual bursts 'added' to movement-related bursts elsewhere on the SC map (in our case, in the rostral SC) in order to alter the movement metrics. We believe that this is a reasonable hypothesis. However, past work might predict otherwise: that visual bursts in the caudal SC, representing the eccentric stimulus locations (e.g. *Figures 1* and *2*), should actually reduce activity in other distant SC sites associated with the movement plans (*Dorris et al., 2007*). From the perspective of microsaccades, this alternative mechanism would mean a reduction of rostral SC activity rather than an increase, since microsaccade-related discharge occurs in the rostral SC (*Hafed et al., 2009*; *Hafed and Krauzlis, 2012*; *Willeke et al., 2019*). Indeed, in the absence of any microsaccades, a peripheral stimulus onset is known to be associated with both a visual burst in the caudal SC as well as a reduction in firing activity in the rostral SC (*Munoz and Istvan, 1998*; *Hafed and Krauzlis, 2008*). Moreover, slice work in rodents suggests the existence of potential lateral inhibition mechanisms in at least some SC layers, consistent with this prior evidence (*Isa and Hall, 2009*; *Kasai and Isa, 2016*). Might it then be the case that our hypothesis of 'added' spikes to the readout is invalid, and that rostral SC activity actually did not burst for our microsaccades?

To directly test this, in experiment 3, we conducted additional recordings using multielectrode arrays inserted into either the rostral SC (representing microsaccade amplitude ranges), the caudal SC (representing eccentric locations associated with 'visual' bursts), or both simultaneously (*Figure 8*). In this case, we presented a white disc of radius 0.5 deg peripherally during maintained fixation (Materials and methods). During rostral SC recordings, we placed the peripheral stimulus at 10 deg eccentricity either to the right or left of fixation across the different trials (i.e. very far in eccentricity from the movement endpoints). During caudal SC and simultaneous rostral and caudal SC

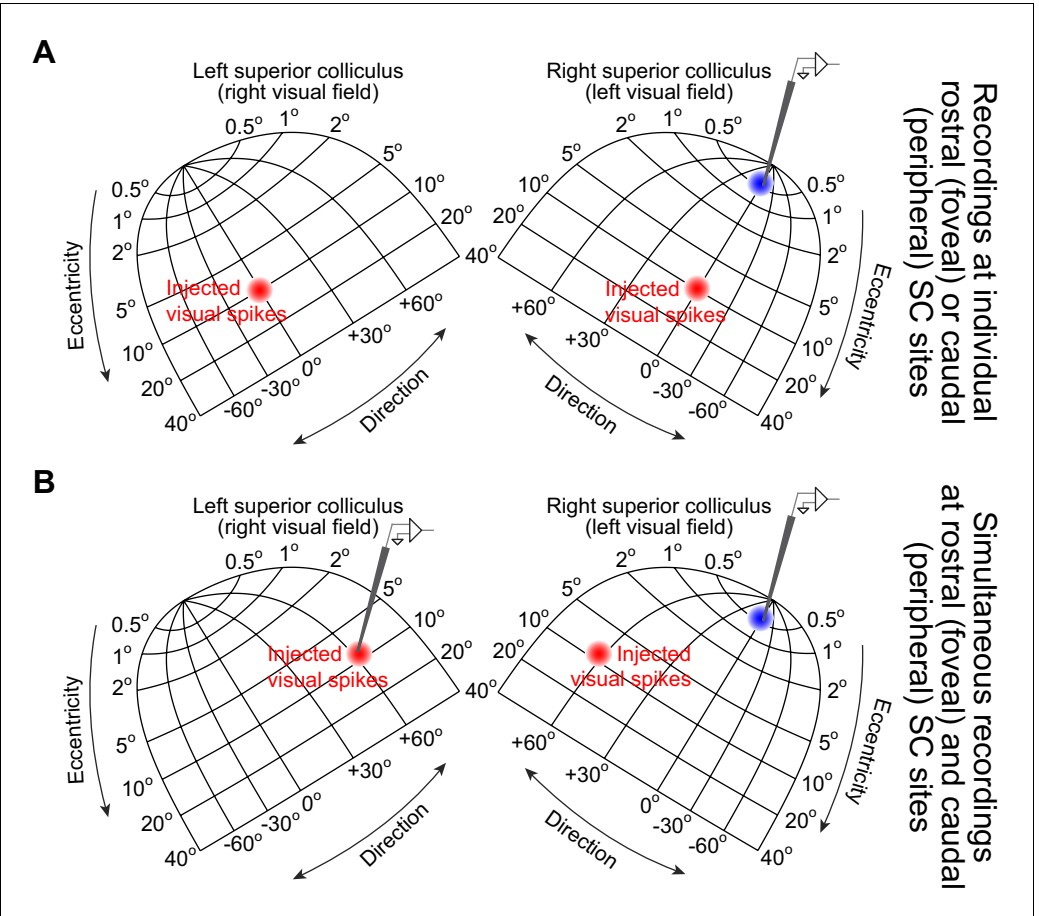

**Figure 8.** Exploring both movement-related and stimulus-driven SC discharge at the time of microsaccade triggering. (**A**) We inserted microelectrode arrays into either the rostral SC (example shown in the right rostral SC) or the caudal SC. We then ran a behavioral fixation task in which the monkey fixated and a peripheral stimulus appeared on either side of fixation (Materials and methods). This meant that we injected 'visual' bursts in either the right or left caudal SC across trials (red), allowing us to measure either rostral SC or caudal SC activity when the injected 'visual' spikes occurred coincidentally with triggered microsaccades (same logic as in *Figure 1*). The caudal SC recordings were meant to support the earlier figures by demonstrating that intra-saccadic visual bursts could still occur in the SC; the rostral SC recordings were meant to investigate what happens to movement-related bursts at the time of the peripheral visual bursts. (**B**) In yet another set of experiments, and using the same behavioral task, we inserted two sets of microelectrode arrays simultaneously into both the rostral and caudal SC together. This allowed us to confirm the results from **A** using simultaneous rostral and caudal recordings. The shown topographic map of the SC is based on our earlier dense mappings, demonstrating both foveal (*Chen et al., 2019*) and upper visual field (*Hafed and Chen, 2016*) tissue area magnification.

recordings, we placed the peripheral stimulus either at the visual field location represented by the caudal SC site (i.e. inside the visual RF's) or at the diametrically opposite location.

We confirmed that movement-related discharge still occurred simultaneously with peripheral 'visual' bursts in the SC. For example, *Figure 9* shows two example neurons recorded from the rostral SC. In *Figure 9A*, we show the movement-related RF of the neuron, which was recorded from the right SC. The neuron preferred primarily horizontal leftward microsaccades. For these microsaccades, the neuron exhibited expected peri-microsaccadic elevations in activity (*Hafed et al., 2009*; *Hafed and Krauzlis, 2012*; *Willeke et al., 2019*) in a standard RF mapping task (Methods; *Figure 9B*). We then asked what happened to this neuron's activity in the main task of experiment three when peripheral stimuli were presented in the absence of any microsaccades during a peri-stimulus interval (−50–200 ms). Consistent with prior observations (*Munoz and Istvan, 1998*; *Dorris et al., 2007*; *Hafed and Krauzlis, 2008*) the neuron indeed decreased its activity (*Figure 9C*).

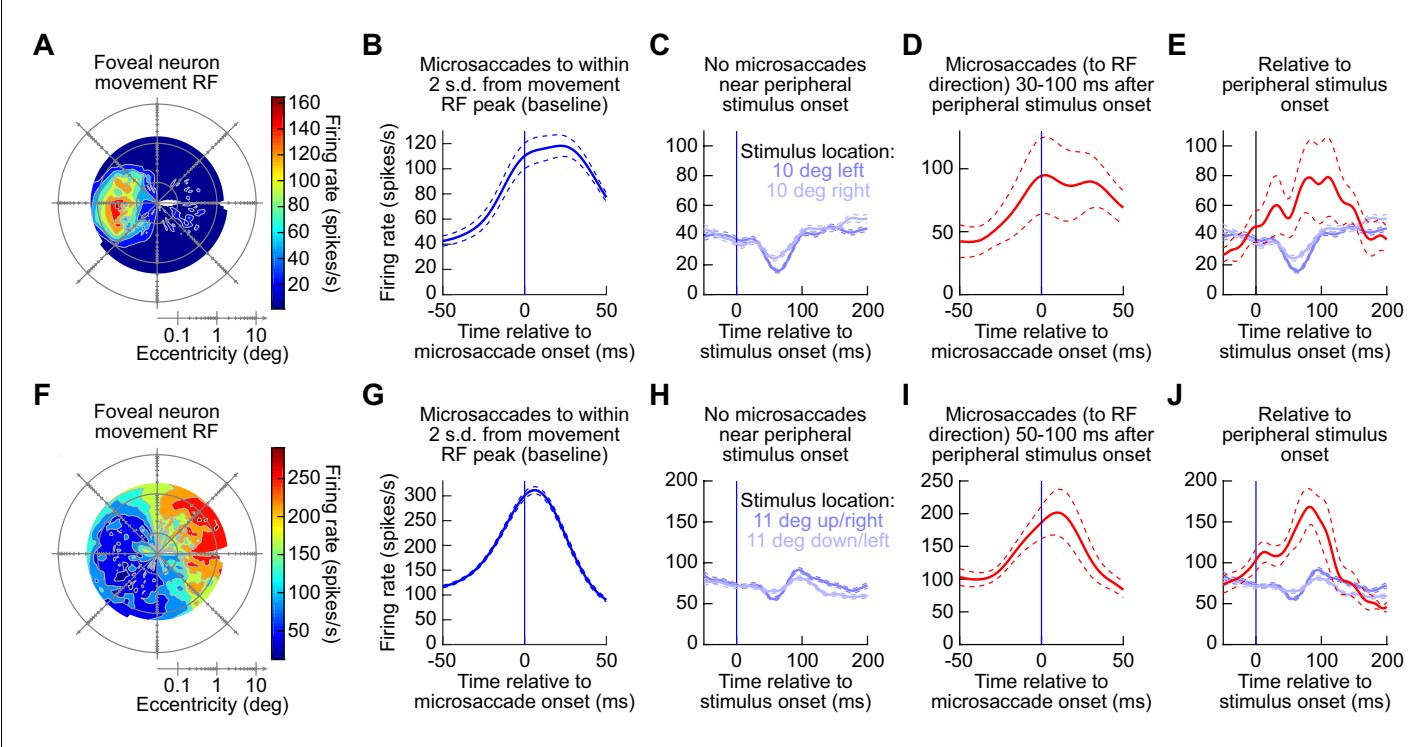

**Figure 9.** Rostral SC activity still exhibited bursts for microsaccades at the same time as caudal SC visual bursts. (**A**) Example movement-related RF of a rostral SC neuron in the right SC (obtained from an RF mapping task; Materials and methods). Peak peri-microsaccadic firing rate is shown as a function of microsaccade radial amplitude and direction. Movement dimensions are plotted on log-polar axes (*Hafed and Krauzlis, 2012*), and the origin represents 0.03 deg radial amplitude. The neuron preferred leftward horizontal microsaccades. (**B**) After obtaining a movement RF like in **A**, we fitted the RF with a two-dimensional gaussian function (Materials and methods), and we then selected all microsaccades to a region within 2 s.d. of the fitted gaussian's peak. We then plotted firing rates as a function of time from microsaccade onset, confirming movement-related discharge (*Hafed et al., 2009*; *Willeke et al., 2019*). (**C**) When a peripheral stimulus appeared in the main task of experiment three and no microsaccades occurred within −50–200 ms from stimulus onset, the neuron reduced its activity, consistent with earlier reports (*Munoz and Istvan, 1998*; *Dorris et al., 2007*; *Hafed and Krauzlis, 2008*). (**D**) However, the same neuron still exhibited a movement-related burst if the microsaccades towards its movement RF occurred within the visual burst interval associated with stimulus onset. (**E**) Thus, when aligned to peripheral stimulus onset, the neuron could either reduce its activity if microsaccades did not occur (blue curves), or it could increase its activity if microsaccades to the movement RF occurred (red). (**F–J**) Similar observations for a second example rostral SC neuron, this time in the left rostral SC. Note that for this particular example neuron, the visual burst interval that we picked was slightly modified because of a rarity of microsaccades of appropriate direction, but the same conclusions were reached as in the first example neuron (also see *Figure 11*). Error bars denote s.e.m.

The online version of this article includes the following source data for figure 9:

**Source data 1.** Excel table with the source data for this figure.

However, and most critically, on the rare occasions in which microsaccades towards the movement RF occurred 30–100 ms after stimulus onset (i.e. coincident with peripheral visual bursts; *Figures 2– 6*), the neuron actually still burst and did not decrease its activity (*Figure 9D*). This means that when we aligned these same trials' activity profiles to stimulus onset rather than to microsaccade onset (*Figure 9E*), we found that the neuron actually increased, rather than decreased, its activity at the same time as the presumptive caudal SC visual burst. In other words, there were two 'bursts' in the SC: one in the rostral SC and one in the caudal SC. Naturally, because microsaccades happened at variable times relative to stimulus onset in *Figure 9E* (also see *Figure 2A*), the activity increase was temporally smeared (giving rise to what appeared like transient modulations) when aligned to stimulus, rather than to microsaccade, onset.

Almost identical observations were made for a second example rostral SC neuron, now from the left SC (*Figure 9F–J*). Therefore, at the time of peripheral visual bursts, it is still possible to observe movement-related bursts in another location in the SC map. This is consistent with the idea that local

excitation dominates lateral connectivity patterns in the deeper motor-related layers of the SC (*Phongphanphanee et al., 2014*), which seems particularly useful (at the expense of long-range inhibition) for rapid burst generation.

To even further support the above conclusion, we recorded from both the caudal and rostral SC simultaneously in some sessions (in addition to other sessions in which we only recorded the caudal SC, in order to confirm our earlier observations in *Figure 2* that peripheral visual bursts could still occur simultaneously with triggered microsaccades). *Figure 10* shows an example pair of neurons that were recorded simultaneously from the same task of *Figures 8* and *9*. The caudal neuron is shown in the top row of the figure (*Figure 10A–C*), and the rostral neuron is shown in the bottom

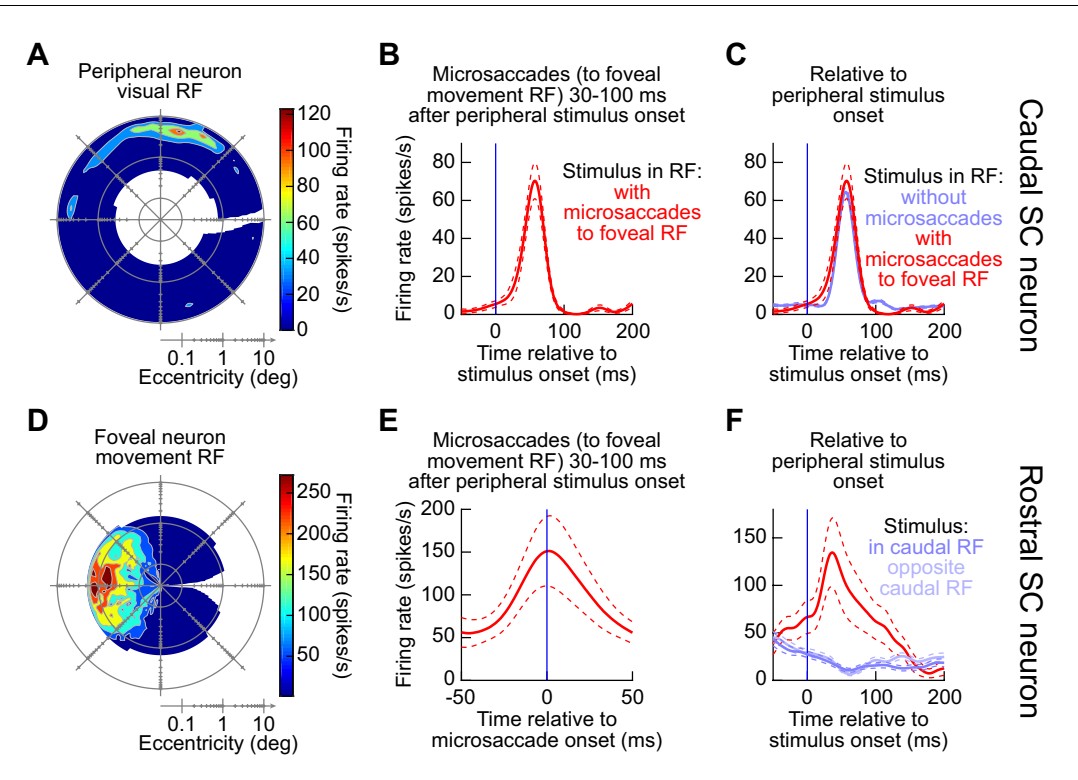

**Figure 10.** Simultaneous rostral and caudal. SC recordings confirmed the simultaneity of peripheral visual bursts and foveal movement-related bursts when microsaccades were triggered around the time of peripheral visual bursts. (**A**) Visual RF of an example neuron from an experiment with both caudal and rostral microelectrode arrays inserted into the SC. This example neuron was recorded from the caudal array inserted into the left SC. The RF mapping task revealed a preferred eccentricity of ~6 deg. (**B**) The neuron still exhibited a robust visual response for a stimulus appearing inside its visual RF (at an eccentricity of ~6 deg) even when there were simultaneous microsaccades (towards the movement RF) occurring 30–100 ms after stimulus onset (i.e. coincident with the time of the visual burst). (**C**) For the same neuron, the visual burst was similar with and without microsaccades occurring within the visual burst interval. (**D**) A foveal movement-related RF of a simultaneously recorded neuron, this time from the second microelectrode array inserted into the right rostral SC. The shown map was obtained from the RF mapping task (Methods). (**E**) In the main task of experiment 3, for microsaccades towards the movement RF occurring within the visual burst interval (i.e. coincident with the visual burst in **B**), the neuron still exhibited a robust microsaccade-related discharge. (**F**) This means that relative to peripheral stimulus onset, this neuron actually had a burst (red) rather than a decrease (blue) in firing rate at the same time as the peripheral visual burst in the caudal SC (**C**). The blue firing rate curves show the same neuron's response when the peripheral stimulus onset occurred in the absence of any microsaccades (same conventions as in *Figure 9*). Therefore, microsaccades occurring at the time of peripheral visual bursts were associated with readout of two burst loci: one in the rostral SC associated with the triggered movement and one in the caudal SC associated with visual stimulus onset. Error bars denote s.e.m.

The online version of this article includes the following source data for figure 10:

**Source data 1.** Excel table with the source data for this figure.

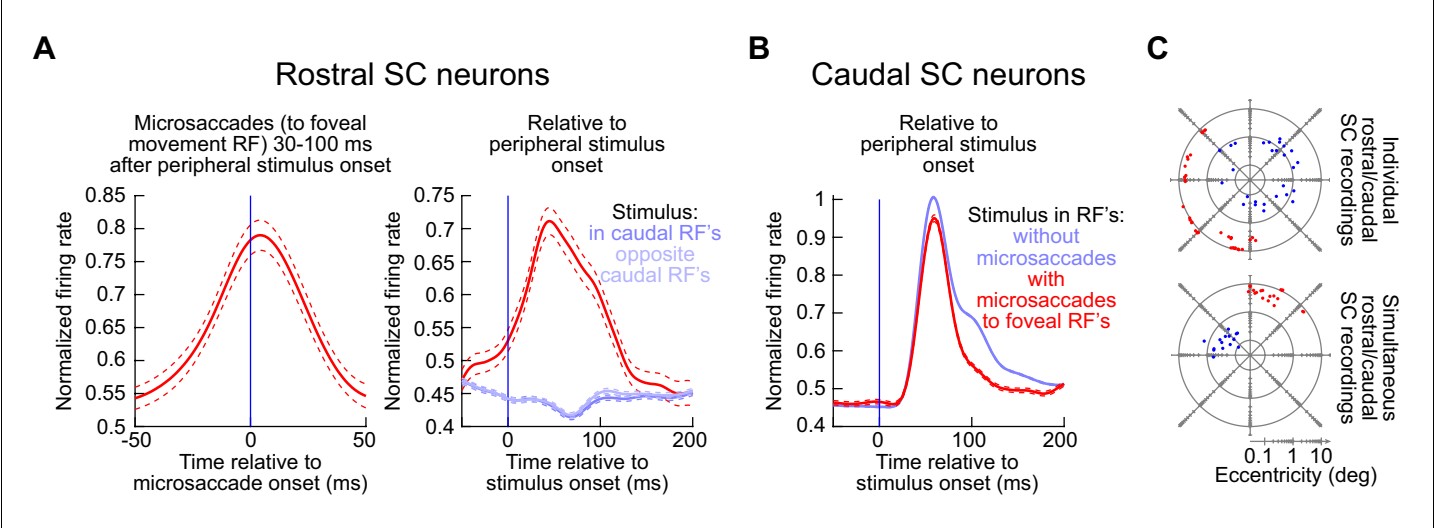

**Figure 11.** Population summary of the experiments in *Figures 8–10*. (A) Left panel: movement-related firing rate for all rostral SC neurons when microsaccades towards the movement RF occurred 30–100 ms after peripheral stimulus onset (i.e. coincident with peripheral visual burst occurrence). For each neuron, we first calculated the microsaccade-related discharge for the preferred microsaccades in the RF mapping task and then divided by this maximum firing rate to normalize the activity of individual trials from the main task. We then averaged across all average normalized firing rates of individual neurons to obtain a population response (error bars denote s.e.m.). Right panel: When the same data were aligned to stimulus onset (as opposed to microsaccade onset), we could see that the rostral SC clearly exhibited bursts at the same time as peripheral visual bursts when microsaccades occurred (red). *Figure 12* shows paired measurements of raw firing rates of all rostral neurons at the time of microsaccade onset, with or without peripheral visual bursts; it confirms that the rostral SC movement-related bursts were similar whether there was a peripheral visual burst or not. When microsaccades did not occur, peripheral stimulus onsets in either direction suppressed rostral SC activity (blue curves). (B) For all caudal SC neurons, we first averaged the firing rate across trials after a stimulus appeared inside the RF (again from the RF mapping task). We then normalized all trial firing rates by this measurement, and we then pooled neurons by averaging their individual normalized firing rate curves in the main task of experiment 3 (to obtain a population average response; error bars denote s.e.m.). Consistent with all of our earlier results, peripheral visual bursts still occurred even when coincident microsaccades occurred (red). Note that the red curve was slightly suppressed. This is because in most of our experiments (see C), the microsaccade site was opposite in direction from the caudal SC site. This is a condition known to be associated with suppressed visual bursts (*Chen et al., 2015*); also see *Figure 2—figure supplement 1*. (C) RF hotspot locations from all recording sites in these experiments (top: individual microelectrode array in either the caudal or rostral SC; bottom: simultaneous caudal and rostral SC recording arrays). The numbers of neurons are described in Materials and methods.

The online version of this article includes the following source data for figure 11:

**Source data 1.** Excel table with the source data for this figure.

row (*Figure 10D–F*). Based on the visual RF of the caudal neuron (*Figure 10A*), which was at an eccentricity of approximately 6 deg, we placed the stimulus inside this RF, and we measured the response when there were microsaccades being triggered towards the movement field of the rostral neuron (shown in *Figure 10D* from the RF mapping task). In other words, in *Figure 10B*, a peripheral stimulus appeared inside the visual RF of the caudal neuron, while leftward microsaccades were occurring simultaneously towards the movement RF of the rostral neuron. As can be seen from *Figure 10B*, the visual burst still occurred in the caudal neuron, consistent with *Figure 2*. At the simultaneously recorded rostral SC site, the rostral neuron also exhibited a peri-microsaccadic movement burst (*Figure 10E*). Therefore, there were two simultaneous SC bursts (*Figure 10B,E*). This observation is rendered clearer when we aligned the activity in *Figure 10E* to peripheral stimulus onset rather than to microsaccade onset (*Figure 10F*, red; again the firing rate was distorted by the variable microsaccade onset times). In this case, *Figure 10C* (red) showed a visual burst, and *Figure 10F* (red) showed a rostral movement burst, simultaneously. Incidentally, and consistent with *Figure 9* and prior reports (*Dorris et al., 2007*; *Hafed and Krauzlis, 2008*), in the absence of any microsaccades, the peripheral visual burst (*Figure 10C*, blue) was indeed accompanied by reduced activity in the rostral neuron (*Figure 10F*, blue), but this was only the case in the absence of microsaccades.

Across the population of rostral and caudal SC neurons recorded during this additional experiment, we observed consistent results with the above examples in *Figures 8–10*. Specifically, we

normalized each neuron's activity (either the microsaccade-related response for rostral neurons or the stimulus-induced visual response for caudal neurons) to its peak response in baseline (Materials and methods). We then averaged across neurons to obtain a population summary. For the rostral neurons, when microsaccades were triggered 30–100 ms after peripheral stimulus onset and they were towards the movement RF's of these neurons (Materials and methods), the neurons still exhibited classic microsaccade-related discharge (*Figure 11A*, left panel). Because the microsaccades happened right after peripheral stimulus onset, aligning this same discharge to the stimulus onset (*Figure 11A*, right panel, red) revealed a clear burst, which was absent (and replaced by a decrease in activity) when no microsaccades occurred near stimulus onset (*Figure 11A*, right panel, blue). For the peripheral neurons, stimulus onsets inside their RF's elicited robust visual bursts, both without microsaccades (*Figure 11B*, blue) and with microsaccades (*Figure 11B*, red). The visual burst with microsaccades was slightly suppressed (see *Figure 2—figure supplement 1*), but this was expected: most of our rostral sites were opposite the caudal sites (*Figure 11C*). Therefore, microsaccades towards the movement RF's of rostral neurons were opposite the direction of the peripheral stimulus, a condition that suppresses visual bursts (*Chen et al., 2015*).

Importantly, for each recorded rostral SC neuron, we also performed a paired comparison between the neuron's unnormalized raw microsaccade-related movement bursts with and without peripheral visual bursts. Specifically, we picked microsaccades directed towards each neuron's movement RF hotspot location (Methods), and we did this both when the movements occurred in a baseline pre-stimulus interval (500–1500 ms before peripheral stimulus onset in the main task) or in the interval 30–100 ms after peripheral stimulus onset. We then measured average firing rate within ±15 ms from microsaccade onset. Across the population, there was no significant alteration of the rostral motor bursts by the presence of a peripheral visual burst (*Figure 12A*; p=0.12539, paired ranksum test, N = 42 neurons). The example rostral SC neurons from *Figures 9* and *10* are all explicitly highlighted in the summary plot of *Figure 12A*.

In contrast, from the perspective of behavioral results in the same experiment, we still found that the stimulus-congruent microsaccades occurring simultaneously with the peripheral visual bursts were enlarged in size when compared to baseline microsaccades (*Figure 12B*), just like with all of our earlier analyses in experiments 1 and 2. This was the case in experiment three despite the fact that this experiment had peripheral stimuli at eccentricities > 5 deg.

Therefore, the results of *Figures 2–7* demonstrate that intra-saccadic visual bursts (and intra-saccadic visual discharge in general, even outside of visual bursts) lawfully 'add' to the metric computation of the executed microsaccades. Moreover, *Figures 8–12* confirm that such visual bursts (and visual discharge in general) are indeed 'additions' to the originally existing movement-related bursts being emitted by the SC for downstream readout, and that the movement-related bursts themselves are minimally affected.

## Discussion

We experimentally injected movement-unrelated spikes into the SC map at the time of saccade generation. We found that such spikes significantly altered the metrics of the generated saccadic eye movement, suggesting an instantaneous readout of the entire SC map for implementing any individual movement.

### The SC and behavioral variability

Our results reveal a component of motor variability that we believe has been previously unaccounted for, namely, the fact that ever-present spiking activity in the entire SC map (whether due to sustained firing rates for a stimulus presented in the RF, like in *Figure 7*, or otherwise) can 'leak' into the readout performed by downstream motor structures when executing a movement. In fact, our results from *Figure 7* showed that any intra-saccadic spikes on the SC map (far from the location of the motor burst) were sufficient to modulate microsaccade metrics, meaning that there was no need for a stimulus onset or even a stimulus-driven visual burst, like in *Figures 2–6*. Indeed, saccades during natural viewing show an immense amount of kinematic variability when compared to simplified laboratory tasks with only a single saccade target (*Berg et al., 2009*). In such natural viewing, natural images with plenty of low spatial frequency image power are expected to strongly activate a large

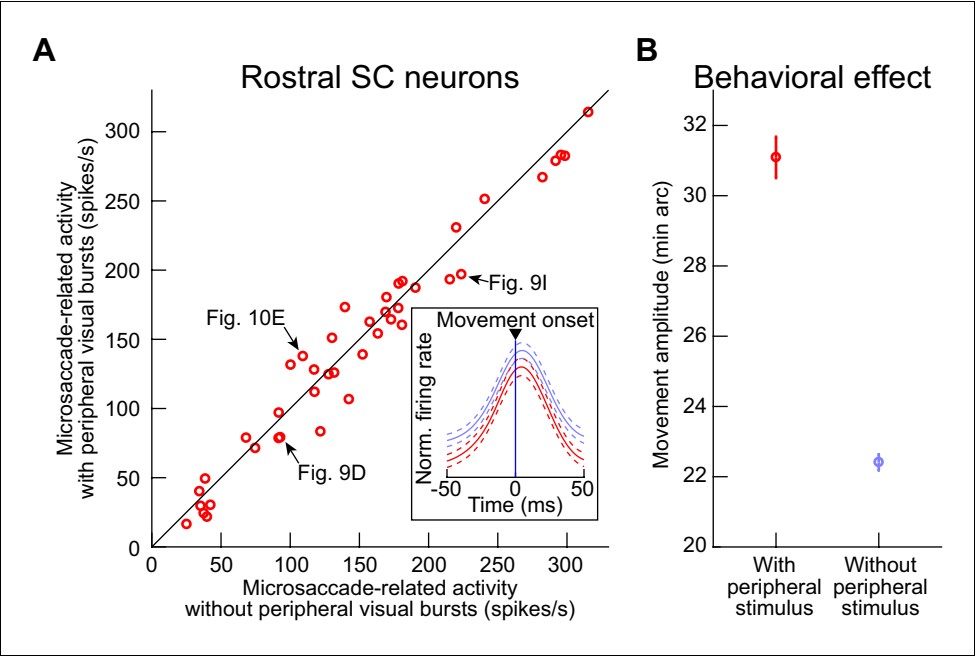

**Figure 12.** Similarity of microsaccade-related motor bursts at the time of peripheral visual bursts. (A) For each rostral SC neuron from *Figures 8–11*, we measured the average firing rate in the interval within ±15 ms from microsaccade onset. We did this for microsaccades directed towards the rostral RF hotspot (Materials and methods). We then plotted this firing rate for the movements with which there was no peripheral stimulus onset (baseline microsaccades 500–1500 ms before stimulus onset; x-axis; Materials and methods) and also when the microsaccades occurred 30–100 ms after peripheral stimulus onset (that is, coincident with the peripheral visual bursts; y-axis). There was no statistically significant difference between the two measurements (p=0.12539; paired ranksum test; N = 42 neurons). The example rostral SC neurons from *Figures 9* and *10* are highlighted with black arrows, and the inset shows the peri-movement population firing rates from all within-neuron paired measurements: red replicates the plot of *Figure 11A*, and blue shows the activity for baseline pre-stimulus microsaccades (error bars denote 95% confidence intervals). (B) For all of the sessions of experiment 3, we measured baseline microsaccade amplitudes and the amplitudes of microsaccades that were coincident with the peripheral visual bursts (occurring 30–100 ms after stimulus onset). In both cases, we picked microsaccades with directions towards the peripheral stimulus locations, because it is these microsaccades that are enlarged in size (e.g. *Figures 2* and *3*). Microsaccade amplitudes were significantly larger for the movements coincident with the peripheral visual bursts (p=7.9073×10$^{-27}$; t-test; N = 650 microsaccades with peripheral stimulus onset, and N = 1956 baseline microsaccades). This occurred even though the peripheral stimuli were more eccentric than 4.5 deg, consistent with our earlier results (e.g. *Figure 3—figure supplement 1* and *Figure 6—figure supplement 2*). Error bars denote s.e.m.

The online version of this article includes the following source data for figure 12:

**Source data 1.** Excel table with the source data for this figure.

number of SC neurons, which prefer low spatial frequencies, around the time of saccades (*Chen et al., 2018*; *Khademi et al., 2020*).

From an ecological perspective, our results demonstrate a remarkable flexibility of the oculomotor system during eye movement generation. Historically, saccades were thought to be controlled by an open-loop control system due to their apparent ballistic nature. However, other evidence, including our current results, clearly showed that individual saccades are actually malleable brain processes. In our case, we experimentally tried to generate a movement-unrelated 'visual' burst of activity that precisely coincided with the time of saccade triggering. We uncovered an instantaneous readout of the entire SC map that includes all the activity related to the ongoing motor program as well as the 'extra' activity. In real life, this extra activity might happen due to external sensory stimulation, such as the presence of a new object in the visual scene. In the laboratory, this extra activity can also be completely artificial, as is the case with electrical microstimulation (*Katnani and Gandhi, 2011*; *Katnani et al., 2012*); for example, dual-site suprathreshold simultaneous SC microstimulation

results in saccades to neither 'burst' location, consistent with the deviations that we observed for our microsaccades.

## Sensory signals in motor structures

The integration that we observed of sensory signals into the motor plan was not merely a 'loose' leakage phenomenon; rather, it exhibited a lawful additive process between the 'visual' spikes injected into the SC population and the altered microsaccade amplitudes (*Figures 4–7*). The more 'visual' spikes that occurred intra-saccadically, the larger the microsaccades became, following a linear relationship. Once again, this was even more remarkable for 'sustained' discharge, in which only few SC neurons might be expected to be active at the very same time (*Figure 7*). We discount the possibility that this effect was due to movement-related bursts per se, because we ensured that the neurons were not exhibiting movement bursts for the ranges of eye movements that we analyzed (*Figure 1—figure supplement 1*).

We suggest that this additive mechanism might underlie many of the effects commonly seen in experimental psychophysics, in which saccade kinematics are systematically altered by the presence of sudden irrelevant visual information available as close as 40 ms to movement onset (*Edelman and Xu, 2009*; *Buonocore and McIntosh, 2012*; *Guillaume, 2012*; *Buonocore et al., 2016*; *Buonocore et al., 2017*; *Malevich et al., 2020b*). Our hypothesis is that these modulations are a behavioral manifestation of the instantaneous readout of the activity on the SC map, as we also previously hypothesized (*Hafed and Ignashchenkova, 2013*; *Buonocore et al., 2017*).

Moreover, similar specification mechanisms can be seen in the instantaneous alteration of eye velocity during smooth pursuit when small flashes are presented (*Buonocore et al., 2019*), and even with ocular position drift during fixation (*Malevich et al., 2020a*). These observations extend the mechanisms uncovered in our study to the pursuit system and beyond, and they also relate to sequential activation of SC neurons during curved saccades associated with planning sequences of movements (*Port and Wurtz, 2003*). These observations are also consistent with experimental manipulations in which the oculomotor 'gate' in the brainstem is 'opened' by blinks (*Jagadisan and Gandhi, 2017*). In such manipulations, the authors exploited the fact that blinks are associated with pauses in brainstem omnipause neuron activity, and they revealed that blinks during saccade planning revealed that preparatory spikes in the SC before saccade onset contain a kind of 'movement potential' (*Jagadisan and Gandhi, 2017*). This is a clear analogous situation to our results.

Our investigations, which were driven by behavioral modulations observed in psychophysical experiments as alluded to above, therefore now provide a means to precisely quantify such behavioral modulations when sensory stimuli arrive in close temporal proximity to saccade generation. They also extend to early microstimulation experiments (*Glimcher and Sparks, 1993*), and to situations in which concurrent saccade motor plans can give rise to vector averaging (*Robinson, 1972*; *Schiller et al., 1979*; *Schiller and Sandell, 1983*; *Sparks and Mays, 1983*; *Edelman and Keller, 1998*; *Katnani and Gandhi, 2011*; *Katnani et al., 2012*) or curved (*McPeek et al., 2003*) saccades to ones in which a sensory burst itself is what is concurrently present with the saccade motor program. In that sense, the sensory burst may act as a motor program itself, as with express saccades having ultra-short latencies that appear to merge SC visual and motor bursts (*Edelman and Keller, 1996*; *Sparks et al., 2000*). In these saccades, it could be that the triggering for the saccades happens exactly at the time of the visual bursts, therefore 'pulling' the saccades to the locations of the bursts. This is not unlike our observations (*Figures 3–6*).

Most intriguingly, our results motivate similar neurophysiological studies on sensory-motor integration in other oculomotor structures. For example, our own ongoing experiments in the lower oculomotor brainstem, at the very final stage for saccade control (*Keller, 1974*; *Büttner-Ennever et al., 1988*; *Gandhi and Keller, 1999a*; *Missal and Keller, 2002*), are revealing highly thought provoking visual pattern analysis capabilities of intrinsically motor neurons (*Buonocore et al., 2020*). These and other experiments will, in the future, clarify the mechanisms behind multiplexing of visual and motor processing in general, across other subcortical areas, like pulvinar, and also cortical areas, like FEF and LIP. Moreover, these sensory-motor integration processes can have direct repercussions on commonly used behavioral paradigms in which microsaccades and saccades happen around the time of attentional cues/probes and can alter performance (*Hafed, 2013*; *Hafed et al., 2015*; *Tian et al., 2016*; *Buonocore et al., 2017*).

## The SC and saccade generation

Consistent with the above sentiment, our study illuminates emerging and classic models of the role of the SC in saccade control. In a recent model by *Goossens and Van Opstal, 2006*, it was suggested that every SC spike during a motor burst contributes a mini-vector of eye movement tendency, such that the aggregate sum of movement tendencies comprises the overall trajectory. Our results are consistent with this model, and related ones also invoking a role of SC activity levels in instantaneous trajectory control (*Waitzman et al., 1991*; *Smalianchuk et al., 2018*), in the sense that we did observe linear contributions of additional SC spikes on eye movement metrics (*Figures 4*, *6* and *7*). However, our results add to this model the notion that there need not be a 'classifier' identifying particular SC spikes as being the movement-related spikes of the current movement and other spikes as being irrelevant. More importantly, we found diminishing returns of relative eccentricity between the 'extra' spikes and the current motor burst (e.g. *Figure 4—figure supplement 3*). According to their model, the more eccentric spikes that we introduced from more eccentric neurons should have each contributed 'mini-vectors' that were actually larger than the 'mini-vectors' contributed by the less eccentric spikes from the less eccentric neurons. So, if anything, we should have expected larger effects for the more eccentric neurons. This was clearly not the case. Therefore, this model needs to consider local and remote interactions more explicitly. The model also needs to consider other factors like input from other areas. Indeed, *Peel et al., 2020* reported that the SC generates fewer saccade-related spikes during FEF inactivation, even for matched saccade amplitudes. Thus, the link between SC motor burst spiking and saccade kinematics is more loose than suggested by the model.

Similarly, we recently found that microsaccades without visual guidance can be associated with substantially fewer active SC neurons than similarly-sized microsaccades with visual guidance, because of so-called visually-dependent saccade-related neurons (*Willeke et al., 2019*). Finally, SC motor bursts themselves are different for saccades directed to upper versus lower visual field locations (*Hafed and Chen, 2016*). All these observations suggest that further research on the functional roles of SC motor bursts is strongly needed.

What other model, then, is more suitable than the mini-vector model and its variants? Classic vector averaging could be appealing. Indeed, dual-site SC microstimulation (*Katnani et al., 2012*) supports vector averaging models, and it is also conceptually similar to our approach. However, in its purest form, vector averaging still cannot account for our behavioral observations with far stimuli and neurons (e.g, *Figure 3—figure supplement 1*). Specifically, like with mini-vectors, vector averaging still predicts larger microsaccades with more eccentric stimuli, but this was not the case.

We are thus left with neither model fully accounting for our observations. However, newer ideas in the literature would be useful here. In recent work, *Jagadisan and Gandhi, 2019* suggested that saccades are not triggered except if there was substantial population temporal alignment among active neurons. Their motivation was to ask why strong SC visual bursts do not automatically trigger saccades, even when they reach similar peak firing rates as motor bursts. They found that motor bursts have stronger temporal alignment between active neurons than visual bursts. This idea is appealing to us because it provides a plausible explanation for why we found the strongest impact of 'visual' spikes on movement metrics in a very specific time window around movement onset (*Figure 6*). In our view, this idea can also account for our weaker behavioral effects with increasing eccentricity. Specifically, because peripheral SC neurons prefer lower spatial frequencies than central SC neurons (*Chen et al., 2018*), the visual bursts of far neurons (for similar stimuli) could be more variable than the bursts of near neurons. This decreases the likelihood of temporal alignment in *Figure 6*, and therefore reduces the behavioral impact of the peripheral spikes. Future work comparing temporal population alignment at different eccentricities, and with and without microsaccade-related motor bursts, would provide experimental support for such a view, and it would extend the *Jagadisan and Gandhi, 2019* hypothesis from one of 'why' a movement is triggered to also one of 'how' the movement specifications are read out by downstream structures.

Another important area that our results can illuminate is related to the question of lateral interactions. In experiment 3 (*Figures 8–12*), we explicitly recorded activity from rostral SC neurons while presenting peripheral visual stimuli. On the one hand, we confirmed that peripheral visual bursts may be associated with reductions in rostral SC activity, as we and others had also previously observed (*Munoz and Istvan, 1998*; *Hafed and Krauzlis, 2008*). This may be consistent with

theories of lateral inhibition across the SC map (*Dorris et al., 2007*; *Isa and Hall, 2009*; *Kasai and Isa, 2016*). However, rostral SC inhibition only occurred in the complete absence of microsaccades (*Figures 9–12*). On the contrary, when microsaccades were triggered simultaneously with peripheral visual bursts, the rostral SC neurons actually exhibited activity bursts (*Figures 9–12*). Therefore, lateral interactions do not necessarily mean that a visual burst at one SC map location automatically implies a pausing of activity at distant locations. Rather, bursts happen in both the rostral and caudal SC, with the caudal 'bursts' kinematically adding to the generated saccades.

### 'Choice probability' in the oculomotor realm

Finally, our analyses in *Figures 4* and *7* are analogous to 'choice probability' analyses in other fields (*Britten et al., 1996*; *Nienborg and Cumming, 2006*). In such analyses, one uncovers a relationship between a single neuron's activity and the global output of the whole brain. In our case, we found that individual 'injected' spikes in the SC correlated remarkably well with saccade metric changes. From this perspective, our observation of differential effects between visual spikes of visual neurons versus visual spikes of visual-motor neurons (*Figure 4—figure supplement 1*) is particularly informative: both visual and visual-motor neurons were linearly related to the saccade amplitudes. However, the impact of a single extra visual spike from a visual-motor neuron was stronger than that of a single extra spike from a visual neuron. This is consistent with suggestions that visual-motor neurons are the SC output neurons (*Mohler and Wurtz, 1976*), and it is also consistent with our earlier observations that under specific conditions, visual-motor neurons are what might dictate saccadic behavior (*Chen and Hafed, 2017*). Revealing functional differences between visual responses of visual versus visual-motor neurons remains to be an interesting open question.

### Conclusion

Our results expose highly plausible neural mechanisms associated with robust behavioral effects on saccades accompanied by nearby visual flashes in a variety of paradigms, and they also motivate revisiting a classic neurophysiological problem, the role of the SC in saccade control, from the perspective of visual-motor multiplexing within individual brain circuits, and even individual neurons themselves.

## Materials and methods

### Animal preparation

We collected data from two adult, male rhesus monkeys (Macaca mulatta) that were 6–8 years of age and weighed 6–8 kg. The experiments were approved (licenses: CIN3/13; CIN4/19G) by ethics committees at the regional governmental offices of the city of Tuebingen and were in accordance with European Union guidelines on animal research and the associated implementations of these guidelines in German law. The monkeys were prepared using standard surgical procedures necessary for behavioral training and intracranial recordings. In short, monkeys N and P had a chamber centered on the midline and aiming at the superior colliculus (SC) with an angle of 35 and 38 degrees posterior of vertical in the sagittal plane, respectively. The details of the surgical procedures were described in previous reports (*Chen and Hafed, 2013*; *Chen et al., 2015*). To record eye movements with high temporal and spatial precision, the monkeys were also implanted with a scleral search coil. This allowed eye tracking using the magnetic induction technique (*Fuchs and Robinson, 1966*; *Judge et al., 1980*). Monkeys N and P were each implanted in the right and left eye, respectively.

### Experimental control system and monkey setup

We used a custom-built real-time experimental control system that drove stimulus presentation and ensured monkey behavioral monitoring and reward delivery. The details of the system are reported in recent publications (*Chen and Hafed, 2013*; *Tian et al., 2016*).

During the testing sessions, the animals were head fixed and seated in a standard primate chair placed at a distance of 74 cm from a CRT monitor. The eye height was aligned with the center of the screen. The room was completely dark with the only light source being the monitor. All stimuli were presented over a uniform gray background (21 Cd/m$^2$). In all the experiments, the fixation spot

consisted of a small square made of 3 by three pixels (about 8.5 by 8.5 min arc) colored in white (72 Cd/m$^2$). The central pixel had the same color as the background.

## Behavioral tasks and electrophysiology

### Experiment 1: injecting visual spikes at the time of saccade generation (contrast task)

We performed a novel analysis of SC data reported on earlier; our behavioral task is therefore described in detail in *Chen et al., 2015*. Briefly, we used a fixation paradigm during which we introduced a peripheral transient visual event at random intervals (see *Figure 1A*). Each trial started with a white fixation spot presented at the center of the display over a uniform gray background. The monkey was required to align its gaze with the fixation spot. Because fixation is an active process, this steady-state fixation paradigm allowed us to have a scenario in which microsaccades were periodically generated (*Hafed and Ignashchenkova, 2013*). After a random interval, we presented a stimulus consisting of a vertical sine wave grating of 2.2 cycles/deg spatial frequency and filling the visual response field (RF) of the recorded neuron. The stimulus onset allowed experimentally injecting visual spikes into the SC at retinotopic locations dissociated from the neurons involved in microsaccade generation (*Hafed et al., 2009*; *Willeke et al., 2019*). Therefore, we could investigate the influence of such injected spiking activity if it happened to occur in the middle of an ongoing microsaccade (see Results). We varied the contrast of the grating across trials in order to vary the amount of injected SC spiking activity around the time of microsaccade generation. Specifically, grating contrast could be one of 5%, 10%, 20%, 40%, or 80% (*Chen et al., 2015*). For the current study, we only analyzed trials with the highest three contrasts. We related microsaccade kinematics to injected 'visual' spiking activity. Overall, we analyzed 84 SC visual (44) and visual-motor (40) neurons in two monkeys. Out of these, 11 neurons (2 visual and nine visual-motor) were also tested on the behavioral task of experiment two below (i.e. within the same sessions). The remaining ones were collected on their own, in separate sessions, and one of them was tested at two stimulus locations in the RF in two successive runs. Across neurons, we analyzed a total of 1150 ± 379 s.d. trials per neuron. These were equally divided across the different stimulus contrasts.

### Experiment 2: injecting visual spikes at the time of saccade generation (spatial frequency task)

We performed a novel analysis of SC data reported on earlier in a different, unrelated study (*Khademi et al., 2020*). The behavioral task was similar to the stimulus contrast task above, but it had two key differences that were particularly useful for the current study. First, the task involved gratings of different spatial frequencies as opposed to different stimulus contrasts. The specific spatial frequencies used were 0.56, 2.2, and 4.4 cycles/deg. This allowed us to demonstrate that any kind of SC visual spiking activity at the time of saccade triggering, irrespective of which source it came from (whether visual contrast or spatial frequency), can be read out in a way to alter ongoing eye movements. Second, and most importantly, this task involved a prolonged fixation period after stimulus onset (up to 1300–3000 ms) (*Khademi et al., 2020*). This allowed us to ask whether our effects were restricted to visual 'bursts' or whether any kind of ongoing SC activity (e.g. during sustained stimulus presentation long after the ends of 'visual' bursts) can be read out to alter ongoing eye movements. We analyzed the activity of 55 neurons from this task (31 visual and 24 visual motor); 11 of these neurons were also tested with the contrast task described above within the same sessions. The remaining neurons were collected in separate sessions. In all cases, the stimulus was placed within the visual RF of a given recorded neuron (*Figure 1*). Across neurons, we analyzed a total of 266 ± 169 s.d. trials per neuron. These trials were equally divided among the three different spatial frequencies presented.

### Experiment 3: microelectrode array recordings in either the rostral SC, the caudal SC, or both simultaneously

In monkey N, we performed new recording experiments to explore modulations in the rostral SC (where movement bursts are expected to occur for microsaccades) at the time of peripheral 'visual' bursts. The monkey maintained fixation on a similar fixation spot to that we used in experiments 1 and 2 above, and we presented a white disc of 0.5 deg radius at an eccentric location. When we

recorded from the rostral SC (representing foveal eccentricities), the eccentric location (i.e. the potential location of the white disc) was chosen to be at 10 deg either to the right or left of fixation (i.e. well away from the microsaccade endpoints). That is, across trials, we sampled visual stimuli activating the caudal portion of either the same or opposite SC as the recorded rostral SC site. When we recorded from the caudal SC (representing more eccentric visual field locations), the white disc appeared either within the visual RF of the caudal SC neurons being recorded at the site or in a diametrically opposite location. The monkey simply maintained fixation. The exact trial sequence in the experiment was similar to our earlier instantiation of the cueing task in *Tian et al., 2018*. That is, the white disc first appeared for approximately 32 ms, and then after a random time of up to 1 s, the disc appeared again at either the same or opposite location. Unlike in *Tian et al., 2018*, the monkey did not generate a saccade; rather, the monkey just maintained fixation and touched a bar after the second stimulus onset. We collected a total of 799 ± 272 s.d. trials per session. We counter-balanced first and second disc appearance location per trial (e.g. first at 10 deg right and second at 10 deg left, or first at 10 deg right and second at 10 deg right, and so on) across all trials. Since we were primarily interested in demonstrating that there is a visual burst no matter whether there are coincident microsaccades being triggered or not, we combined measurements of visual bursts for both the first and second disc appearance. Similarly, for the rostral SC, we were primarily interested in demonstrating that there is a motor microsaccade-related burst whether or not there is a peripheral visual burst; we therefore again combined first and second disc appearances per trial in analyses.

We inserted 16-channel linear microelectrode arrays (V Probes; Plexon, Inc) into either the rostral or caudal SC (23 and 17 sessions, respectively). In yet an additional set of sessions (13 sessions), we inserted two arrays simultaneously, one in the rostral SC and one in the caudal SC. To aid in the technical insertion of the two arrays, we inserted them into separate SC's (e.g. right rostral SC and left caudal SC) because this gave us slightly more lateral separation between the microelectrode arrays for simultaneous insertion. Across all sessions, we isolated offline (see below) 42 rostral SC neurons and 54 caudal SC neurons for further analysis in the current study. Out of these, 15 rostral and 19 caudal neurons were recorded during the simultaneous recording sessions. To identify the sites that we were recording from, the monkey was also engaged in standard eye movement tasks similar to those described using similar experiments recently (*Willeke et al., 2019*). These tasks allowed us to map the visual and movement-related RF's of the recorded neurons.

## Data analysis
### Experiment one data analysis
All analyses were performed with Matlab (MathWorks, Inc). Most of the analyses involved grouping the eye movement data into groups of movements going either towards or opposite a recorded neuron's RF. To make this classification, we first calculated the angle of the RF relative to the fixation spot. Then, all eye movements with an angle ±90 degrees around the RF direction were classified as being 'towards' the RF. All remaining eye movements were classified as being directed 'opposite' to the RF. Movement angles were defined as the arctangent subtended by the horizontal and vertical component between movement onset and end. RF angles were defined as the arctangent subtended by the horizontal and vertical coordinates of the RF locations relative to the fixation spot.

To confirm that none of our neurons exhibited movement-related discharge for the saccades that we studied (*Figure 1—figure supplement 1*), we searched for all microsaccades occurring in a pre-stimulus baseline interval of 25–100 ms before stimulus onset (i.e. in the absence of any eccentric visual stimuli). We then aligned all neural discharge to all microsaccades, and we confirmed that there was no activity elevation either towards or away from the recorded neuron's RF. We also did this analysis separately for visual and visual-motor neurons. The former were expected not to show any movement-related discharge, by definition. The latter were only expected to exhibit discharge for much larger saccades than the ones we studied, and this analysis confirmed this. This means that our modulated microsaccade amplitudes in Results (e.g. *Figure 4*) were unlikely to be simply explained by the idea that the spikes that we measured were 'motor' bursts for microsaccades (e.g. see Discussion).

To analyze peak firing rates 'without saccades' (e.g. *Figures 2* and *3B*, *Figure 2—figure supplement 1*, *Figure 3—figure supplement 1B*), we selected all trials in which there were no

microsaccades btween −100 ms and 200 ms relative to stimulus onset. We then averaged all the firing rates across trials, and we determined the peak firing rate for each neuron from the across-trial average curve. For the peak firing rate 'with saccades', we took all the trials in which a microsaccade was either starting or ending during the so-called visual burst interval, which we defined to be the interval 30–100 ms after stimulus onset. Paired-sample t-tests were performed to test the influence of saccades on the peak visual burst with an α level of 0.05 unless otherwise stated (e.g. *Figure 2*, *Figure 2—figure supplement 1*).

To summarize the time courses of microsaccade amplitudes after stimulus onset (e.g. *Figure 3A*, *Figure 3—figure supplement 1A*), we selected the first saccade of each trial that was triggered within the interval from −100 ms to +150 ms relative to stimulus onset. All the microsaccade amplitudes were then pooled together across monkeys and sessions. The microsaccade amplitude time course was obtained by filtering the data with a running average window of 50 ms with a step size of 10 ms. To statistically test the effect of grating contrasts on these time courses, we performed a one-way ANOVA on saccade amplitudes for all saccades occurring between 50 ms and 100 ms after stimulus onset. To compare the effect of grating contrast on SC visual bursts (e.g. *Figure 3B*, *Figure 3—figure supplement 1B*), for each neuron, we normalized the firing rate based on the maximum firing rate elicited by the strongest contrast. Subsequently, we calculated the mean firing rate of the population and the 95% confidence interval for the different contrast levels. Both the amplitude and the firing rate analyses focused on the three highest contrasts because the first two contrast levels did not have a visible impact on eye movement behavior.

For some analyses (*Figures 4–7* and their supplements), we explored the relationship between the number of 'visual' spikes emitted by a recorded neuron and saccade amplitude. This allowed us to directly investigate the effect of each single additional spike per recorded neuron in the SC map on an ongoing saccade, irrespective of other variables. We selected all the trials in which an eye movement was performed in the direction of the grating soon after its presentation (0–200 ms from stimulus onset). The analysis was restricted to the three highest contrasts, since they had a clear effect on the eye movement behavior and also had a clear visual burst (e.g. *Figure 3A*). For each selected saccade, we counted the number of spikes happening from a given recorded neuron during the interval 0–20 ms after movement onset. This interval was constant, irrespective of saccade size, and it was early enough to be read out and influence the eye movement. Note that if there was no spike in this interval, then this meant that the microsaccade was classified as having '0 spikes' (e.g. in *Figure 4*); therefore, the microsaccade was still occurring soon after stimulus onset but without a coincidence of peripheral spiking at its onset. For *Figure 4A*, *Figure 4—figure supplement 2A*, we calculated the 'radial eye position' from saccade start as the Euclidian distance of any eye position sample (i.e. at any millisecond) recorded during an eye movement relative to the eye position at movement onset (see for similar procedures: *Hafed et al., 2009*; *Buonocore et al., 2017*). We also plotted radial eye velocity for the same movements (e.g. *Figure 4C*). To make statistical inferences on the effects of the number of spikes on saccade amplitude (e.g. *Figure 4B*, *Figure 4—figure supplement 2B*), we proceeded by fitting a generalized linear model to the raw data with equation: $y = \beta_0 + \beta_1 * x$ where 'x' was our predictor variable, the number of spikes, and 'y' was the predicted amplitude. The parameters fitted were: $\beta_0$ the intercept, $\beta_1$ the slope. We imposed a cutoff of at least 15 trials for each level of the predictor, leading to exclusion of spike counts bigger than five. In a different variant of this analysis, we repeated it for only either visual or visual-motor neurons (e.g. *Figure 4—figure supplement 1*). Note that in all other analyses in this study, we decided to combine visual and visual-motor neurons for purposes of clarity. This was not problematic; if anything, it only muted our results slightly (rather than amplified them) since visual-motor neurons showed even stronger effects in general than visual neurons (see Results).

Also note that our choice of 0–20 ms from movement onset in the above analyses was to strictly enforce that the 'visual' spikes associated with bigger movements were intra-saccadic. Our other analyses (e.g. *Figure 6*) showed that we would have obtained even bigger effects had we include 'visual' spikes also right before movement onset. Either way, 0–20 ms was still early enough within the eye movements to cause measurable impacts (e.g. *Figure 4*).

To explore the impact of neuronal preferred eccentricity on the relationship between the number of spikes emitted by a given eccentric neuron and movement amplitude, we repeated the above generalized linear model analysis approach, but this time for specific eccentricity ranges of neurons. Specifically, we used a sliding window on neuronal preferred eccentricity. Starting with a preferred

eccentricity of 1 deg, we defined a window centered on this and having 2 deg width. We then slid this 2 deg window in steps of 1 deg eccentricity. For each sliding window eccentricity center, we estimated the slope of the linear model described above, but this time from only neurons having a preferred eccentricity within the current sliding window location. We then plotted the slope of the relationship (i.e. the slope of the line relating saccade amplitude and the number of spikes) as a function of neuronal preferred eccentricity. This allowed us to confirm that our choice to focus on eccentricities less than or equal to 4.5 deg in most of our analyses in this study was a valid approach (*Figure 4—figure supplement 3*). It also allowed us to investigate whether the slope ever turned negative for the most eccentric neurons, which was not the case (see Results). For *Figure 4—figure supplement 3B*, we used the estimates of s.e.m. obtained from the generalized linear model fits to visually present the s.e.m. ranges of the shown slopes.

In all the above generalized linear model analyses, we included the 0 spike movements in the data fits. However, it turned out that we could have also excluded these movements, and the same results would have been obtained (and this was also true in the eccentricity effects of *Figure 4—figure supplement 3*). For simplicity, in Results, we showed only the analyses with the '0 spike' movements included as parts of the generalized linear model fits.

To study the time window of influence of each added spike on saccade amplitudes (*Figure 5*), we generated raster plots from all trials of all sessions by aligning each spike raster trace to the saccade occurring on the same trial (either saccade start, peak velocity, or end). The saccades were chosen as those that happened after stimulus onset and being directed towards the RF locations, and the alignment was based on the time of peak visual burst after stimulus onset on a given trial relative to the time of the movement. We selected data from the three highest stimulus contrasts and also with eye movements directed towards the RF, since the modulation in behavior was most pronounced in these cases (e.g. *Figure 4*). We also focused on neurons with preferred eccentricities $\leq$ 4.5 deg. To identify the point at which the amplitude diverged from a baseline level in *Figure 5D*, we first sorted all the amplitudes based on burst time relative to saccade onset (*Figure 5A*). Then, we made bins of 30 trials each from which we derived the mean amplitude values (*Figure 5D* shows something similar but with a moving average of 20 trials, in steps of 1 trial, just for visualization purposes). We also tested the amplitudes of each 30-trial bin against the first one (baseline amplitude) to determine when the amplitude increase was significant. To do so, we performed two-sample independent t-tests between each pair (i.e. the current time bin and the first baseline time bin), and we adjusted the alpha level with Bonferroni correction. We chose as an index for a significant increase in amplitude the point at which three consecutive bins were significantly different from the baseline (first horizontal blue line in the trial sorting of *Figure 5D*). The next three consecutive bins that did not differ anymore from the baseline indicated that the amplitude increase was not significant anymore (second horizontal blue line in the trial sorting of *Figure 5D*).

We also repeated the analyses of *Figure 5* but for the far neurons having preferred eccentricities > 4.5 deg (*Figure 5—figure supplement 2*). We used the same procedures as described above. However, since there was a significantly larger number of trials, we used 100-trial bins rather than 30-trial bins for the running statistical tests used to assess the locations of the horizontal blue lines in *Figure 5—figure supplement 2*. For the visualization of the microsaccade amplitudes in panel D of the same figure, we also used a moving average of 50 trials (in steps of 10 trial) instead of a moving average of 20 trials as in *Figure 5*. Again, this was done only to result in smoother looking curves in panel D; the conclusions based on the running statistical tests did not depend on the visualization binning. We also did not use Bonferroni correction in the running tests, just for simplicity. This was fine, especially because, in any case, we quantitatively assessed the time window of maximal impact by peripheral spiking on microsaccade amplitudes in *Figure 6—figure supplement 2* for the same data set; therefore, in this sense, *Figure 5—figure supplement 2* was more suitable as a visualization of the impact of spike timing relative to movement onset, whereas *Figure 6—figure supplement 2* provided a more quantitative estimate of the relative timing relationship between the far spiking and the movements that were altered by such far spiking.

For *Figure 6*, for each movement amplitude range from the data in *Figure 5*, we identified 'how many' and 'when' visual spikes in an eccentric neuron occurred in association with this amplitude range. This allowed us to identify a window of time in which injected 'visual' spikes had the most effect on saccade amplitudes. For each movement, we estimated the average number of spikes to occur within a given 5 ms time window around saccade onset. We also repeated this same analysis

for the far neurons of the same experiment, which had preferred eccentricities > 4.5 deg (*Figure 6—figure supplement 2*).

## Experiment two data analysis

We repeated all the visual burst analyses described above for experiment 1 (the contrast task). This allowed us to confirm the robustness of all results from experiment 1. We pooled across spatial frequencies when investigating the impact of a given number of spikes on movement amplitude (e.g. *Figure 4—figure supplement 4* or *Figure 6—figure supplement 1*).

For *Figure 5—figure supplement 1*, we repeated the same analysis steps of *Figure 5*. Here, we used 20-trial moving average bins (in panel D of the figure), and also 20-trial bins for the Bonferroni corrected running statistical tests used to obtain the horizontal blue lines in the figure. Similarly, we also assessed the specific time window for maximal impact of visual spiking on microsaccades in *Figure 6—figure supplement 1*.

To demonstrate that even spiking activity long after visual bursts could still influence movement amplitudes (*Figure 7*), we collected all microsaccades occurring >550 ms after stimulus onset. We then analyzed the relationship between spiking activity and movement metrics exactly like we did for the analyses with spikes coming during the 'visual burst interval'. The results are shown in *Figure 7* and *Figure 7—figure supplement 1*. For the summary plots of amplitude as a function of number of intra-saccadic spikes, we included trials from all spatial frequencies.

## Experiment three data analysis

Neurons in experiments 1 and 2 were sorted online for their respective previous studies (*Chen et al., 2015*; *Khademi et al., 2020*). For experiment 3, we sorted the neurons offline using the Kilosort toolbox (*Pachitariu et al., 2016*). Briefly, after a semi-automated spike detection and classification step, we manually inspected all isolated units based on auto- and cross-correlograms, as well as waveform shapes. Any atypical sorting results (such as irregular waveforms, no clear characteristic auto-correlograms, or violations of refractory periods) were excluded from our analyzed database. Besides these strict mathematical sorting quantifications, we also verified the sorting results based on RF and neuronal properties in the classic delayed visually guided and memory-guided saccade tasks. All isolated units within a simultaneously recorded session (e.g. across the 16 channels of a single microelectrode array) had high similarity in their movement RF locations (quantified by Pearson's correlations: average correlation value $0.875 \pm 0.002$ s.e.m., $p<0.0001$).

Visual and movement RF's in these new recordings were assessed using our standard delayed visually guided and memory-guided saccade tasks (*Munoz and Wurtz, 1995*; *Li and Basso, 2008*; *Chen et al., 2015*; *Willeke et al., 2019*). For plotting microsaccade-related movement RF's of the rostral neurons, we plotted the raw peak firing rate as a function of microsaccade amplitude and direction (the illustrations of rostral RF's in Results are based on the memory-guided saccade mapping task). We then fitted the resulting data with a two-dimensional gaussian function. This allowed us to estimate a region of interest (e.g. movements within 2 s.d. radii from the peaks) for plotting firing rates as a function of time from microsaccade onset (e.g. *Figure 9B,G*). For visual and movement RF plots, we also used log-polar axes to cover the large range of eccentricities used (*Hafed and Krauzlis, 2012*; *Willeke et al., 2019*).

Our comparison of interest for caudal visual burst analyses in this experiment was firing rates with no microsaccades occurring −50–200 ms relative to peripheral stimulus onset or firing rates with microsaccades occurring towards the rostral movement RF direction during a visual burst interval (e.g. 30–100 ms). We therefore split trials based on whether microsaccades happened or not. For rostral SC neurons, in *Figure 12*, our comparison of interest was microsaccade-related bursts in baseline (pre-stimulus interval) or when microsaccades occurred during the visual burst interval (e.g. 30–100 ms after peripheral stimulus onset), with the microsaccades being directed towards the RF hotspot location in both cases. The baseline microsaccades were defined as those occurring 500–1500 ms before peripheral stimulus onset.

To summarize results across neurons, we obtained average population firing rates by first normalizing each neuron's response to its maximum and then averaging across neurons. For rostral neurons, we identified the 'preferred' microsaccades (i.e. those resulting in the highest peri-microsaccadic firing rates; our region of interest in the RF). We then normalized each neuron's

activity by the peak microsaccade-related activity for the preferred microsaccades from the RF mapping tasks. We then averaged across neurons in the main task. The average could be either for baseline microsaccades in the absence of a peripheral visual stimulus (for microsaccades occurring 500–1500 ms before stimulus onset), or it could be for microsaccades towards the movement RF starting 30–100 ms after stimulus onset (i.e. during the peripheral visual burst); see for example *Figure 12* with the raw measurements. When we did show normalized firing rates (e.g. *Figure 11A*), the normalization factor for both conditions was the same (the peak firing of the neuron in baseline, based on the region of interest in the RF mapping tasks). For caudal neurons, the stimulus in the visual RF was always at a fixed location across trials in the main task. We therefore took all trials in which no microsaccades occurred −50–150 ms from stimulus onset in the main task. We then averaged the firing rate for these trials, and we then used this as the normalization factor. We normalized all trial firing rates by this normalization factor (again, dividing the individual trial firing rates by this normalization factor), including the trials in which microsaccades happened during the visual burst interval (30–100 ms after stimulus onset). We then averaged the neurons' normalized firing rates to obtain a population average response.

For the behavioral analyses in *Figure 12B*, we measured microsaccade amplitudes from all of the sessions of experiment 3, but for either pre-stimulus microsaccades (500–1500 ms before peripheral stimulus onset) or for microsaccades occurring 30–100 ms after peripheral stimulus onset. In both cases, we picked microsaccades with angular directions within ±25 deg from peripheral stimulus direction. This was so because amplitude increases for the microsaccades are expected to happen for movements towards the recently appearing peripheral stimuli (e.g. *Figure 3*).

## Acknowledgements

We were funded by the Deutsche Forschungsgemeinschaft (DFG) through the Research Unit: FOR1847 (project A6: HA6749/2-1). We were also funded by the Werner Reichardt Centre for Integrative Neuroscience (CIN; DFG EXC307). ZMH and FK were additionally supported by the DFG Collaborative Research Centre: Robust Vision (SFB1233; TP 11; project number 276693517).

## Additional information

### Funding

| Funder | Grant reference number | Author |
|---|---|---|
| Deutsche Forschungsgemeinschaft | FOR1847 (project A6: HA6749/2-1) | Antimo Buonocore Ziad M Hafed |
| Deutsche Forschungsgemeinschaft | DFG EXC307 | Antimo Buonocore Xiaoguang Tian Ziad M Hafed |
| Deutsche Forschungsgemeinschaft | SFB1233 "Robust Vision" (project TP11; project number 276693517) | Fatemeh Khademi Ziad M Hafed |

The funders had no role in study design, data collection and interpretation, or the decision to submit the work for publication.

### Author contributions

Antimo Buonocore, Conceptualization, Formal analysis, Validation, Visualization, Writing - original draft, Writing - review and editing; Xiaoguang Tian, Data curation, Formal analysis, Validation, Investigation, Visualization, Writing - review and editing; Fatemeh Khademi, Formal analysis, Validation, Visualization, Writing - review and editing; Ziad M Hafed, Conceptualization, Resources, Data curation, Formal analysis, Supervision, Funding acquisition, Validation, Investigation, Visualization, Writing - original draft, Project administration, Writing - review and editing

## Author ORCIDs

Antimo Buonocore (iD) https://orcid.org/0000-0003-3917-510X
Fatemeh Khademi (iD) https://orcid.org/0000-0002-1854-4343
Ziad M Hafed (iD) http://orcid.org/0000-0001-9968-119X

## Ethics

Animal experimentation: The experiments were approved (licenses: CIN3/13; CIN4/19G) by ethics committees at the regional governmental offices of the city of Tuebingen and were in accordance with European Union guidelines on animal research and the associated implementations of these guidelines in German law.

## Decision letter and Author response

Decision letter https://doi.org/10.7554/eLife.64150.sa1
Author response https://doi.org/10.7554/eLife.64150.sa2

# Additional files

## Supplementary files

• Transparent reporting form

## Data availability

All data generated or analysed during this study are included in the manuscript and supporting files. Source data files have been provided for all figures.

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
