## [Decision Letter]

**Acceptance summary:**

The brain can rapidly adjust its actions on the fly when new sensory information becomes available. This study addresses how microsaccadic eye movements are influenced by a suddenly presented visual stimulus in the periphery, and how this depends on interactions within the superior colliculus. Whenever visually evoked activity occurs in a small temporal window around microsaccade execution, the visual spikes modify the amplitude of the microsaccade. These visual spikes influence the motor command in a lawful manner such that each extra spike from the same SC increases systematically the amplitude of the microsaccade. These findings shed new light on how visual information is integrated into ongoing saccadic motor commands.

**Decision letter after peer review:**

[Editors’ note: the authors submitted for reconsideration following the decision after peer review. What follows is the decision letter after the first round of review.]

Thank you for submitting your work entitled "Instantaneous movement-unrelated midbrain activity modifies ongoing eye movements" for consideration by *eLife*. Your article has been reviewed by 4 peer reviewers, one of whom is a member of our Board of Reviewing Editors, and the evaluation has been overseen by a Senior Editor. The following individual involved in review of your submission has agreed to reveal their identity: Terrence R Stanford (Reviewer #3).

Our decision has been reached after consultation between the reviewers. Based on these discussions and the individual reviews below, we regret to inform you that your work will not be considered further for publication in *eLife*. As you will see from the reviews, the reviewers expressed general interests and made many positive remarks. However, they also agreed on major issues that would require a substantial amount of additional experimentation and analysis. Some of the major issues include (1) There were major concerns about the relationship and consistency between experiment 1 and 2, and the interpretation of the microstimulation experiment. (2) The results of Experiment 1 might have several other interpretations including visual enhancement and competitive interactions, and could be influenced by several confounds. Within the domain on mechanisms of saccade generation, the analyses are not detailed enough to test competing models and provide conclusive mechanistic insight.

We would welcome submission of a new manuscript if it is possible to address these issues, but we judged them to be sufficiently significant that we thought it better to reject at this stage. Should you go this route, we would likely go to the same reviewers.

*Reviewer #1:*

This study investigates the influence of visual stimuli and spiking activity in distant SC (superior colliculus) locations on small microsaccadic movements during fixation. The authors show that visual stimuli and activity in these distant locations modulates the ongoing microsaccadic movements. The authors find however that nearby locations have more powerful effects than distant locations, which challenges existing models based on vector-averaging (as one would expect distant locations to exert more powerful effects). Overall these findings provide a very interesting addition to the literature. My main critique is the possibility of lateral interactions or influence on the activity of more foveal neurons.

1. If I understand correctly, the authors suppose that the effects on the saccade trajectory are mediated by the projections of neurons in distal locations to other brain areas. However there are lateral inhibitory interactions within the superior colliculus and the effects reported here could potentially be mediated by these lateral interactions that modulate the activity of neurons driving the microsaccadese.

2. A related point is that the visual stimulus might exert some effects directly on the neurons driving the microsaccades, either through lateral mechanisms within the retina or within the superior colliculus.

The best control would be to analyze directly the neurons in foveal regions. What happens there? Is their firing rate affected or not? It appears that the authors have this data but they do not report this.

3. The authors should provide more raster plots of activity of neurons and describe more clearly how many neurons were recorded/analyzed throughout the legends and results text.

4. What functional cell types are recorded by the authors? This should be described.

*Reviewer #2:*

The superior colliculus (SC) is recognized for its role in sensorimotor transformations. A vast number of SC neurons discharge both when a stimulus is presented in their response fields and also when a saccade is directed to that location. Mechanisms have been proposed for differentiating 'sensory' from 'motor' activity across the SC population and, furthermore, computing the metrics and/or kinematics of the observed saccade. This study shows that if additional spikes, whether evoked by a visual stimulus or through electrical stimulation, are introduced at a SC location away from the active 'motor' population and around the time of saccade, then these spikes contribute to the size of the saccade. Amazingly, effects of adding single spikes per neuron (but from many neurons) are noticeable in the analyses. It is a thought-provoking finding and, provided several serious concerns are mitigated satisfactorily, has implications on multiple facets of SC role in saccade generation.

1. The data shown in Figure 2 do not align with the working model of the SC. A fairly well-known study by Dorris, Olivier, and Munoz (https://doi.org/10.1523/JNEUROSCI.4212-06.2007), which surprisingly is not cited here, showed that a visual stimulus (distractor) interferes with motor preparatory activity elsewhere in the SC. The preparatory activity is enhanced if the target and distractor populations are in close proximity but suppressed if the distractor is far way, including in the opposite hemifield. This finding – complemented by slice studies by Isa, Hall, and others -

has established a framework for local excitation and mutual distal inhibition in the SC. What is shown in Figure 2 does not conform to this framework and, if true, it is too important a result to be overlooked.

a. I realize that the activity recorded in this study is from neurons at the "visual", rather than the "motor", site. However, if interactions between the two populations are mutual, then the effect should arguably be observed in both populations. Should we be rethinking this conclusion – are the competitive interactions not mutually effective?

b. The motor activity relevant here is in the rostral SC and associated with microsaccades, while the Dorris study focused on more caudal regions that encode larger amplitude movements. However, an underlying theme of a large body of Hafed's studies is that our knowledge of large amplitude saccades and their neural control extend to microsaccades also. In fact, this is the justification used for the microstimulation experiments reported in the manuscript. Should we be second guessing this conclusion – perhaps there is something unique about microsaccade control?

c. The relative timing of activity in the 'visual' and 'motor' populations are different in the two studies. I believe Dorris et al. only focused on trials when distractor-driven activity occurred during saccade preparation rather than during execution. In contrast, this Buonocore et al. manuscript draws our attention to when the two bursts are effectively coincident. But it is not clear why competitive interaction effect should disappear during a saccade.

d. These results are also inconsistent with the large body of literature on saccadic or visual suppression in the SC (e.g., D.L. Robinson and Wurtz).

e. It would be tremendously valuable if the authors also show what happens to the motor burst when visual spikes are added elsewhere in the SC, not just during the execution phase of a microsaccade but also in the preparatory period preceding it. Given that the Hafed lab does a lot of neural recordings in the rostral SC during microsaccades, the data may already be available or easily collectable. Without this data, it is difficult to offer full support of the results presented in this manuscript.

2. The data in Figures 4 and 5 are rather remarkable. They make a compelling case for systematic increase in saccade amplitude as additional spikes/neuron are coincident with the motor burst. A few points of clarification:

a. Please also include velocity profiles in Figure 4A. Are they bell-shaped or like control movements? Or do the velocity profiles exhibit an inflection point or multiple peaks (like that observed for the microstimulation experiments reported later in the manuscript)?

b. As best as I can tell, the data are pooled across all trials and all neurons (like in Figure 5). If true, please make this (more) explicit in the text and figure caption. It also raises the concern that the effect may be weighted more by the subset of neurons that have substantially more trials than other neurons. Please provide summary of number of trials per neuron. Importantly, provide insights into how many neurons' data conform to the results shown in Figure 4.

c. If I am mistaken and the analysis is performed one neuron at a time, then please underlay in Figure 4B the distribution of all points for each x-axis value (as in Figure S4).

3. Several aspects of Figure 6 are bothersome and require clarity. Frankly, it is not clear that the effect the authors want to emphasize is actually present in the data.

a. In Figure 6C, there aren't enough points for 3 and 4 intra-saccadic spikes to justify using mean as the average measure. Median is a better choice and, in this case, the suggested effect will likely not exist. This will likely change the text on Lines 374-378, which incidentally deserves to be backed with a statistical test.

b. In Figure 6C and 6D, color the dots according to the histograms in Figure 6A and 6B, respectively. Basically, the color of dots should change from dark red (green) to light orange (yellow) as y-axis value increases.

c. Data in Figure 6A and 6C suggest some discrepancy. Figure 6A indicates that microsaccades larger than 1 deg were observed, but Figure 6C does not show any saccades greater than one degree. Same concern for Figures 6B and 6D.

4. I am less excited about the experiments using microstimulation during visually-guided saccades.

a. The authors want us to accept that the relationship between the sensory and motor populations of neurons remains the same in the two experimental paradigms (L424-430). I would've accepted this premise prior to reading this manuscript, but the lack of competitive interactions makes me question this assumption, particularly as the electrode placement moves away from the rostral SC.

b. Example velocity traces are not unimodal. Perhaps this is due to stimulation occurring late within a saccade, but it looks more like the ongoing saccades is truncated and potentially replaced by another movement command. These results are indicative of competitive interactions.

5. Data in Figure 8 are difficult to interpret.

a. Normalized amplitude increases earlier for 1 electrical spike compared to 2 and 3 spikes. Is this significant and how can this be? We can't interpret this result until we have a sense of variability in normalized saccadic amplitude in the absence of stimulation.

b. Given the arguments raised above about Figure 7, this analysis should arguably not include displacement associated with the second bell-shape in the velocity profile.

c. Saccade amplitude peaks when microstimulation and saccade onsets are coincidental, but according to Figure 7, the effect is observed as a second peak in velocity, which occurs 40-50 ms later, and this is substantially longer than the efferent delay. So, pieces are not adding up.

*Reviewer #3:*

Overall, this is a strong manuscript that considers a topic of significant interest to those interested in visuomotor control and attention. Main strengths: 1) It provides important insight for understanding the kinematics of saccades that would be made under natural viewing circumstances and 2) good experimental design and analysis have yielded a compelling dataset that speaks to the main conclusion that currently executed SC motor commands are influenced by the instantaneous readout of any concurrent visually-evoked activity within the SC map. Weakness: Although the study provides an important additional piece to the SC readout puzzle, it does seem to strongly constrain possible mechanistic frameworks.

Specific comments:

The experiment in which visually evoked activity is "injected" and alters the metrics of microsaccades is elegant and compelling. To me this is what makes the manuscript worthy of publication as it shows quite definitely that such activity is read-out as part of the ongoing motor command. This has important implications for understanding the execution of saccades during natural scanning when the perception/action sequence is considerably more fluid than for a typical laboratory task. With respect to this main strength, the results are clearly presented and interpreted.

The results of Experiment 2 are also well presented, but, in my view, there are some issues with interpretation that might be remedied with further discussion.

The microstimulation results are interesting but think there are some issues that limit the conclusions that one might draw regarding the SC readout mechanism.

1. The injection of microstimulation pulses is suggested as an analog of "injecting" visual activity. But not all SC saccade-related neurons are visually responsive, and the electrical stimulation is certain to impact visual-motor and motor cells alike. Thus, microstimulation does not so cleanly tap visually driven activity as does the visual stimulus employed in the first experiment. That microstimulation alone does not trigger saccades does not mitigate this concern since 1-3 stimulus pulses would not be expected to trigger a saccade in any case. Some consideration of this distinction seems warranted.

2. There have been many variants of SC readout models that are or are not consistent with producing the form of input required of brainstem local feedback circuit models as originally proposed by David Robinson and others (e.g., Jurgens et al., 1981) and which likewise either support or supplant the vector averaging scheme proposed by Sparks and colleagues (Lee et al., 1988). It is not entirely clear how the present microstimulation data inform or distinguish between traditional or later developed frameworks. For example, given the spatial configuration employed here (i.e., in line, stim always more eccentric), I could imagine how a simple vector averaging scheme could yield the linear increase in amplitude with additional pulses if one assumes the pulses produce a transient modification of the SC's desired displacement signal. But then, I think more recent schemes (e.g., that of vector summation) could work equally well. More discussion of how these data constrain current models would be welcome.

3. Related to the above, the microstimulation results are not unexpected based on how one would expect it to interact with an ongoing movement. This we could intuit from extant literature (as properly cited by the authors). The weakening of the effect with eccentricity is new and interesting, but I wonder if it is specific to eccentricity or if it generalizes in some way to distance in an SC spatial coordinate frame.

4. I understand that microsaccades, provide only the option of a more eccentric stimulus or stimulation, however, it seems like an experimental variant in which the "ectopic" stimulus was less eccentric might have been of use in distinguishing between readout mechanisms. I'm not suggesting this as a necessary addition, but some discussion might be valuable.

5. There is a recent paper by Gandhi and colleagues (Smalianchuk et al., 2018) that seems quite relevant as it relates to instantaneous SC readout mechanisms. Although it does not deal with "visual" activity, there is enough conceptual overlap that it should be referenced and/or discussed at the authors' discretion.

*Reviewer #4:*

In this study, Buonocore and colleagues investigate the influence of visual neural activity in the superior colliculus (SC) on eye-movements. In the first experiment, they show that visual activity in the SC – induced by a peripheral grating – was unaffected by concurrent microsaccades. Interestingly, the size of microsaccades was biased by the visual stimulation, and the higher the number of spikes around the its onset, the larger the amplitude of the eye-movement. In a second experiment, using electrical microstimulation of the SC during a saccade task, they show that even a small number of induced spikes (3) was enough to bias the animal's eye-movement response towards the stimulated site. Overall, this is an interesting study, with significant implications for saccade generation models in the SC.

The authors often use selection criteria for their data analysis that is not clearly justified. One criterion that might be particularly problematic is to focus most of their analysis on eccentricities <= 4.5 deg. Their justification for this is that "these had the strongest effects on microsaccades" (line 189). Analyses for values > 4.5 deg, showing usually just weaker effects, are presented in the supplementary materials. My main concern is that this approach might be hiding an underlying interaction between eccentricity, SC firing rate, and microsaccade amplitude, which could contradict the other findings presented. The firing rate values in Figure S4 suggest that there might be a positive correlation between eccentricity and SC firing rate (higher firing for more eccentric RFs). While microsaccade amplitudes seem negatively correlated with eccentricity. If this is true, the analysis pooling together all data would suggest that higher firing would be associated with smaller amplitudes (the opposite of what is shown here). Of course, that eccentricity could be considered a confounding variable. So, one possible way to deal with this problem would be to perform a regression analysis to predict microsaccade amplitudes using both eccentricity and SC activity as independent variables.

The analysis presented in Figure 4 shows a clear relationship between the number of spikes in SC and microsaccade amplitude. This is a very interesting finding. However, I see two potential problems here. One, that by sorting their data on 'number of spikes', they might inadvertently be also splitting their data by contrast (a result already shown in Figure 3). Second, they count the number of spikes from saccade onset to peak velocity as a criterion. The problem is that, because of the ballistic nature of saccades, this interval would linearly increase with saccade amplitude. Given the result presented in Figure 6A, I would not expect that this would compromise their main finding, but the authors should nonetheless correct for this by using a fixed time-window across all saccade amplitude conditions.

The statistical inferences from experiment 2 are mostly done one "saccadic displacement", calculated as the "radial position of the eyes from saccade start to 100 ms thereafter" (line 814). I might be missing something, but I don't understand why they use this criterion instead of, for example, the normalized distance of the saccade offset to the target. It seems to me that they might yield similar results, but since most saccades are finished by 60-80ms (Figure 7CD), the 100ms zone might be contaminated by corrective eye-movements.

[Editors’ note: further revisions were suggested prior to acceptance, as described below.]

Thank you for submitting your article "Instantaneous movement-unrelated midbrain activity modifies ongoing eye movements" for consideration by *eLife*. Your article has been reviewed by 4 peer reviewers, one of whom is a member of our Board of Reviewing Editors, and the evaluation has been overseen by Richard Ivry as the Senior Editor. The following individual involved in review of your submission has agreed to reveal their identity: Terrence R Stanford (Reviewer #4).

The reviewers have discussed the reviews with one another and the Reviewing Editor has drafted this decision to help you prepare a revised submission.

Summary:

This manuscript explores how microsaccades produced during active fixation are altered when a visual stimulus is flashed at parafoveal and more eccentric locations in the visual world. Focusing on neural activity in the superior colliculus (SC), the authors show that whenever visually evoked activity occurs in a small temporal window around microsaccade execution, the "visual" spikes modify the amplitude of the microsaccade. More specifically, they propose the additional spikes "leak" into the motor command in a lawful manner such that each extra spike from the same SC increases systematically the amplitude of the microsaccade.

Essential revisions:

Overall several concerns remain about the data analysis and interpretation of the findings of the authors, especially concerning the mechanistic interpretation of the findings. These mechanistic interpretations require additional analysis and further discussion or modification.

1. Rostral SC and intracollicular interactions: Concerns related to intracollicular interactions were salient in the initial round of reviews. These concerns still remain to some extent. It remains unclear whether the authors have convincingly determined that every additional spike contributes to the movement metrics. Much of the document is written like the microsaccade-related activity in the rostral SC is unaltered by a visual burst produced elsewhere in the SC. In the new figures added in this version, the focus is on proving that there exists a burst in the rostral SC while a visual burst occurs elsewhere in the SC. They do not test whether this burst is attenuated or statistically similar. The authors are aware of this possibility because they mention it several times in the response document, but the realization is scant in the manuscript.

If the change in microsaccade amplitude is entirely due to leakage of the visual activity, the activity in the rostral SC should be the same in the presence and absence of a visual burst. That is, the number of spikes in a window around the saccade duration should be unaltered. It is unclear whether this will be the case because, as the authors stated, the velocity profiles do not show double-peaks or inflection points to represent a separate addition of spikes from a different source. Two alternatives might be more likely and should be considered by the authors. One, the rostral SC neuron fires MORE spikes when the microsaccade is larger, which confounds the importance of the lawful relationship between additional the number of intrasaccadic visual spikes and microsaccade amplitude. Two, the rostral SC neuron fires FEWER spikes during the visual burst. Both alternatives suggest that intracollicular interactions may play a far more important role than additional intrasaccadic spikes from other parts of the SC.

The authors should analyze the activity of rostral SC neurons during microsaccades with and without a visual burst. If neural activity (either burst profiles or the number of spikes 15 ms before saccade onset to 15 ms before saccade end) is not altered, this would indicate that their lawful relationship is robust. In contrast, any alteration in rostral SC activity would support intracollicular interactions. At this point, one cannot exclude that the observed saccades is due to a newly organized population response and, crucially, one cannot differentiate between vector averaging and vector summation mechanisms; the authors favor the latter mechanism. Consequently, the lawful linear relationship might be merely an observation; it is no longer informative of any mechanism.

Even given intracollicular effects, it is possible that most of the results could be reconciled with a vector averaging scheme while also considering that averaging becomes less likely with spatial disparity. The authors should consider this possibility in their revision.

2. Eccentricity of stimulus. The dichotomy of the lawful relationship for stimulus presentation closer than or further than 4.5 deg is crucial to how the data should be interpreted. For eccentric stimulus locations (beyond 4.5 deg), there is clear evidence of bursts in both rostral and caudal SC. However, the visual spikes of caudal SC have minimal impact on microsaccade amplitude. This does not align with any reasonable model of saccade generation. If SC output, defined as number of active neurons or total number of output spikes, is relatively invariant across all saccade vectors and if visual spikes are the same as motor spikes, as the current paper asserts, then microsaccade amplitude must increase at a higher rate as the target is placed at increasingly eccentric locations. Not observing this effect implies that there may in fact be something different about "visual" and "motor" spikes, especially when the two bursts do not merge, such that visual spikes do not contribute to saccade amplitude. Concepts of motor-potent and motor-null subspaces borrowed from skeletomotor research or differences in the temporal properties of the two active populations offer viable alternatives (https://www.biorxiv.org/content/10.1101/132514v3.full). For parafoveal stimulus presentation (inside 4.5 deg), the active populations of neurons representing visual and motor bursts interact in an excitatory manner. They may effectively merge into one larger population with two local peaks, not unlike SC activity during averaging saccades of ultrashort latencies. Moreover, the combined population may take on the properties of the motor burst, which accounts for an increase in saccade amplitude with additional spike. The point is that the spatial distribution of the active population in the rostral SC is now altered, and one cannot convincingly conclude that the increase in microsaccade amplitude is merely due to more spikes added elsewhere. Given the arguments in the paragraph, it may be incorrect to state that "instantaneous readout of the SC map during movement generation, irrespective of activity source…explains a significant component of kinematic variability of motor outputs" (lines 40-42) and "the entire landscape of SC activity, not just at the movement burst site, can instantaneously contribute to individual saccade metrics" (lines 90-92).

To summarize, if visual spikes add to microsaccade amplitude "regardless of activity source"…"across the entire landscape of SC", then spikes beyond the 4.5 deg eccentricity must add more to saccade vector than spikes induced at more rostral sites. Lack of this effect might refute the authors' hypothesis.

3. Choice of saccade analysis window. It is unclear what precisely motivates the choice of window (0-20 ms after saccade onset) to assess the impact of visual spikes. Given the brief duration of microsaccades and after accounting for transduction time, the movement is nearly over by the time these spikes can impact the amplitude. Wouldn't a rigorous choice be to use the 20 to 25 ms window before saccade onset? This could maximize the impact of the visual spikes, as Figure 6 suggests.

4. Regression analysis. Readers will have difficulty with the linear regression analyses of Figure 4B and related figures. Specific comments: (a) It is questionable whether the linear regression is convincing for data beyond 4.5 deg eccentricity given the distribution of the data. The accompanying figure is an image of Figure 4 – —figure supplement 3, with the dark blue points circled in red. The linear fit does reflect the trend in the plotted points. Thus, either the regression analysis is incorrect or the illustration is inappropriate for the analysis. (b) The number of data points for 0 intrasaccadic spikes is about 5 times greater than for 1 and 2 spikes. Does this have an impact on the regression analysis output? Should another strategy be used? (c) It appears that the data for 0 intrasaccadic spikes are largely obtained from a baseline period. It is possible that the neural state during baseline could be different than around the time visual processing occurs. Thus, a stricter criterion for selection of microsaccades with 0 intrasaccadic visual spikes might be warranted.

---

## [Author Response]

[Editors’ note: the authors resubmitted a revised version of the paper for consideration. What follows is the authors’ response to the first round of review.]

As you will see from the reviews, the reviewers expressed general interests and made many positive remarks. However, they also agreed on major issues that would require a substantial amount of additional experimentation and analysis. Some of the major issues include (1) There were major concerns about the relationship and consistency between experiment 1 and 2, and the interpretation of the microstimulation experiment. (2) The results of Experiment 1 might have several other interpretations including visual enhancement and competitive interactions, and could be influenced by several confounds. Within the domain on mechanisms of saccade generation, the analyses are not detailed enough to test competing models and provide conclusive mechanistic insight.We would welcome submission of a new manuscript if it is possible to address these issues, but we judged them to be sufficiently significant that we thought it better to reject at this stage. Should you go this route, we would likely go to the same reviewers.

Thank you very much for the opportunity to resubmit our work. We have now introduced major edits to the manuscript, including the addition of extensive new experiments and new analyses that directly address all of the reviewer comments.

Below is an executive summary of our changes to the manuscript (detailed explanations are provided below in the specific responses to individual reviewers, and also in the actual revised manuscript):

1. We have now conducted substantial new experiments using linear microelectrode arrays inserted into the rostral SC (where movement bursts for microsaccades occur), the caudal SC (where visual bursts for the appearing stimuli occur), or simultaneously in both the rostral and caudal SC together. Please see the new Figures 8-11 of the revised manuscript. These experiments have allowed us to:

a. Confirm previous observations that a visual stimulus onset somewhere on the SC map is associated with reduced activity elsewhere in the map as long as there are no saccades directed towards that second SC location (e.g. Dorris et al., 2007; also see Hafed and Krauzlis, 2008)

b. Convincingly demonstrate that if the visual bursts in the SC come simultaneously with the triggering of a saccade to another site (in our case, a rostral site because of the small size of microsaccades), then the reduced activity alluded to above is actually replaced by movement bursts, confirming our presumption in the original version of the manuscript that there were two sites of elevated SC activity in our experiments. We believe that this is a very important addition to our manuscript, and to the literature as a whole (as we describe in more detail below)

c. Support our original experiments with recordings in single SC sites, because we still found that visual bursts occurred peripherally at the time of microsaccade triggering

2. We have now also added new experiments for our original visual burst analyses. This time, the stimuli used had other visual features (spatial frequency gratings versus stimulus contrast manipulations), and, more critically, there was a prolonged period of fixation after stimulus onset. Please see the new Figure 7 and Figure 7 —figure supplement 1 of the manuscript. These additional experiments were critical for the manuscript because:

a. They have allowed us to confirm that our original results simply depend on the presence of movement-unrelated spiking activity on the SC map, irrespective of the source of activity

b. They have allowed us to substantially strengthen our interpretations by looking at spiking activity long after any visual bursts (up to >1 second later). Long after visual bursts, the SC might maintain low level discharge (that can sometimes be completely absent before stimulus onset in many neurons). This has allowed us to ask whether any spiking activity during such low-level discharge would still be read out and influence movement metrics if such spiking occurred simultaneously with movement triggering. Remarkably, this was indeed the case (please see Figure 7 and Figure 7 —figure supplement 1). Therefore, one does not even need visual “bursts” to see modification of movement metrics. We consider this new analysis critical for supporting our study

c. They had sufficiently long fixation periods to allow us to demonstrate that our effects are robust enough to also be observed on a single neuron basis (please see Figure 7 and Figure 7 —figure supplement 1)

3. In the original visual burst analyses, we have now compared the relative relationship between either visual or visual-motor neurons and the final behavioral outcomes (saccade amplitudes). This was important because it allowed constraining the mechanisms of SC readout during saccades even further, and to clarify potential functional differences between visual responses in visual neurons versus visual responses in visual-motor neurons (a topic that remains wide open in the SC field)

4. Because the above experiments and analyses were extensive (they have added 5 new figures and 9 new or modified figure supplements), we have decided to defer publication of our microstimulation experiments to a later subsequent study. Otherwise, allocating sufficient space for the microstimulation experiments (in addition to the new data) would have resulted in an unnecessarily large and unwieldy paper. The additional new experiments described above are much more valuable than the microstimulation component for the current manuscript, as also suggested by the reviewers

Please note that our omission of the Dorris et al., 2007 citation in the original version was purely erroneous due to a mistake in our citation manager. We sincerely apologize for missing this error during proofreading. In fact, in our original submission, we wanted to suggest Michael Dorris as a reviewer for this manuscript, and we were only prevented from doing so because we could not find his new affiliation on the internet for inclusion in the online submission system.

In any case, we believe that all of the reviewers will be happy to see that our new microelectrode array experiments (Figures 8-11) both confirm and add to his (and others’) earlier experiments. Specifically, if no microsaccades occur at the time of peripheral visual bursts, then we do indeed observe a reduction in SC activity at the rostral sites (“lateral interaction”). However, if the peripheral visual bursts occur simultaneously with the triggering of microsaccades, then there are actually two sites of SC activity bursts: the rostral SC site for the movement bursts, and the caudal SC site for the visual bursts. We strongly believe that this is a very important addition to the literature.

Reviewer #1:This study investigates the influence of visual stimuli and spiking activity in distant SC (superior colliculus) locations on small microsaccadic movements during fixation. The authors show that visual stimuli and activity in these distant locations modulates the ongoing microsaccadic movements. The authors find however that nearby locations have more powerful effects than distant locations, which challenges existing models based on vector-averaging (as one would expect distant locations to exert more powerful effects). Overall these findings provide a very interesting addition to the literature. My main critique is the possibility of lateral interactions or influence on the activity of more foveal neurons.

Thank you very much for bringing up both of these interesting points. We have now added new experiments directly addressing both of them. Specifically, please see Figures 8-11 of the revised manuscript. We have now recorded from foveal sites, eccentric sites, and simultaneously from both foveal and eccentric sites. The new recordings in the eccentric sites confirmed all of our other analyses in the original manuscript (i.e. that visual bursts still happen intrasaccadically). The new recordings at the foveal sites confirmed that movement bursts still happen when the movement triggering coincides with the presence of eccentric spikes (this was an important addition, and we thank you for bringing it up). And, most importantly, the new simultaneous recordings confirmed that at the time of movement triggering, it is still possible to have simultaneous peripheral spikes. This means that at the time of the readout of the SC population to implement the actual saccade, additional spikes from the peripheral visual burst are present on the SC map. Our other analyses (e.g. Figures 4-6 and the new Figure 7, and their associated figure supplements) clearly demonstrate that such additional “visual” spikes do indeed matter at the time of readout (because they systematically add to saccade amplitude).

Most interestingly, the new additional experiments do not deny that there can be lateral interactions. In fact, we confirmed that if no microsaccades are triggered, then peripheral visual bursts are indeed associated with reduced rostral SC activity (please see Figures 8-11). This is consistent with prior work (e.g. Dorris et al., 2007; also see Hafed and Krauzlis, 2008, and others). However, the key difference is that our study was not focusing on the case of “no movement triggering” at the time of visual bursts (i.e. the interaction between visual bursts and “movement preparation”); rather, we were interested in what happens if there are both visual bursts and simultaneous movement triggering. In that case, one expects (and we confirmed) that the movement bursts will still happen.

1. If I understand correctly, the authors suppose that the effects on the saccade trajectory are mediated by the projections of neurons in distal locations to other brain areas. However there are lateral inhibitory interactions within the superior colliculus and the effects reported here could potentially be mediated by these lateral interactions that modulate the activity of neurons driving the microsaccadese.

The most classic and accepted view of lateral interactions would suggest that peripheral visual bursts are associated with decreased rostral SC activity. For example, in Hafed and Krauzlis, 2008, we recorded rostral SC activity at the time of a peripheral visual stimulus onset. This rostral SC activity (in the absence of microsaccades) was indeed reduced, and this is also consistent with Dorris et al., 2007 as well as other work by Munoz and colleagues (also see our confirmation of this in the current manuscript – in the absence of microsaccades – in Figures 8-11). However, a reduction of rostral SC activity would mean smaller microsaccades at the time of the flashes, not larger microsaccades like we saw.

In fact, one can modify the location of the visual flash and change the likelihood of whether to get an increase or a decrease in microsaccade amplitudes: decrease for foveal flashes (Rolfs et al., 2008) and increase for eccentric flashes (Hafed and Ignashchenkova, 2013; Buonocore et al., 2017; also confirmed in our current study). Indeed, even with foveal flashes, one can get increased microsaccade amplitudes under specific conditions related to the relative position of the movement goal and the visual flash location (Buonocore et al., 2017). All of these very disparate behavioral effects on microsaccade amplitudes (increases versus decreases) cannot be parsimoniously explained by inhibition of the rostral SC by peripheral visual bursts. Instead, they can best be explained by the readout idea that we put forward in our current study. Indeed, our new experiments (Figures 8-11) confirm that if visual bursts occur peripherally at the same time as microsaccade triggering, then rostral SC activity still exhibits movement-related bursts (and not decreases as predicted by lateral inhibition).

Therefore, when the SC map is being read out to implement the current saccade, there are two loci of elevated activity (Figure 11). Moreover, analyses like in Figures 47 demonstrate that such readout is systematically dependent on every spike emitted by every active neuron during movement execution.

In addition to the above figures (4, 7-11, and the associated figure supplements), please also see the associated text edits documenting the above ideas (e.g. lines 441-538).

To summarize, we believe that the key issue here is the idea of temporal coincidence. All we say is that if there happens to be peripheral spiking activity at the time of movement triggering (i.e. if there is a temporal coincidence of spiking), then the readout will take all simultaneous spikes into account. The individual visual bursts or movement bursts may be slightly modulated due to lateral interaction (e.g. visual enhancement and visual suppression around the time of microsaccades/saccades), but the key point is that they still happen.

2. A related point is that the visual stimulus might exert some effects directly on the neurons driving the microsaccades, either through lateral mechanisms within the retina or within the superior colliculus.The best control would be to analyze directly the neurons in foveal regions. What happens there? Is their firing rate affected or not? It appears that the authors have this data but they do not report this.

Thank you for this very important suggestion. As stated above, we have now added new experiments directly recording rostral SC activity at the time of peripheral visual bursts (please see Figures 8-11, and the associated text edits). We have now confirmed that rostral SC activity indeed still bursts for microsaccades that are temporally coincident with peripheral visual bursts in the SC.

Once again, we do not deny that there may be modulatory influences of visual stimuli on the rostral SC or vice versa. For example, we do see evidence for visual burst modulation (Figure 2 —figure supplement 1). Moreover, microsaccade motor bursts might be slightly modulated relative to when microsaccades happen in the absence of any visual transients (such slight modulation would actually be conceptually similar to modulation of saccade motor bursts when making saccades towards moving targets in the work of Goffart, Gandhi, and Keller). However, our point is simply that at the time of movement triggering, there are two loci of activity elevation on the SC map, and the locus of activity in the periphery seems to matter significantly because each additional spike by each active neuron in the periphery matters (e.g. Figures 4-7). As stated above, lateral interaction alone cannot explain the large diversity of possible behavioral effects of visual flashes on saccade metrics.

3. The authors should provide more raster plots of activity of neurons and describe more clearly how many neurons were recorded/analyzed throughout the legends and results text.

We have now gone through the entire text to clarify all of this information. We have also added raster plots in Figure 2, as also suggested by Reviewer 2. Please also note that Figure 5 and the new Figure 5 —figure supplement 1 show all spiking activity from all of our trials in the first two experiments (contrast task and spatial frequency task). These figures show spikes from each individual trial in our database used in the analysis. Finally, in Figure 7, we showed an example neuron (with raster plots) in panel A, and we also showed that our results could be observed on an individual neuron basis in panels B, C (a second example neuron is also shown in Figure 7 —figure supplement 1). Moreover, Figures 9 and 10 show 4 different example neuron results.

4. What functional cell types are recorded by the authors? This should be described.

This information was presented in Methods (line 762-765; 784-785), but we have now also added it to Results and Discussion (please see lines 267-273; 682-692).

Please note that we have now also added an interesting new analysis comparing the relationship between spiking and microsaccade amplitudes in visual versus visual-motor neurons (Figure 4 —figure supplement 1). This turned out to be very informative because it demonstrated that similar visual bursts in visual and visual-motor neurons have different correlations with saccade amplitudes. Visual responses in visual-motor neurons seem to be more relevant for saccade amplitude metric changes than visual responses in visual neurons, which is (in retrospect) consistent with several other findings in the literature (a recent example from our lab would be Chen and Hafed, 2017).

Reviewer #2:The superior colliculus (SC) is recognized for its role in sensorimotor transformations. A vast number of SC neurons discharge both when a stimulus is presented in their response fields and also when a saccade is directed to that location. Mechanisms have been proposed for differentiating 'sensory' from 'motor' activity across the SC population and, furthermore, computing the metrics and/or kinematics of the observed saccade. This study shows that if additional spikes, whether evoked by a visual stimulus or through electrical stimulation, are introduced at a SC location away from the active 'motor' population and around the time of saccade, then these spikes contribute to the size of the saccade. Amazingly, effects of adding single spikes per neuron (but from many neurons) are noticeable in the analyses. It is a thought-provoking finding and, provided several serious concerns are mitigated satisfactorily, has implications on multiple facets of SC role in saccade generation.

Thank you very much for such encouragement. We have made a faithful effort to address all of the comments, and we have added extensive new data that we believe strongly supports our original findings.

1. The data shown in Figure 2 do not align with the working model of the SC. A fairly well-known study by Dorris, Olivier, and Munoz (https://doi.org/10.1523/JNEUROSCI.4212-06.2007), which surprisingly is not cited here, showed that a visual stimulus (distractor) interferes with motor preparatory activity elsewhere in the SC. The preparatory activity is enhanced if the target and distractor populations are in close proximity but suppressed if the distractor is far way, including in the opposite hemifield. This finding – complemented by slice studies by Isa, Hall, and others -has established a framework for local excitation and mutual distal inhibition in the SC. What is shown in Figure 2 does not conform to this framework and, if true, it is too important a result to be overlooked.

We sincerely apologize for the error in citation with respect to Dorris et al., 2007. As stated above, this was a mistake in our citation manager, and we apologize once again for not catching this mistake during proofreading. We have now included citations to Dorris et al., 2007 and also Isa et al.

More importantly, we have also added new experiments in which we explicitly recorded at the movement generation sites (rostral SC) at the same time as peripheral visual bursts (please see Figures 8-11). Briefly, in the absence of microsaccades, we indeed replicated Dorris et al., 2007. That is, there was a reduction in rostral SC activity at the same time as there was a peripheral visual burst. However, critically, if a microsaccade were to be triggered at the same time as the peripheral visual burst, then the movement burst still actually happened. Therefore, our results clearly show that there are two loci of “bursts” on the SC map when microsaccades occur simultaneously with perhipheral visual bursts (please see Figures 8-11). This is, in our opinion, an important addition to the literature, as you also agree.

Please note that the keyword here is your mention of “preparatory”. We do not deny that in the absence of microsaccades, rostral SC activity would be reduced simultaneously with peripheral visual bursts (please see Figures 8-11 and our earlier work: Hafed and Krauzlis, 2008). However, if a microsaccade were to be triggered simultaneously with the visual bursts, then there will still be a motor burst instead of activity reduction. Even in Dorris et al., 2007, when an early saccade was erroneously generated by the monkeys, there was still a burst of activity instead of activity reduction. Our focus in the current study was exactly on the simultaneity of movement triggering and movement-unrelated activity.

a. I realize that the activity recorded in this study is from neurons at the "visual", rather than the "motor", site. However, if interactions between the two populations are mutual, then the effect should arguably be observed in both populations. Should we be rethinking this conclusion – are the competitive interactions not mutually effective?b. The motor activity relevant here is in the rostral SC and associated with microsaccades, while the Dorris study focused on more caudal regions that encode larger amplitude movements. However, an underlying theme of a large body of Hafed's studies is that our knowledge of large amplitude saccades and their neural control extend to microsaccades also. In fact, this is the justification used for the microstimulation experiments reported in the manuscript. Should we be second guessing this conclusion – perhaps there is something unique about microsaccade control?c. The relative timing of activity in the 'visual' and 'motor' populations are different in the two studies. I believe Dorris et al. only focused on trials when distractor-driven activity occurred during saccade preparation rather than during execution. In contrast, this Buonocore et al. manuscript draws our attention to when the two bursts are effectively coincident. But it is not clear why competitive interaction effect should disappear during a saccade.

We lump our responses here to a, b, and c above exactly because these points (a, b, c) all progressed in a logic that ultimately reached a conclusion (in c) that is directly consistent with our message: we fully agree that our focus here is on the coincidence of bursts. In our new experiments of Figures 8-11, we found that if the visual burst happens in the absence of microsaccades, rostral SC activity is indeed reduced, consistent with Dorris et al., 2007. However, when the microsaccades are triggered at the time of the visual bursts, there is still a rostral microsaccade-related burst. As a result, there are two loci of elevated activity at the time of movement triggering (i.e. at the time of readout). The most intriguing finding in this regard, as all reviewers agree, is that the elevated activity at the visual burst site clearly matters for behavioral readout, because there is a systematic relationship between the addition of individual spikes by individual active neurons and the movement metrics (e.g. Figures 4-7 and their associated figure supplements).

So, we believe that there is no contradiction with the prior literature. Rather, we believe that we add interesting new observations that extend such prior literature.

There is also no contradiction with the idea that microsaccades are similar to larger saccades. In fact, even in Dorris et al., 2007, when the monkeys erroneously triggered the saccades at the time of the “distractor”, there was a burst rather than a reduction in activity, similar to what we show here in Figures 811.

Concerning the idea that competitive interaction might disappear entirely during a saccade, this need not be the case. Motor bursts are known to be modulated by several factors. For example, when making saccades to moving targets (the work of Goffart, Gandhi, and Keller), SC motor bursts can be modulated (i.e. the burst of the intended movement still occurs but the actual movement might be modified by the moving stimulus, which can give the impression that the movement field was modified or modulated). So, it is very conceivable that both the visual and motor bursts may appear to be somewhat “modulated” at the time of movement triggering. However, the key is that they still occur and will therefore matter for readout. In fact, we know from our prior work that peripheral visual bursts can be modulated if stimulus onset comes within a specific time window from microsaccade onset (e.g. Chen et al., 2015). The difference in the current study is that we focused on rare microsaccades that are triggered right at the time of the visual bursts (i.e. slightly later than those in Chen et al., 2015). For these “rare” microsaccades, the visual bursts were still modulated by microsaccade direction in a manner consistent with Chen et al., 2015 (see Figure 2 —figure supplement 1), but the key message, once again, is that these bursts still occurred. Moreover, their occurrence systematically mattered for behavioral readout (Figures 4-7).

In other words, at the time of “gate opening” to implement a current movement, there are two bursts that are both read out by downstream populations. Competitive interactions might slightly modify both of these bursts, but they do not suddenly eliminate one of them for readout. This is exactly our message.

d. These results are also inconsistent with the large body of literature on saccadic or visual suppression in the SC (e.g., D.L. Robinson and Wurtz).

We now do a better job of clarifying that there is no inconsistency. In Figure 2, microsaccade directions were pooled together. When we separated movements according to their direction (towards versus opposite the RF location), we did, in fact, see evidence consistent with visual burst modulation as a function of microsaccade direction in our data (e.g. Figure 2 —figure supplement 1; also please see Figure 11 where the visual burst was suppressed because most microsaccade sessions were in the opposite direction from the peripheral stimulus location). Such evidence is directly consistent with Chen et al., 2015, which itself was the microsaccade variant of the early pioneering work by Robinson and Wurtz and colleagues.

So, there is no contradiction. The key point to keep in mind here is that our alignment in this study was between microsaccade triggering and visual burst onset (and, more generally, spike occurrence) and not stimulus onset. Aligning microsaccade onset to visual burst onset (as opposed to stimulus onset) allowed us to focus on a few rare movements that are triggered right at the time of SC visual burst spiking (the idea of simultaneity between two separate bursts). Such alignment was highly informative with respect to the mechanisms of SC readout for saccades.

Please also note that we added new experiments on sustained activity in the complete absence of visual bursts (Figure 7 and Figure 7 —figure supplement 1, as well as their associated text edits). We found a very similar impact of peripheral spiking activity on movement metrics. In other words, there is not even a need for a visual burst to occur in order for us to make our point. In our mind, Figure 7 is one of the most important analyses of the current manuscript, because it demonstrates that any spiking activity, irrespective of its source, can be read out if it simply happens to occur simultaneously with saccade execution.

e. It would be tremendously valuable if the authors also show what happens to the motor burst when visual spikes are added elsewhere in the SC, not just during the execution phase of a microsaccade but also in the preparatory period preceding it. Given that the Hafed lab does a lot of neural recordings in the rostral SC during microsaccades, the data may already be available or easily collectable. Without this data, it is difficult to offer full support of the results presented in this manuscript.

Thank you very much for this suggestion, which echoes suggestions by Reviewer 1. We have now explicitly added exactly these data. We have recorded, using linear microelectrode arrays, in both the rostral and caudal SC while presenting visual stimuli peripherally. Please see Figures 8-11 and their associated text descriptions. As stated above, these data were very important because:

– They confirmed that in the absence of microsaccades, peripheral visual bursts are associated with reductions in rostral SC activity (consistent with Dorris et al., 2007 and others; this also confirms our earlier observations of reductions in rostral SC activity at the onset of peripheral stimulus onsets in Hafed and Krauzlis, 2008).

– They demonstrated that with coincident triggering, there are two loci of bursts in the SC (one caudally and one rostrally).

– They also confirmed (Figure 11) that there can still be modulatory effects on the visual bursts themselves – for example, most of our sites were associated with microsaccade movement fields opposite to the peripheral visual burst location. As a result, visual bursts were suppressed (directly consistent with Chen et al., 2015; please also see Figure 2 —figure supplement 1).

2. The data in Figures 4 and 5 are rather remarkable. They make a compelling case for systematic increase in saccade amplitude as additional spikes/neuron are coincident with the motor burst. A few points of clarification:a. Please also include velocity profiles in Figure 4A. Are they bell-shaped or like control movements? Or do the velocity profiles exhibit an inflection point or multiple peaks (like that observed for the microstimulation experiments reported later in the manuscript)?

We have now added these velocity plots. Please see the new Figure 4. We have also added example eye position and velocity plots to Figure 2. Please also note that we have previously analyzed microsaccadic kinematic alterations at the time of visual flashes at length in Buonocore et al., 2017. In fact, that earlier study directly motivated our current experiments.

To directly answer your question here, the velocity curves did not exhibit multiple peaks like observed with microstimulation, so this is an important difference between the effects of visual flashes and the effects of electrical microstimulation (please see below for further thoughts concerning microstimulation).

b. As best as I can tell, the data are pooled across all trials and all neurons (like in Figure 5). If true, please make this (more) explicit in the text and figure caption. It also raises the concern that the effect may be weighted more by the subset of neurons that have substantially more trials than other neurons. Please provide summary of number of trials per neuron. Importantly, provide insights into how many neurons' data conform to the results shown in Figure 4.

We have now gone through the entire manuscript to clarify details about neuron counts, numbers of trials per neuron, and other variables (e.g. lines 763-768; 784-789; 811; 820-828).

We have also added a whole new set of experiments related to Figure 4. Specifically, in a new experiment, we have now used visual onsets consisting of different spatial frequencies rather than stimulus contrasts (the new experiment 2). The key here was to add even more neurons to our database, and to also demonstrate that the effects do not depend on a specific visual stimulus type. The results of Figure 4 held up even though not all neurons in this new experiment were shared with the original experiments (please see the new Figure 4 —figure supplement 4).

Most importantly, these new recordings in the new experiment 2 also had a prolonged fixation interval to allow us to do a much better job on the original Figure 6C, D, which was indeed based on very sparse data. The prolonged fixation interval had an important added advantage: we now had many more microsaccades during sustained fixation per individual neuron. Therefore, we could pick example neurons and replicate the new Figure 7 analysis on them. Indeed, the example neuron in Figure 7 showed results consistent with our population analyses. Figure 7 —figure supplement 1 also shows a second example neuron. We are therefore very confident that our results are consistent across the SC population, and even robust on a per-neuron basis (see Figure 2A, B). Please also see Figure 7 and its supplement.

Finally, please note that metric effects on microsaccades by visual onsets are very robust, in both humans and monkeys, as we mentioned above (e.g. Rolfs et al., 2008; Hafed and Ignashchenkova, 2013; Buonocore et al., 2017). This increases our confidence in the robustness of our results.

c. If I am mistaken and the analysis is performed one neuron at a time, then please underlay in Figure 4B the distribution of all points for each x-axis value (as in Figure S4).

No, you are correct. The variable of interest for us was the behavior (i.e. the individual microsaccade). We have clarified this throughout the manuscript (e.g. lines 220-224; 389-390; 410-412). Also, we have added a whole new experiment with additional neurons and additional analyses beyond the visual burst interval (e.g. Figure 7). This new experiment has even allowed us to demonstrate individual neuron results more convincingly (Figure 7 and Figure 7 —figure supplement 1). Finally, Figures 9, 10 show 4 different example neurons from the new experiment 3 as well.

3. Several aspects of Figure 6 are bothersome and require clarity. Frankly, it is not clear that the effect the authors want to emphasize is actually present in the data.

We have now split the figure into two separate figures: Figure 6 containing the original A, B panels, and a new Figure 7 containing completely new experiments. For Figure 6 (i.e. the original A, B panels), this figure was important to add in the study because in Figure 5, we had just arbitrarily picked a “visual burst interval” (30-100 ms after stimulus onset). This is arbitrary and does not describe the temporal relationship between individual spiking activity and amplitude changes in a more assumption-free manner. As a result, Figure 6 (i.e. the original A, B panels) became absolutely necessary in our mind, and the data in it are robust (we even added new experiments with spatial frequency gratings to support our findings; please see Figure 6 —figure supplement 1).

For Figure 7 (i.e. the original Figure 6C, D), we fully agree that the original results were too sparse. The problem was that the original experiments had too brief a fixation interval. We therefore did not have much data long after the visual bursts to analyze the potential influence of “spontaneous” spiking activity outside of the realm of visual bursts. As a result, we have now decided to add a whole new set of experiments (the “spatial frequency task”). In these experiments, the monkeys were presented with gratings of different spatial frequencies (as opposed to stimulus contrasts), and, critically, the stimulus in the RF was maintained for >1 second in every trial. As a result, we could now ask the important question of whether any spiking activity (well after the end of the visual burst) can have similar effects on movement amplitudes. This was indeed the case, as we show in Figure 7. As stated above, this effect was robust in individual neurons as well (Figure 7 and Figure 7 —figure supplement 1). Therefore, with the addition of these new data, we are quite confident that the new figure shows robust results, which we view as being perhaps the most important confirmation of our hypothesis.

a. In Figure 6C, there aren't enough points for 3 and 4 intra-saccadic spikes to justify using mean as the average measure. Median is a better choice and, in this case, the suggested effect will likely not exist. This will likely change the text on Lines 374-378, which incidentally deserves to be backed with a statistical test.b. In Figure 6C and 6D, color the dots according to the histograms in Figure 6A and 6B, respectively. Basically, the color of dots should change from dark red (green) to light orange (yellow) as y-axis value increases.c. Data in Figure 6A and 6C suggest some discrepancy. Figure 6A indicates that microsaccades larger than 1 deg were observed, but Figure 6C does not show any saccades greater than one degree. Same concern for Figures 6B and 6D.

As stated above, we fully agree that Figure 6C, D in the original manuscript had data that were too sparse, for reasons explained above. As a result, we have now added completely new data with prolonged fixation intervals. The results are now relegated to their own figures (Figure 7 and Figure 7 —figure supplement 1). We believe that there is now no inconsistency.

4. I am less excited about the experiments using microstimulation during visually-guided saccades.a. The authors want us to accept that the relationship between the sensory and motor populations of neurons remains the same in the two experimental paradigms (L424-430). I would've accepted this premise prior to reading this manuscript, but the lack of competitive interactions makes me question this assumption, particularly as the electrode placement moves away from the rostral SC.

Because we have added extensive new data (5 new figures and 9 new or modified figure supplements), we have now decided to defer our microstimulation results to a later follow-up manuscript. We sincerely hope that you agree with our choice.

Just to clarify, concerning your statement on competitive interactions, please see our detailed responses above. We do not think that there is a contradiction.

b. Example velocity traces are not unimodal. Perhaps this is due to stimulation occurring late within a saccade, but it looks more like the ongoing saccades is truncated and potentially replaced by another movement command. These results are indicative of competitive interactions.

As stated above, we have decided to defer publication of our microstimulation results to a follow-up study since the new recording experiments were extensive and added many figures (and much more support to our hypotheses). Just to clarify, we do not believe that the saccades were truncated; they reached similar peak velocity to the baseline saccades in their “first” velocity pulse. Instead, we believe that an additional velocity pulse was added by microstimulation. In any case, this is no longer being discussed in the current manuscript for the reasons described above.

5. Data in Figure 8 are difficult to interpret.a. Normalized amplitude increases earlier for 1 electrical spike compared to 2 and 3 spikes. Is this significant and how can this be? We can't interpret this result until we have a sense of variability in normalized saccadic amplitude in the absence of stimulation.b. Given the arguments raised above about Figure 7, this analysis should arguably not include displacement associated with the second bell-shape in the velocity profile.c. Saccade amplitude peaks when microstimulation and saccade onsets are coincidental, but according to Figure 7, the effect is observed as a second peak in velocity, which occurs 40-50 ms later, and this is substantially longer than the efferent delay. So, pieces are not adding up.

As stated above, we have decided to defer publication of our microstimulation results to a later manuscript, since the current one grew substantially in size with the extensive new recording data that were added. We hope that our follow-up study will provide a nice culmination of our current one and the recent Buonocore et al., 2016, 2017, which also discussed behavioral analyses of saccade kinematic alterations with visual flashes.

Reviewer #3:Overall, this is a strong manuscript that considers a topic of significant interest to those interested in visuomotor control and attention. Main strengths: 1) It provides important insight for understanding the kinematics of saccades that would be made under natural viewing circumstances and 2) good experimental design and analysis have yielded a compelling dataset that speaks to the main conclusion that currently executed SC motor commands are influenced by the instantaneous readout of any concurrent visually-evoked activity within the SC map. Weakness: Although the study provides an important additional piece to the SC readout puzzle, it does seem to strongly constrain possible mechanistic frameworks.

Thank you very much for this assessment. We have added extensive new analyses and experiments that we believe strongly constrain readout mechanisms for the SC:

1. In the new Figure 7, we have added new data with substantial periods of prolonged fixation. This has allowed us to demonstrate that any kind of spiking activity, not just within a visual “burst”, can be read out in a manner that influences saccade size if it occurs simultaneously with saccade triggering

2. In Figure 4 —figure supplement 1, we have compared the relationship between “injected” visual spikes and microsaccade amplitude for visual versus visual-motor neurons. This has implications for the functional role of otherwise very similar visual bursts in these two types of SC neurons

3. In Figure 4 —figure supplements 2 and 3, we have demonstrated that eccentricity matters for the impact of visual spikes on microsaccade amplitudes. This directly constrains the current mini-vector models (e.g. van Opstal and colleagues), because our experimental measurements show opposite effects from what would be predicted by these models

4. Most importantly, we have performed additional recordings in the rostral and caudal SC to demonstrate that there will still be rostral SC activity bursts at the time of peripheral visual bursts. This is fundamentally important to document, especially given the apparent perception by all reviewers that lateral inhibition should mean no rostral bursts at the time of the caudal visual bursts. We show that there can be lateral competition in the absence of saccade triggering, but that there are simultaneous motor and visual bursts when a saccade is triggered. This is fundamental, because it demonstrates that lateral interaction is not always sufficient to completely eliminate any spiking activity at the non-visual-burst sites.

Specific comments:The experiment in which visually evoked activity is "injected" and alters the metrics of microsaccades is elegant and compelling. To me this is what makes the manuscript worthy of publication as it shows quite definitely that such activity is read-out as part of the ongoing motor command. This has important implications for understanding the execution of saccades during natural scanning when the perception/action sequence is considerably more fluid than for a typical laboratory task. With respect to this main strength, the results are clearly presented and interpreted.The results of Experiment 2 are also well presented, but, in my view, there are some issues with interpretation that might be remedied with further discussion.The microstimulation results are interesting but think there are some issues that limit the conclusions that one might draw regarding the SC readout mechanism.1. The injection of microstimulation pulses is suggested as an analog of "injecting" visual activity. But not all SC saccade-related neurons are visually responsive, and the electrical stimulation is certain to impact visual-motor and motor cells alike. Thus, microstimulation does not so cleanly tap visually driven activity as does the visual stimulus employed in the first experiment. That microstimulation alone does not trigger saccades does not mitigate this concern since 1-3 stimulus pulses would not be expected to trigger a saccade in any case. Some consideration of this distinction seems warranted.

As mentioned above to the editors and to the other two reviewers, the addition of extensive new data and experiments (5 new figures and 9 new or modified figure supplements) has made us decide to defer publication of the microstimulation results to a later paper. This would allow us to present a much more complete microstimulation study (without compromising the current study because the new data that we have now added richly clarifies our mechanisms).

2. There have been many variants of SC readout models that are or are not consistent with producing the form of input required of brainstem local feedback circuit models as originally proposed by David Robinson and others (e.g., Jurgens et al., 1981) and which likewise either support or supplant the vector averaging scheme proposed by Sparks and colleagues (Lee et al., 1988). It is not entirely clear how the present microstimulation data inform or distinguish between traditional or later developed frameworks. For example, given the spatial configuration employed here (i.e., in line, stim always more eccentric), I could imagine how a simple vector averaging scheme could yield the linear increase in amplitude with additional pulses if one assumes the pulses produce a transient modification of the SC's desired displacement signal. But then, I think more recent schemes (e.g., that of vector summation) could work equally well. More discussion of how these data constrain current models would be welcome.

Yes, this is one more reason why it made sense for us to defer the microstimulation results to a later study, especially given the large amount of additional recording data that we have added.

Having said that, we do believe that our recording data (especially the new results in Figures 8-11 and the eccentricity analyses in Figure 4 —figure supplement 3) can be quite informative with respect to the different models. Specifically, we fully agree that the role of SC motor bursts in controlling saccades remains to be open, and we believe that our current manuscript helps to clarify some sticking points. For example, the mini-vector model (e.g. van Opstal and colleagues) would predict that more eccentric visual bursts would cause even larger effects on microsaccades than less eccentric visual bursts (i.e. 1 spike from a more eccentric neuron should contribute a larger “mini-vector” than 1 spike from a less eccentric neuron). However, this is clearly not the case in our data, so this is already a very important constraint on models (please see the new Figure 4 —figure supplement 3 explicitly parameterizing the impact of eccentricity). Similarly, our new experiments (Figures 8-11) constrain the role of lateral interaction interpretations, as discussed at length with the other two reviewers above. Briefly, it is not necessarily true that if there is a visual burst somewhere, then there can be no other burst elsewhere. Our Discussion section mentions these and other constraints at length (lines 654-677).

3. Related to the above, the microstimulation results are not unexpected based on how one would expect it to interact with an ongoing movement. This we could intuit from extant literature (as properly cited by the authors). The weakening of the effect with eccentricity is new and interesting, but I wonder if it is specific to eccentricity or if it generalizes in some way to distance in an SC spatial coordinate frame.

We are glad that you think that the microstimulation results are consistent with the literature. Our original intent was to support the recording data by injecting single, double, or triple pulse microstimulation. However, as stated above, it makes sense to defer such data to a later study. We have therefore removed the microstimulation results from the current manuscript. We believe that the new recording additions throughout the manuscript (e.g. the new Figure 7 and several analyses related to Figures 4-6) now do a better job in supporting the original recording data.

4. I understand that microsaccades, provide only the option of a more eccentric stimulus or stimulation, however, it seems like an experimental variant in which the "ectopic" stimulus was less eccentric might have been of use in distinguishing between readout mechanisms. I'm not suggesting this as a necessary addition, but some discussion might be valuable.

We have now added additional mention of this issue, as suggested. Please see, for example, lines 137-145 and 578-596. Please also note that we had previously discussed the different variants of saccadic alterations from the perspective of SC readout in our earlier work. For example, Buonocore et al., 2016 showed that saccade amplitude can be decreased with less eccentric flashes, and Buonocore et al., 2017 showed that saccade amplitude can be increased with more eccentric flashes. These two previous studies directly motivated our current manuscript, and they are now mentioned early on (e.g. near Figure 2), as well as in Discussion.

5. There is a recent paper by Gandhi and colleagues (Smalianchuk et al., 2018) that seems quite relevant as it relates to instantaneous SC readout mechanisms. Although it does not deal with "visual" activity, there is enough conceptual overlap that it should be referenced and/or discussed at the authors' discretion.

Yes, we now realize that we cited the paper erroneously as Ivan et al., 2018. This was due to an error with our citation manager, as we also explained above concerning the missing Dorris et al., 2007 citation. We apologize for that. We have now proofread all citations more carefully.

Reviewer #4:In this study, Buonocore and colleagues investigate the influence of visual neural activity in the superior colliculus (SC) on eye-movements. In the first experiment, they show that visual activity in the SC – induced by a peripheral grating – was unaffected by concurrent microsaccades. Interestingly, the size of microsaccades was biased by the visual stimulation, and the higher the number of spikes around the its onset, the larger the amplitude of the eye-movement. In a second experiment, using electrical microstimulation of the SC during a saccade task, they show that even a small number of induced spikes (3) was enough to bias the animal's eye-movement response towards the stimulated site. Overall, this is an interesting study, with significant implications for saccade generation models in the SC.The authors often use selection criteria for their data analysis that is not clearly justified. One criterion that might be particularly problematic is to focus most of their analysis on eccentricities <= 4.5 deg. Their justification for this is that "these had the strongest effects on microsaccades" (line 189). Analyses for values > 4.5 deg, showing usually just weaker effects, are presented in the supplementary materials. My main concern is that this approach might be hiding an underlying interaction between eccentricity, SC firing rate, and microsaccade amplitude, which could contradict the other findings presented. The firing rate values in Figure S4 suggest that there might be a positive correlation between eccentricity and SC firing rate (higher firing for more eccentric RFs). While microsaccade amplitudes seem negatively correlated with eccentricity. If this is true, the analysis pooling together all data would suggest that higher firing would be associated with smaller amplitudes (the opposite of what is shown here). Of course, that eccentricity could be considered a confounding variable. So, one possible way to deal with this problem would be to perform a regression analysis to predict microsaccade amplitudes using both eccentricity and SC activity as independent variables.

Thank you for this very helpful comment.

First, just to clarify, the slope of the relationship between the number of spikes and saccade amplitude was still positive at the more eccentric sites. That is, the dependence of saccade amplitude on the number of injected visual spikes was similar (more intra-saccadic spikes mean more microsaccadic amplitude), but with a weaker effect overall. So, it is not the case that our effect reverses with higher eccentricity. It is just weaker. This is an important observation to make especially because a class of current models in which each spike contributes a mini-vector (e.g. van Opstal and colleagues) would predict the opposite: they would predict that a mini-vector from a more eccentric site is larger, meaning that the same number of injected spikes should lead to a larger amplitude change.

Concerning eccentricity analyses, we fully agree, and we have now directly followed your advice. We have repeated the analysis of Figure 4B (i.e. amplitude as a function of number of injected spikes) for different eccentricity ranges across our population. The results are shown in Figure 4 —figure supplement 3. These results support our choice of 4.5 deg eccentricity as the threshold in our main analyses, and they critically demonstrate that the slope of the relationship always remains positive even at the most eccentric sites. This is important because, once again, it demonstrates that injected spikes still “add” to individual saccade amplitudes; it’s just that with more eccentric injected spikes, the addition is smaller (i.e. less effective).

The analysis presented in Figure 4 shows a clear relationship between the number of spikes in SC and microsaccade amplitude. This is a very interesting finding. However, I see two potential problems here. One, that by sorting their data on 'number of spikes', they might inadvertently be also splitting their data by contrast (a result already shown in Figure 3). Second, they count the number of spikes from saccade onset to peak velocity as a criterion. The problem is that, because of the ballistic nature of saccades, this interval would linearly increase with saccade amplitude. Given the result presented in Figure 6A, I would not expect that this would compromise their main finding, but the authors should nonetheless correct for this by using a fixed time-window across all saccade amplitude conditions.

Thank you. These are valid points. We have now addressed them using multiple ways. First, we have replaced all similar analyses with ones that count spikes in the interval of 0-20 ms after microsaccade onset (regardless of microsaccade size) as opposed to 0 to peak velocity time. Second, we have added new experiments with different visual stimuli (spatial frequency) to show that the results were not specific to the stimulus type (and therefore not specific to stimulus contrast). Third, we extended the old Figure 6C, D to look at new data with much longer sustained fixation intervals. This has allowed us to explore spiking activity completely independently of visual bursts (and therefore independently of stimulus contrast). Please see the new Figure 7. In all cases, we counted spikes in a fixed interval (0-20 ms) rather than relative to peak velocity time, as you suggested. Our choice of 20 ms was that we wanted to ensure that our measurement interval remained early within even the smallest microsaccades, especially because Figure 6A demonstrates that this is when the readout of SC spiking activity seems to take place.

The statistical inferences from experiment 2 are mostly done one "saccadic displacement", calculated as the "radial position of the eyes from saccade start to 100 ms thereafter" (line 814). I might be missing something, but I don't understand why they use this criterion instead of, for example, the normalized distance of the saccade offset to the target. It seems to me that they might yield similar results, but since most saccades are finished by 60-80ms (Figure 7CD), the 100ms zone might be contaminated by corrective eye-movements.

As stated above to the editors and to the other reviewers, we have now decided to defer the microstimulation results to a later study. Our reasoning is that the new recording data that we have added (5 new figures and 9 new or modified figure supplements) are extensive and provide better insight for the current study than that provided by the brief microstimulation results as they were originally presented. A follow up study will hopefully delve into our microstimulation results/analyses in much more detail.

[Editors’ note: what follows is the authors’ response to the second round of review.]

Essential revisions:Overall several concerns remain about the data analysis and interpretation of the findings of the authors, especially concerning the mechanistic interpretation of the findings. These mechanistic interpretations require additional analysis and further discussion or modification.

We have now added new analyses of the rostral neurons that directly address the most major comment of the reviewers (please see Figure 11 —figure supplement 1).

We have also modified the Introduction, Discussion, and other parts of the text with respect to our interpretations (e.g. please see lines 93-104, 407-417, 454-465, 595-598, 642-658, 695-699, 801-827).

We have also added several other minor analyses and figures recommended by the reviewers (copies of which are included in this document for easier parsing).

All details are provided in the responses below.

We are strongly convinced that our results and analyses represent an important contribution to the literature, and we sincerely hope that you will agree with us.

1. Rostral SC and intracollicular interactions: Concerns related to intracollicular interactions were salient in the initial round of reviews. These concerns still remain to some extent. It remains unclear whether the authors have convincingly determined that every additional spike contributes to the movement metrics. Much of the document is written like the microsaccade-related activity in the rostral SC is unaltered by a visual burst produced elsewhere in the SC. In the new figures added in this version, the focus is on proving that there exists a burst in the rostral SC while a visual burst occurs elsewhere in the SC. They do not test whether this burst is attenuated or statistically similar. The authors are aware of this possibility because they mention it several times in the response document, but the realization is scant in the manuscript.

We have now added a new figure directly comparing the rostral SC microsaccade bursts with and without the peripheral visual bursts, exactly as suggested by the reviewers. Please see the new Figure 11—figure supplement 1A. As can be seen, there was no statistically significant change in the motor bursts in the presence of the peripheral bursts. Note how panel B shows that there was still a behavioral effect of increased microsaccade size in the same experiment (despite the more eccentric stimuli than in most of our main figures from experiments 1 and 2). Please see lines 642-658.

If the change in microsaccade amplitude is entirely due to leakage of the visual activity, the activity in the rostral SC should be the same in the presence and absence of a visual burst. That is, the number of spikes in a window around the saccade duration should be unaltered. It is unclear whether this will be the case because, as the authors stated, the velocity profiles do not show double-peaks or inflection points to represent a separate addition of spikes from a different source. Two alternatives might be more likely and should be considered by the authors. One, the rostral SC neuron fires MORE spikes when the microsaccade is larger, which confounds the importance of the lawful relationship between additional the number of intrasaccadic visual spikes and microsaccade amplitude. Two, the rostral SC neuron fires FEWER spikes during the visual burst. Both alternatives suggest that intracollicular interactions may play a far more important role than additional intrasaccadic spikes from other parts of the SC.

As described above, the microsaccade bursts were not significantly altered.

We also strongly disagree with the statement that a “normal” velocity profile (no double-peaks) means that there cannot be multiple loci of activity on the SC map. The well-known global effect (Findlay, 1982), in which a saccade lands in between two simultaneously appearing targets, results in normal-looking saccades, and this effect is believed to be mediated by multiple loci of activity on the SC map (e.g. Glimcher and Sparks, 1993; Katnani et al., 2012; Vokoun et al., 2014). Even microsaccades show a version of the global effect, and the movements look normal (Hafed and Ignashchenkova, 2013). More importantly, with suprathreshold dual microstimulation of the SC at two different sites, the evoked saccades are largely normal (Katnani and Gandhi, 2011; Katnani et al., 2012). This is despite the fact that the microstimulation pulses are highly unnatural (uniform pulse trains) when compared to physiological bursts. Even when a physiological burst at one site (i.e. normal visually-guided saccade) is paired with electrical microstimulation at another site, the evoked saccades are normal (Katnani et al., 2012), but they are deviated in trajectory (from either burst location) exactly like our microsaccades are deviated in trajectory. Naturally, the saccades can be slightly off the main sequence relationship of peak velocity versus amplitude (as seen in Katnani et al., 2012), but our microsaccades also deviate from the main sequence relationship (Buonocore et al., 2017).

The example of Katnani et al., 2012 is particularly relevant here. In that study, the authors used exactly the idea of simultaneous activity on the SC map to test for predictions of SC read-out (they injected simultaneous activity to modify saccades). This is the same logic that we are invoking in our study.

Finally, please also keep in mind our results from Figure 7 and its supplement. Here, there was no peripheral burst at all. There were simply “spontaneous” spikes in the peripheral SC, and we related them to microsaccade amplitudes and still found a systematic relationship. As we stated previously, this is, in our view, perhaps the most important aspect of our study: there is no need for a visual burst at all to see our effects (Figure 7).

We have now further clarified the points above in the revised manuscript (e.g. please see lines 359-366, 695-699, 801-827).

The authors should analyze the activity of rostral SC neurons during microsaccades with and without a visual burst. If neural activity (either burst profiles or the number of spikes 15 ms before saccade onset to 15 ms before saccade end) is not altered, this would indicate that their lawful relationship is robust. In contrast, any alteration in rostral SC activity would support intracollicular interactions. At this point, one cannot exclude that the observed saccades is due to a newly organized population response and, crucially, one cannot differentiate between vector averaging and vector summation mechanisms; the authors favor the latter mechanism. Consequently, the lawful linear relationship might be merely an observation; it is no longer informative of any mechanism.

As stated above, we have now checked the rostral SC bursts with and without the peripheral bursts, exactly as suggested by the reviewers, and there was no significant alteration. Please see Figure 11—figure supplement 1A.

Please also note that the visual bursts (that are simultaneous with the motor bursts) do indeed create an altered “population response”. Specifically, read-out of the SC map by downstream structures cannot know whether a spike in one part of the map is “visual” or “motor” in label. Therefore, if there is a peripheral visual spike at the same time of SC read-out (which we show to be clearly the case; e.g. Figures 8-11), then it is indeed part of the entire population response being implemented for the saccade. This is exactly our message, and it is exactly the logic that was used in previous studies (e.g. dual microstimulation at two different sites; Katnani and Gandhi, 2011; Katnani et al., 2012). In a later comment below, we also clarify how the new ideas of Jagadisan and Gandhi, 2019 on the population temporal structure at read-out are also consistent with our interpretations.

Put another way, our results show the existence of two bursts at the time of “gate opening” to trigger a saccade (one rostral and one peripheral; e.g. Figures 8-11). Even if both bursts were to be slightly modified by each other due to intra-collicular lateral interactions (which we do not deny could happen; see Figures 8-11), it is inevitable that the visual bursts should have at least some impact at read-out. Our contribution is to show that this inevitable impact is not just to cause random variability in the saccades, but it is highly systematic and lawful (e.g. see Figure 6). Please also again remember the results of Figure 7, in which there was no peripheral visual burst at all. Here, the monkeys were simply fixating, and there was low-level discharge in the periphery. At the time of read-out at saccade triggering, whatever “spontaneous” activity that was present still had a measurable and systematic impact on the movements. This is, in our view, very striking.

Finally, we would like to clarify that we do not favor vector summation. In fact, we clearly stated that it is not entirely sufficient to understand SC read-out (please see lines 359-366, 775-793). We have now further clarified this point to avoid confusion (please see lines 801-827).

As for vector averaging, it also cannot fully account for our results. For example, at the behavioral level, even before considering any neural activity, it is clear that vector averaging should result in bigger microsaccades for the farther visual stimuli; this is not the case (e.g. Figure 3—figure supplement 1). Therefore, we do not wish to take sides between two models that both cannot fully account for our observations. Rather, our results will motivate many new interesting future research directions. For one, we are now completely revisiting the role of saccade-related bursts in the SC as a result of our observations in this manuscript, as well as others mentioned in the Discussion section. In our view, the present manuscript has opened our eyes to amazing possibilities on the mechanistic role of SC saccade bursts that we will hopefully publish soon.

So, we respectfully disagree that our results are “merely an observation”.

Even given intracollicular effects, it is possible that most of the results could be reconciled with a vector averaging scheme while also considering that averaging becomes less likely with spatial disparity. The authors should consider this possibility in their revision.

Vector averaging is definitely less likely with spatial disparity, as can be clearly seen from the behavioral data (e.g. Figure 3 —figure supplement 1).

In any case, we do agree that vector averaging is worthwhile to mention, and we did indeed discuss it in the previous versions of the manuscript. We have now done so more explicitly in the revised manuscript (e.g. please see lines 801-827).

However, as stated above, we do not wish to pick sides between vector summation and vector averaging, which both are not fully sufficient to account for our observations (or the observations of other studies, as we described in the manuscript). We also do not wish to try to add constraints to an existing model (e.g. by saying: vector averaging but with a distance constraint) just to keep the model name. We would much rather stick to our point that read-out at the time of “gate opening” for saccades is lawfully sensitive to spiking elsewhere on the SC map, regardless of the source of such spiking. As we discuss below, and in the revised manuscript (e.g. please see lines 363-366, 454-464, 809-827), we also think that the new hypotheses of Jagadisan and Gandhi, 2019 can be reconciled very well with this interpretation.

2. Eccentricity of stimulus. The dichotomy of the lawful relationship for stimulus presentation closer than or further than 4.5 deg is crucial to how the data should be interpreted. For eccentric stimulus locations (beyond 4.5 deg), there is clear evidence of bursts in both rostral and caudal SC. However, the visual spikes of caudal SC have minimal impact on microsaccade amplitude. This does not align with any reasonable model of saccade generation. If SC output, defined as number of active neurons or total number of output spikes, is relatively invariant across all saccade vectors and if visual spikes are the same as motor spikes, as the current paper asserts, then microsaccade amplitude must increase at a higher rate as the target is placed at increasingly eccentric locations. Not observing this effect implies that there may in fact be something different about "visual" and "motor" spikes, especially when the two bursts do not merge, such that visual spikes do not contribute to saccade amplitude. Concepts of motor-potent and motor-null subspaces borrowed from skeletomotor research or differences in the temporal properties of the two active populations offer viable alternatives (https://www.biorxiv.org/content/10.1101/132514v3.full). For parafoveal stimulus presentation (inside 4.5 deg), the active populations of neurons representing visual and motor bursts interact in an excitatory manner. They may effectively merge into one larger population with two local peaks, not unlike SC activity during averaging saccades of ultrashort latencies. Moreover, the combined population may take on the properties of the motor burst, which accounts for an increase in saccade amplitude with additional spike. The point is that the spatial distribution of the active population in the rostral SC is now altered, and one cannot convincingly conclude that the increase in microsaccade amplitude is merely due to more spikes added elsewhere. Given the arguments in the paragraph, it may be incorrect to state that "instantaneous readout of the SC map during movement generation, irrespective of activity source…explains a significant component of kinematic variability of motor outputs" (lines 40-42) and "the entire landscape of SC activity, not just at the movement burst site, can instantaneously contribute to individual saccade metrics" (lines 90-92).To summarize, if visual spikes add to microsaccade amplitude "regardless of activity source"…"across the entire landscape of SC", then spikes beyond the 4.5 deg eccentricity must add more to saccade vector than spikes induced at more rostral sites. Lack of this effect might refute the authors' hypothesis.

Once again, to be absolutely clear, there was no lack of effect with the more eccentric spikes; it was merely weaker. Please see, for example, Figure 3 —figure supplement 1: the microsaccades were almost doubled in size at the time of the (far) peripheral visual bursts. Similarly, Figure 4 —figure supplements 2 and 3 show that the effect was still clearly present for far visual bursts.

To further support this idea, we have now also analyzed the behavioral data from the experiments of Figures 8-11, in which the visual bursts were more eccentric than 4.5 deg. Again, with these eccentric visual stimuli, the behavioral effect was still clearly present (i.e. the microsaccades got larger in size when they occurred at the same time as the peripheral visual bursts). Please see Figure 11—figure supplement 1B.

In addition, we have now replicated Figures 5 and 6 for the far neurons (as suggested by the reviewers in later comments below), and we have found that there was still a precise temporal window of impact of the peripheral spikes, just like with the nearer neurons (please see Figure 5 —figure supplement 2 and Figure 6 —figure supplement 2).

In our opinion, the fact that our results “do not align” with existing models of saccade generation is no reason to consider them non-important. Rather, we believe that it is exactly this difference that makes our results very intriguing.

Concerning the Jagadisan and Gandhi study, thank you for pointing it out. We do like this study, and we have now discussed it in more detail because it is indeed an important and relevant study. Please see, for example, lines 363-366, 454-464, 809-827 of the revised manuscript. In fact, this study can provide a plausible explanation for the diminishing peripheral burst effects that we see with increasing eccentricity. Specifically, if peripheral visual bursts are more variable for farther eccentric neurons, which is entirely plausible given the predominantly low spatial frequency tuning of the peripheral SC neurons (Chen et al., 2018), then the likelihood of disparity between their “population temporal alignment” and the “population temporal alignment” of the motor bursts would be larger than for nearer visual bursts. This would result in weaker behavioral effects. In other words, it could be less likely that the injected peripheral visual spikes are temporally aligned with the motor spikes at read-out, and this gives these visual spikes a weaker impact on the read-out according to the Jagadisan and Gandhi hypothesis.

This idea fits very well with our observations, and we have clarified this in the text (e.g. please see lines 809-827). In fact, the new analysis of Figure 6 with our far neurons (Figure 6 —figure supplement 2) shows a very similar quantitative influence of peri-saccadic spikes to that with the near neurons. This means that once the visual spikes are properly temporally aligned with the population motor burst, an impact on read-out is inevitable (and also quantitatively similar for near and far visual bursts; please see the y-axis values in Figure 6A and Figure 6 —figure supplement 2A).

Please also note that the Jagadisan and Gandhi hypothesis primarily asks the question of why saccades are triggered at all by a neural burst. Their hypothesis is, therefore, relatively ambivalent of the implementation of the saccade itself (i.e. the question of read-out that we focus on here). So, we view our results from Figure 6 and Figure 6 —figure supplement 2 as not only supporting the temporal alignment hypothesis of Jagadisan and Gandhi, 2019, but also generalizing it to the problem of read-out and not just triggering.

In our view, our interpretation of our results vis-à-vis Jagadisan and Gandhi is a more plausible mechanism than suggesting that near visual bursts simply get merged with microsaccade bursts into one large motor burst. This merging is inconsistent with all of our studies of the foveal and peri-foveal SC. For example, in Chen et al., 2015, 2019, we deliberately checked whether “motor bursts” for microsaccades “leak” into the peri-foveal visual burst representation (as might be predicted from a merged motor burst); please also see Figure 1 —figure supplement 1. Similarly, in Buonocore et al., 2017 (please see Figures 5-8 of that particular study), we observed the very same microsaccade amplitude effects as we described in the present study even when the microsaccades and visual flashes were virtually-colocalized (the flash was literally a transient alteration in fixation spot luminance, and the microsaccades were directed towards the fixation spot). All of this evidence makes sense in retrospect: if there was a merging of population activity in the central 5 deg of the SC, then small saccades during high acuity visual behavior would not be possible. This is the opposite of the remarkable behavioral precision of microsaccades (e.g. Ko et al., 2010). In fact, the majority of saccades during natural scene viewing are smaller than 5 deg; if the central 5 deg of the SC map were all “one hill” even with distractor onsets, then the high precision of scanning saccades would not be possible.

3. Choice of saccade analysis window. It is unclear what precisely motivates the choice of window (0-20 ms after saccade onset) to assess the impact of visual spikes. Given the brief duration of microsaccades and after accounting for transduction time, the movement is nearly over by the time these spikes can impact the amplitude. Wouldn't a rigorous choice be to use the 20 to 25 ms window before saccade onset? This could maximize the impact of the visual spikes, as Figure 6 suggests.

We have now further clarified why we chose this window for some of our analyses. Please see, for example, lines 263-265, 1097-1102 in the revised manuscript.

Briefly, we fully agree that our results would look even more compelling if we had included a slightly earlier time window (e.g. please see Figure 6). However, we wanted to demonstrate, at least in some figures, that even a very strict enforcement of “simultaneity” of visual spikes with movement bursts would still result in movement modification. Therefore, by way of example in Figure 4 (and related figures), we took a very strict criterion of “intra-saccadic” spikes; however, in subsequent figures (e.g. Figures 5 and 6), we additionally showed the full time course of effects. We have also now repeated Figures 5 and 6 for the far neurons, for completeness (please see Figure 5 —figure supplement 2 and Figure 6 —figure supplement 2).

Please also note that the particular choice of 0-20 ms was made in direct response to one of the reviewer comments from the previous manuscript draft. The reviewer had suggested to pick a fixed time window to avoid the possibility that our effects were dominated by only a subset of movements with certain durations (in the end, the same results were obtained with either approach).

In any case, we believe that Figures 5, 6 (and their new and old supplements) clearly indicate the full times of influence.

4. Regression analysis. Readers will have difficulty with the linear regression analyses of Figure 4B and related figures. Specific comments: (a) It is questionable whether the linear regression is convincing for data beyond 4.5 deg eccentricity given the distribution of the data. The accompanying figure is an image of Figure 4—figure supplement 3, with the dark blue points circled in red. The linear fit does reflect the trend in the plotted points. Thus, either the regression analysis is incorrect or the illustration is inappropriate for the analysis. (b) The number of data points for 0 intrasaccadic spikes is about 5 times greater than for 1 and 2 spikes. Does this have an impact on the regression analysis output? Should another strategy be used? (c) It appears that the data for 0 intrasaccadic spikes are largely obtained from a baseline period. It is possible that the neural state during baseline could be different than around the time visual processing occurs. Thus, a stricter criterion for selection of microsaccades with 0 intrasaccadic visual spikes might be warranted.

a. We checked, and this was a mistake in the visualization script. The figure is now corrected. We apologize for this mistake. The different eccentricity bins all follow linear trends. Please see Figure 4 —figure supplement 3.

b. Similar conclusions can be reached if we do not include 0 spikes at all in our regression analyses. We have now confirmed this, and also mentioned it in the manuscript (e.g. please see lines 297-298, 1117-1121).

c. Actually, the data for 0 spikes were indeed always for microsaccades occurring within the same time windows as the microsaccades analyzed with >0 spikes. This was also true in Figure 7 and its supplement. That is, whatever the time window that we analyzed for the presence of peripheral activity at the time of microsaccade triggering, we looked for microsaccades in the same time window but without accompanying spikes. Therefore, in Figure 4, the neural state with 0 spikes was similar to that with >0 spikes (i.e. microsaccades occurring shortly after stimulus onset). Similarly, in Figure 7, the neural state was always the presence of a sustained stimulus inside the RF, irrespective of whether there was an intra-movement spike or not. We have now clarified this point further in the revised manuscript (please see lines 281-284, 1077-1080).